# Dedicated chaperones coordinate co-translational regulation of ribosomal protein production with ribosome assembly to preserve proteostasis

**Benjamin Pillet[1], Alfonso Méndez-Godoy[1†], Guillaume Murat[1†], Sébastien Favre[1], Michael Stumpe[1,2], Laurent Falquet[1,3], Dieter Kressler[1]\***

[1]Department of Biology, University of Fribourg, Fribourg, Switzerland; [2]Metabolomics and Proteomics Platform, Department of Biology, University of Fribourg, Fribourg, Switzerland; [3]Swiss Institute of Bioinformatics, University of Fribourg, Fribourg, Switzerland

**Abstract** The biogenesis of eukaryotic ribosomes involves the ordered assembly of around 80 ribosomal proteins. Supplying equimolar amounts of assembly-competent ribosomal proteins is complicated by their aggregation propensity and the spatial separation of their location of synthesis and pre-ribosome incorporation. Recent evidence has highlighted that dedicated chaperones protect individual, unassembled ribosomal proteins on their path to the pre-ribosomal assembly site. Here, we show that the co-translational recognition of Rpl3 and Rpl4 by their respective dedicated chaperone, Rrb1 or Acl4, reduces the degradation of the encoding *RPL3* and *RPL4* mRNAs in the yeast *Saccharomyces cerevisiae*. In both cases, negative regulation of mRNA levels occurs when the availability of the dedicated chaperone is limited and the nascent ribosomal protein is instead accessible to a regulatory machinery consisting of the nascent-polypeptide-associated complex and the Caf130-associated Ccr4-Not complex. Notably, deregulated expression of Rpl3 and Rpl4 leads to their massive aggregation and a perturbation of overall proteostasis in cells lacking the E3 ubiquitin ligase Tom1. Taken together, we have uncovered an unprecedented regulatory mechanism that adjusts the de novo synthesis of Rpl3 and Rpl4 to their actual consumption during ribosome assembly and, thereby, protects cells from the potentially detrimental effects of their surplus production.

**\*For correspondence:** dieter.kressler@unifr.ch

[†]These authors contributed equally to this work

**Competing interest:** The authors declare that no competing interests exist.

## Editor's evaluation

The work describes an exciting new mechanism for how r-proteins are produced in the correct abundances. Specifically, the authors find that the co-translational recognition of Rpl3/4 by their respective chaperones maintains the stability of RPL3 and RPL4 mRNAs. This mechanism is reminiscent of mechanisms of translation regulation in yeast mitochondria where oxidative phosphorylation complex assembly factors similarly regulate RNA stability and translation to ensure subunits are not produced in excess.

## Introduction

Ribosomes are the molecular machines that synthesize all cellular proteins from mRNA templates (*Melnikov et al., 2012*). Eukaryotic 80S ribosomes are made up of two unequal ribosomal subunits (r-subunits): the small 40S and the large 60S r-subunit. In the yeast *Saccharomyces cerevisiae*, the 40S

**eLife digest** Living cells are packed full of molecules known as proteins, which perform many vital tasks the cells need to survive and grow. Machines called ribosomes inside the cells use template molecules called messenger RNAs (or mRNAs for short) to produce proteins. The newly-made proteins then have to travel to a specific location in the cell to perform their tasks. Some newly-made proteins are prone to forming clumps, so cells have other proteins known as chaperones that ensure these clumps do not form.

The ribosomes themselves are made up of several proteins, some of which are also prone to clumping as they are being produced. To prevent this from happening, cells control how many ribosomal proteins they make, so there are just enough to form the ribosomes the cell needs at any given time. Previous studies found that, in yeast, two ribosomal proteins called Rpl3 and Rpl4 each have their own dedicated chaperone to prevent them from clumping. However, it remained unclear whether these chaperones are also involved in regulating the levels of Rpl3 and Rpl4.

To address this question, Pillet et al. studied both of these dedicated chaperones in yeast cells. The experiments showed that the chaperones bound to their target proteins (either units of Rpl3 or Rpl4) as they were being produced on the ribosomes. This protected the template mRNAs the ribosomes were using to produce these proteins from being destroyed, thus allowing further units of Rpl3 and Rpl4 to be produced. When enough Rpl3 and Rpl4 units were made, there were not enough of the chaperones to bind them all, leaving the mRNA templates unprotected. This led to the destruction of the mRNA templates, which decreased the numbers of Rpl3 and Rpl4 units being produced.

The work of Pillet et al. reveals a feedback mechanism that allows yeast to tightly control the levels of Rpl3 and Rpl4. In the future, these findings may help us understand diseases caused by defects in ribosomal proteins, such as Diamond-Blackfan anemia, and possibly also neurodegenerative diseases caused by clumps of proteins forming in cells. The next step will be to find out whether the mechanism uncovered by Pillet et al. also exists in human and other mammalian cells.

r-subunit is composed of the 18S ribosomal RNA (rRNA) and 33 ribosomal proteins (r-proteins), while the 60S r-subunit contains 3 rRNA species (25S, 5.8S, and 5S) and 46 r-proteins (*Melnikov et al., 2012*). Accordingly, the making of ribosomes corresponds to a gigantic molecular jigsaw puzzle, which, when accurately pieced together, results in the formation of translation-competent ribosomes. Our current understanding of ribosome biogenesis is mostly derived from studying this multistep assembly process in the model organism *S. cerevisiae*. An exponentially growing yeast cell contains ~200,000 ribosomes and, with a generation time of 90 min, needs to produce more than 2000 ribosomes per minute, thus, requiring the synthesis of at least ~160,000 r-proteins per minute (*Warner, 1999*). Given the enormous complexity of the process, it is not surprising that a plethora (>200) of mostly essential biogenesis factors is involved to ensure its fast and faultless completion (*Kressler et al., 2010*; *Woolford and Baserga, 2013*; *Kressler et al., 2017*; *Pena et al., 2017*; *Bassler and Hurt, 2019*; *Klinge and Woolford, 2019*). While atomic structures of eukaryotic ribosomes have already been obtained 10 years ago (*Ben-Shem et al., 2011*; *Klinge et al., 2011*; *Rabl et al., 2011*), recent advances in cryo-EM have now enabled to solve high-resolution structures of several distinct pre-ribosomal particles, thereby starting to provide a detailed molecular view of ribosome assembly (*Greber, 2016*; *Kressler et al., 2017*; *Pena et al., 2017*; *Bassler and Hurt, 2019*; *Klinge and Woolford, 2019*).

The early steps of ribosome synthesis take place in the nucleolus where the rDNA genes are transcribed into precursor rRNAs (pre-rRNAs). Three of the four rRNAs (18S, 5.8S, and 25S) are transcribed by RNA polymerase I (RNA Pol I) into a 35S pre-rRNA, which undergoes covalent modifications and endo- and exonucleolytic cleavage reactions (*Watkins and Bohnsack, 2012*; *Fernández-Pevida et al., 2015*; *Turowski and Tollervey, 2015*), whereas the fourth rRNA (5S) is transcribed as a pre-5S rRNA by RNA Pol III. The stepwise association of several biogenesis modules, additional biogenesis factors, and selected small-subunit r-proteins with the nascent 35S pre-rRNA leads to the formation of the 90S pre-ribosome. Then, endonucleolytic cleavage of the pre-rRNA separates the two assembly paths and gives rise to the first pre-40S and pre-60S particles, which are, upon further maturation, exported to the cytoplasm where they are converted into translation-competent 40S and 60S r-subunits (*Kressler et al., 2017*; *Pena et al., 2017*; *Bassler and Hurt, 2019*; *Klinge and Woolford, 2019*).

To sustain optimal rates of ribosome assembly, each of the 79 r-proteins must be produced in an assembly-competent amount that, at least, matches the abundance of the newly synthesized 35S pre-rRNA. This enormous logistic task is complicated by the fact that 59 r-proteins are synthesized from duplicated r-protein genes (RPGs) and that most primary RPG mRNA transcripts (102 of 138) contain introns (*Planta and Mager, 1998*; *Woolford and Baserga, 2013*). As a first mechanism to ensure the roughly equimolar supply of each r-protein, RPG transcription is regulated such that the output for each of the 79 RPG mRNAs, regardless of whether derived from a single-copy or duplicated RPG, is within a similar range (*Zeevi et al., 2011*; *Knight et al., 2014*). This co-regulation of the three different RPG promoter types is mediated by the complementary action of the two TORC1-controlled transcription factors Ifh1 and Sfp1, which are either mainly required for activation of category I and II (Ifh1) or category III (Sfp1) promoters (*Zencir et al., 2020*; *Shore et al., 2021*). Moreover, RPG transcription is also coordinated with RNA Pol I activity via Utp22-dependent sequestration of Ifh1 in the CURI complex (*Albert et al., 2016*). However, transcriptional harmonization is likely not sufficient because the quantitative and qualitative production of r-proteins is influenced by additional parameters, such as the stability and translatability of the different RPG mRNAs as well as the intrinsic stability and aggregation propensity of each individual r-protein. Despite their difficult structural characteristics and highly basic nature, which make them susceptible for aggregation (*Jäkel et al., 2002*), r-proteins are nevertheless, as shown in mammalian cells, continuously produced beyond their actual consumption in ribosome assembly (*Lam et al., 2007*). Apparently, cells can readily cope with a moderate excess of unassembled r-proteins in the nucleus as these are selectively recognized and ubiquitinated by the conserved E3 ubiquitin ligase Tom1 (ERISQ pathway) and subsequently degraded by the proteasome (*Sung et al., 2016a*; *Sung et al., 2016b*). However, when orphan r-proteins are more excessively present, owing to a severe perturbation of ribosome assembly, and start to aggregate, a stress response pathway, termed RASTR or RPAS, is activated, which alleviates the proteostatic burden by upregulating Hsf1-dependent target genes and downregulating RPG transcription (*Albert et al., 2019*; *Tye et al., 2019*).

In order to not unnecessarily strain cellular proteostasis under normal growth conditions, cells have evolved general as well as highly specific mechanisms to protect newly synthesized r-proteins from aggregation and safely guide them to their pre-ribosomal assembly site. For instance, the two ribosome-associated chaperone systems, the RAC-Ssb chaperone triad and the nascent polypeptide-associated complex (NAC) (*Zhang et al., 2017*; *Deuerling et al., 2019*), functionally cooperate to promote the soluble expression of many r-proteins (*Koplin et al., 2010*). However, most r-proteins associate with pre-ribosomal particles in the nucle(ol)us; thus, their risky journey does not end in the cytoplasm. Despite their small size, nuclear import of r-proteins largely depends on active transport mediated by importins (*Rout et al., 1997*; *Bange et al., 2013*; *de la Cruz et al., 2015*), which exhibit, likely by recognizing and shielding the exposed rRNA-binding regions of r-proteins, a dual function as transport receptors and chaperones (*Jäkel et al., 2002*; *Melnikov et al., 2015*; *Huber and Hoelz, 2017*). Besides being assisted by these general mechanisms, some r-proteins also rely on tailor-made solutions. For instance, 9 of the 79 r-proteins are transiently associated with a selective binding partner belonging to the heterogeneous class of dedicated chaperones (*Espinar-Marchena et al., 2017*; *Pena et al., 2017*; *Pillet et al., 2017*). These exert their beneficial effects by, for example, already capturing the nascent r-protein client in a co-translational manner (*Pausch et al., 2015*; *Pillet et al., 2015*; *Black et al., 2019*; *Rössler et al., 2019*), coupling the co-import of two r-proteins with their ribosomal assembly (*Kressler et al., 2012*; *Calviño et al., 2015*), or facilitating the nuclear transfer from an importin to the assembly site (*Schütz et al., 2014*; *Ting et al., 2017*). In addition, some r-proteins regulate their own expression levels through autoregulatory feedback loops, for example, by repressing translation, inhibiting splicing, or promoting degradation of their own (pre-) mRNA (*Fewell and Woolford, 1999*; *Gudipati et al., 2012*; *Johnson and Vilardell, 2012*; *He et al., 2014*; *Gabunilas and Chanfreau, 2016*; *Petibon et al., 2016*; *Roy et al., 2020*).

In this study, we show that a common regulatory machinery subjects the *RPL3* and *RPL4* mRNAs to co-translational downregulation when the dedicated chaperone Rrb1 or Acl4 is not available for binding to nascent Rpl3 or Rpl4, respectively. Central to the here-described regulatory mechanism is the Caf130-mediated connection between the NAC and, via the N-terminal domain of Not1, the Ccr4-Not complex, which is assembled around the essential Not1 scaffold and implicated in many aspects of mRNA metabolism, notably including cytoplasmic mRNA degradation (*Parker, 2012*;

*Collart, 2016*). The tight regulation of Rpl3 and Rpl4 levels appears to be of physiological relevance as their deregulated expression in cells lacking Tom1 leads to their massive aggregation and cell inviability. Taken together, our data indicate that this novel, co-translational regulatory mechanism specifically operates to continuously adjust the expression levels of Rpl3 and Rpl4 to their actual consumption during ribosome assembly, thereby avoiding that their surplus production might negatively affect cellular proteostasis.

## Results

### The growth defect of *Δacl4* cells is suppressed by the absence of Caf130, Cal4, and the nascent polypeptide-associated complex

We and others have previously shown that the dedicated chaperone Acl4 associates with the r-protein Rpl4 in a co-translational manner and protects Rpl4 from aggregation and degradation on its path to its assembly site on nucleolar pre-60S particles (*Pillet et al., 2015*; *Stelter et al., 2015*; *Sung et al., 2016a*; *Huber and Hoelz, 2017*). While growing *Δacl4* null mutant cells on YPD plates, we observed that spontaneous suppressors of the severe slow-growth (sg) phenotype arose at a relatively high frequency (*Figure 1—figure supplement 1A*). Since mild overexpression of Rpl4a from a centromeric plasmid almost completely restored the *Δacl4* growth defect (*Pillet et al., 2015*), we hypothesized that the *Δacl4* suppressor mutations might either increase the expression level or stability of Rpl4 or facilitate the incorporation of Rpl4 into pre-60S particles. To unravel the reason for this observation, we isolated a large number of *Δacl4* and *Δacl4/Δrpl4a* suppressors and identified causative candidate mutations by whole-genome sequencing (see Materials and methods). Bioinformatics analysis of the sequenced genomes revealed that the 48 independent suppressors harbored 47 different candidate mutations, which mapped to only four different genes: *CAF130* (35 different mutations), *YJR011C/CAL4* (7), *NOT1* (4), and *RPL4A* (1) (see *Supplementary file 3*). Notably, Caf130 is a sub-stoichiometric subunit of the Ccr4-Not complex (*Chen et al., 2001*; *Nasertorabi et al., 2011*) and, as shown below, interacts directly with the previously uncharacterized Yjr011c, which we have accordingly named Cal4 (*C*af130-*a*ssociated regulator of Rp*l4*). Given that the suppressor screen yielded early frameshift mutations in both *CAF130* and *CAL4*, we first tested whether their complete deletion would restore the severe growth defect of *Δacl4* cells. As shown in *Figure 1A and B*, this was indeed the case; however, while both the absence of Caf130 and Cal4 restored growth of *Δacl4* cells virtually to the wild-type extent at 16, 23, and 30°C, only the *Δcal4/Δacl4* double mutant combination grew well at 37°C as the single *Δcaf130* mutant already exhibited a temperature-sensitive (ts) phenotype (*Figure 1A and B*).

Considering that both Caf130 and Cal4 have been suggested to be physically connected with Btt1, the minor β-subunit of NAC, and the NAC α-subunit Egd2 by previous studies (*Ito et al., 2001*; *Krogan et al., 2006*; *Cui et al., 2008*), we next explored this potential link to the co-translational sensing of nascent polypeptides by assessing whether the absence of either of the two NAC subunits would restore the growth defect of *Δacl4* cells. While absence of Btt1 (*Δbtt1*) resulted in a modest growth amelioration of *Δacl4* cells at 23 and 30°C, full suppression could be observed at 37°C; however, no restoration of the growth defect could be discerned at 16°C (*Figure 1C*). Given that there was no suppression at any of the tested temperatures when *Δacl4* cells were lacking the major NAC β-subunit Egd1 (*Δegd1*) (*Figure 1D*), we tested whether the complete absence of NAC-β (*Δegd1*, *Δbtt1*) would enhance the extent of suppression. Indeed, a very robust suppression of the *Δacl4* growth defect could be witnessed from 16 to 30°C (*Figure 1E*); but, in line with the ts phenotype of *Δegd1/Δbtt1* double mutant cells (*Figure 1—figure supplement 1B*), there was no mutual suppression of the respective growth defects at 37°C in *Δegd1/Δbtt1/Δacl4* triple mutant cells. Similarly, absence of NAC-α (*Δegd2*), which conferred a ts phenotype, also rescued the *Δacl4* growth defect to the wild-type extent at temperatures up to 30°C (*Figure 1F*). In support of a specific role of NAC, deletion of Zuo1 (*Δzuo1*), a component of the ribosome-associated RAC-Ssb chaperone triad, did not enable suppression of the *Δacl4* growth defect (*Figure 1—figure supplement 1C*).

We conclude that the absence of either the accessory Ccr4-Not component Caf130, the previously uncharacterized Cal4, or the NAC compensates for the lack of Acl4, suggesting that these factors may be part of a regulatory network controlling the expression levels of Rpl4. Moreover, with respect to NAC's two paralogous β-subunits, the suppression analyses indicate that the Btt1-containing NAC

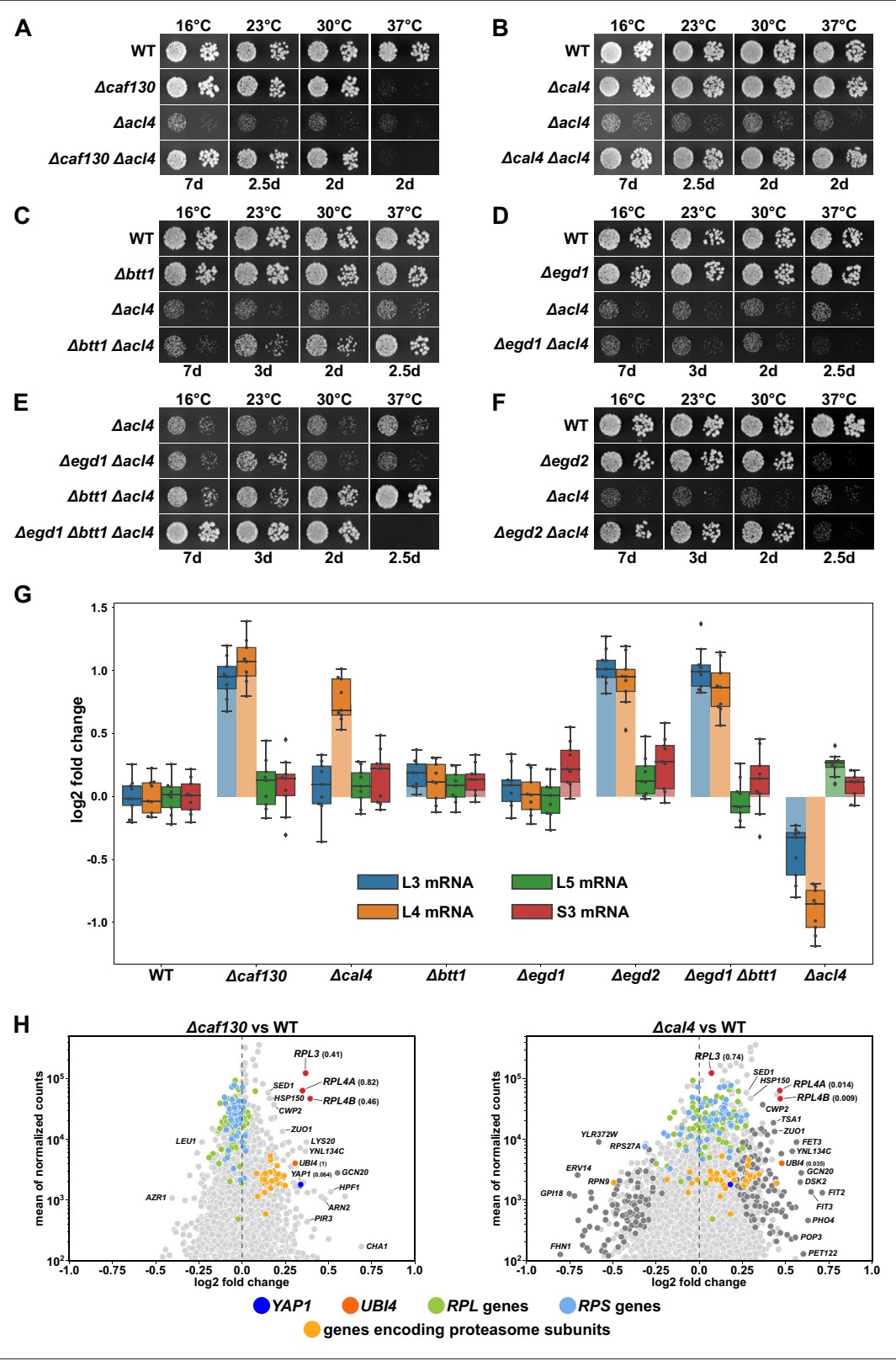

**Figure 1.** Absence of Caf130, Cal4, or the nascent polypeptide-associated complex (NAC) suppresses the *Δacl4* growth defect by increasing *RPL4* mRNA levels. (**A–F**) Suppression of the *Δacl4* growth defect. The indicated wild-type (WT), single, double, and triple deletion strains, all derived from tetratype tetrads, were spotted in 10-fold serial dilution steps onto YPD plates, which were incubated for the indicated times at 16, 23, 30, or 37°C. (**G**)

*Figure 1 continued on next page*

*Figure 1 continued*

Cells lacking Caf130, Cal4, or the NAC exhibit increased *RPL4* mRNA levels. Cells of the indicated genotype were grown in YPD medium at 30°C to an $OD_{600}$ of around 0.6, and relative changes in mRNA levels were determined by qRT-PCR (see Materials and methods). The data shown were obtained from three independent strains of the same genotype (biological triplicates), in each case consisting of a technical triplicate. The darker-colored boxes highlight the quartiles of each dataset, while the whiskers indicate the minimal and maximal limits of the distribution; outliers are shown as diamonds. The horizontal line in the quartile box represents the median log2 fold change of each dataset. (**H**) Christmas tree representation of differential gene expression analysis between *Δcaf130* (left panel) or *Δcal4* (right panel) and WT cells. The RNA-seq data were generated from the same total RNA samples used for the above qRT-PCRs. Genes exhibiting statistically significant differential mRNA levels are colored in dark gray (adjusted p-value, padj<0.05). The adjusted p-values for the selected mRNAs are indicated in parentheses. Categories of genes or specific genes, regardless of the adjusted p-value, are colored as indicated.

The online version of this article includes the following figure supplement(s) for figure 1:

**Figure supplement 1.** Differential gene expression analysis between nascent polypeptide-associated complex (NAC)-deficient and wild-type (WT) cells.

**Figure supplement 2.** Absence of Caf130 does not lead to increased transcription of the *RPL3*, *RPL4A*, and *RPL4B* genes.

---

heterodimer provides the main contribution, especially at elevated temperature, although Egd1-containing NAC appears to operate in a partially redundant manner, as evidenced by the finding that full *Δacl4* suppression at temperatures below 37°C can only be observed when both NAC β-subunits are simultaneously absent.

## *RPL4* mRNA levels are increased in the absence of Caf130, Cal4, and the nascent polypeptide-associated complex

To obtain insight into how the above-described components might regulate Rpl4 expression, we first compared the total *RPL4* mRNA levels between wild-type and mutant cells, grown in YPD medium at 30°C, by quantitative reverse transcription PCR (qRT-PCR). In good correlation with the suppression efficiency, we observed an about twofold relative increase of the *RPL4* mRNA levels in *Δcaf130*, *Δcal4*, *Δegd2*, and *Δbtt1/Δegd1* mutant cells but no increase in *Δegd1* and *Δbtt1* cells (**Figure 1G**). Given that mild overexpression of Rpl4a efficiently restores the *Δacl4* growth defect (**Pillet et al., 2015**), the moderate rise in *RPL4* mRNA levels likely accounts for the observed suppression. To evaluate the specificity of this upregulation, we next determined the levels of the *RPL3*, *RPL5*, and *RPS3* mRNA. While there were only minor changes in the abundance of the *RPL5* and *RPS3* mRNAs, the *RPL3* mRNA exhibited a similar increase as the *RPL4* mRNA in *Δcaf130*, *Δegd2*, and *Δbtt1/Δegd1* mutant cells; conspicuously, however, the absence of Cal4 did not augment *RPL3* mRNA levels, indicating that Cal4 may be specifically required for the regulation of the *RPL4* mRNA. Notably, the inverse effect was observed in *Δacl4* cells, which exclusively displayed decreased levels of the *RPL4* and, to a lesser extent, the *RPL3* mRNA, suggesting that co-translational capturing of nascent Rpl4 by Acl4 may have a positive impact on the abundance of the *RPL4* mRNA (see below).

To discern whether altered transcription initiation or mRNA stability could be the reason for the observed changes in *RPL3* and/or *RPL4* mRNA abundance, we assessed RNA Pol II occupancy around the transcription start site (TSS) of several RPGs by chromatin immunoprecipitation (ChIP) and qPCR in wild-type and *Δcaf130* cells (see Materials and methods). In support of mRNA stability being the responsible feature, absence of Caf130 did not lead to an increased association of initiating RNA Pol II on the *RPL3* and *RPL4A/B* promoters (**Figure 1—figure supplement 2A**). Moreover, while inactivation of TORC1 by rapamycin treatment similarly reduced transcription of all tested RPGs both in wild-type and *Δcaf130* cells (**Figure 1—figure supplement 2A**), *RPL3* and *RPL4* mRNA levels nevertheless remained around twofold higher in cells lacking Caf130, suggesting that the *RPL3* and *RPL4* mRNAs, even when present at lower abundance, are still subjected to negative regulation under these conditions in wild-type cells (**Figure 1—figure supplement 2B**).

Since the above results indicated that the regulation mediated by Caf130, Cal4, and NAC may only operate on a limited number of common mRNAs, we wished to obtain a global overview of the regulated transcripts. To this end, we assessed, using the same total RNA extracts as for the above qRT-PCRs, the relative abundance of individual mRNAs within the entire transcriptome by RNA-seq (see

Materials and methods). Strikingly, when compared to the levels in wild-type cells, the *RPL3* mRNA and both *RPL4* mRNAs, transcribed from the paralogous *RPL4A* and *RPL4B* genes, were amongst the most prominently upregulated transcripts in Δcaf130 cells (**Figure 1H**; see also **Supplementary file 4**). In line with the above qRT-PCR data, only the *RPL4A* and *RPL4B* transcripts, but not the *RPL3* mRNA, belonged to the markedly upregulated transcripts in Δcal4 cells (**Figure 1H**). Individual deletion of NAC-α (Δegd2) or NAC-β (Δegd1, Δbtt1) also resulted in an observable, albeit less outstanding, upregulation of the *RPL3*, *RPL4A*, and *RPL4B* mRNAs (**Figure 1—figure supplement 1D**), presumably due to more pronounced global changes in their transcriptomes. A common feature of all four mutant transcriptomes, although to a lesser extent in the one of Δcal4 cells, appears to be the upregulation of transcripts encoding components of stress response pathways, including, for example, proteins of the ubiquitin-proteasome system (UPS), the transcription factor Yap1, which is known to mediate oxidative stress tolerance, or proteins involved in iron uptake and homeostasis. On the other hand, the downregulated transcripts are more diverse, but often belong to different anabolic processes that mediate cell growth, such as translation (e.g., genes coding for r-proteins and biogenesis factors), the provisioning of building blocks (e.g., genes coding for permeases and enzymes involved in amino acid synthesis), and mitochondrial metabolism. Importantly, no other RPG transcripts were found to be upregulated in the same manner as the *RPL3*, *RPL4A*, and *RPL4B* mRNAs, suggesting that these three are specific common targets of Caf130 and the NAC, while Cal4 only contributes to the negative regulation of the two *RPL4* mRNAs.

## The full-length translational isoform of Not1 enables negative regulation of *RPL3* and *RPL4* mRNA levels

Encouraged by the above results, we next examined the involvement of Not1, the largest subunit and scaffold protein of the Ccr4-Not complex (**Collart, 2016**), in the regulatory process. Intriguingly, the four identified Δacl4 suppressor mutations, even though *NOT1* is an essential gene (**Collart and Struhl, 1993**), either change the start codon (M1L), introduce a premature stop codon (L112*), or result in early frameshifts (K21[fs] and I128[fs]) (**Figure 2A**); they are therefore predicted to interfere with the synthesis of a functional Not1 protein. Moreover, Western analysis of C-terminally TAP-tagged Not1, expressed from the genomic locus, consistently resulted in the detection of two Not1-TAP bands (**Figure 2B**); hence, the shorter, major Not1 isoform must correspond, as also previously suggested (**Liu et al., 1998**), to an N-terminally truncated Not1 protein, which could either be generated from different mRNA isoforms or by an alternative translation initiation event. In support of the second possibility, only a single *NOT1* mRNA species was detected in a previous study (**Collart and Struhl, 1993**). Notably, the *NOT1* sequence does not contain any out-of-frame ATG trinucleotides between the start codon and the second in-frame ATG coding for M163, strongly suggesting that a leaky scanning mechanism enables the synthesis of the N-terminally truncated Not1 variant. To experimentally corroborate this plausible conjecture, we mutated the *NOT1* coding sequence by either changing codon 163 such that it codes for another amino acid (construct M163L and M163A) or introducing an out-of-frame ATG trinucleotide at two different positions by silent mutagenesis of codons 40 and 156 (construct N40(oofATG) and N156(oofATG)). These plasmid-borne constructs, expressing the four C-terminally TAP-tagged Not1 variants from the *NOT1* promoter, were transformed into a *NOT1* shuffle strain. Then, upon plasmid shuffling on 5-fluoroorotic acid (5-FOA)-containing plates, complementation was assessed by growth assays on YPD plates. Importantly, all four Not1 variants sustained growth in the absence of endogenous Not1 equally well as wild-type Not1-TAP (**Figure 2—figure supplement 1A**). Western analysis of total protein extracts prepared by an alkaline lysis protocol, using antibodies recognizing the protein A moiety of the TAP tag, revealed that Not1-TAP was expressed at higher levels from plasmid than from the genomic locus. Despite the slightly changed start context owing to the introduction of an NdeI site (tac-ATG versus cat-ATG), expression of Not1-TAP from plasmid still resulted in the detection of a full-length and an N-terminally truncated isoform at similar ratios as when expressed from the native context (**Figure 2B**). In line with ATG codon 163 being the second translation initiation site, only the upper band corresponding to full-length Not1-TAP persisted in the M163L and M163A mutant variants. Concerning the two variants containing out-of-frame ATG trinucleotides upstream of the M163 codon, the N40(oofATG) and, to a lesser extent, the N156(oofATG) construct suppressed the synthesis of the major, N-terminally truncated Not1 isoform, presumably reflecting the relative strength of the two ATG contexts as translation initiation signals.

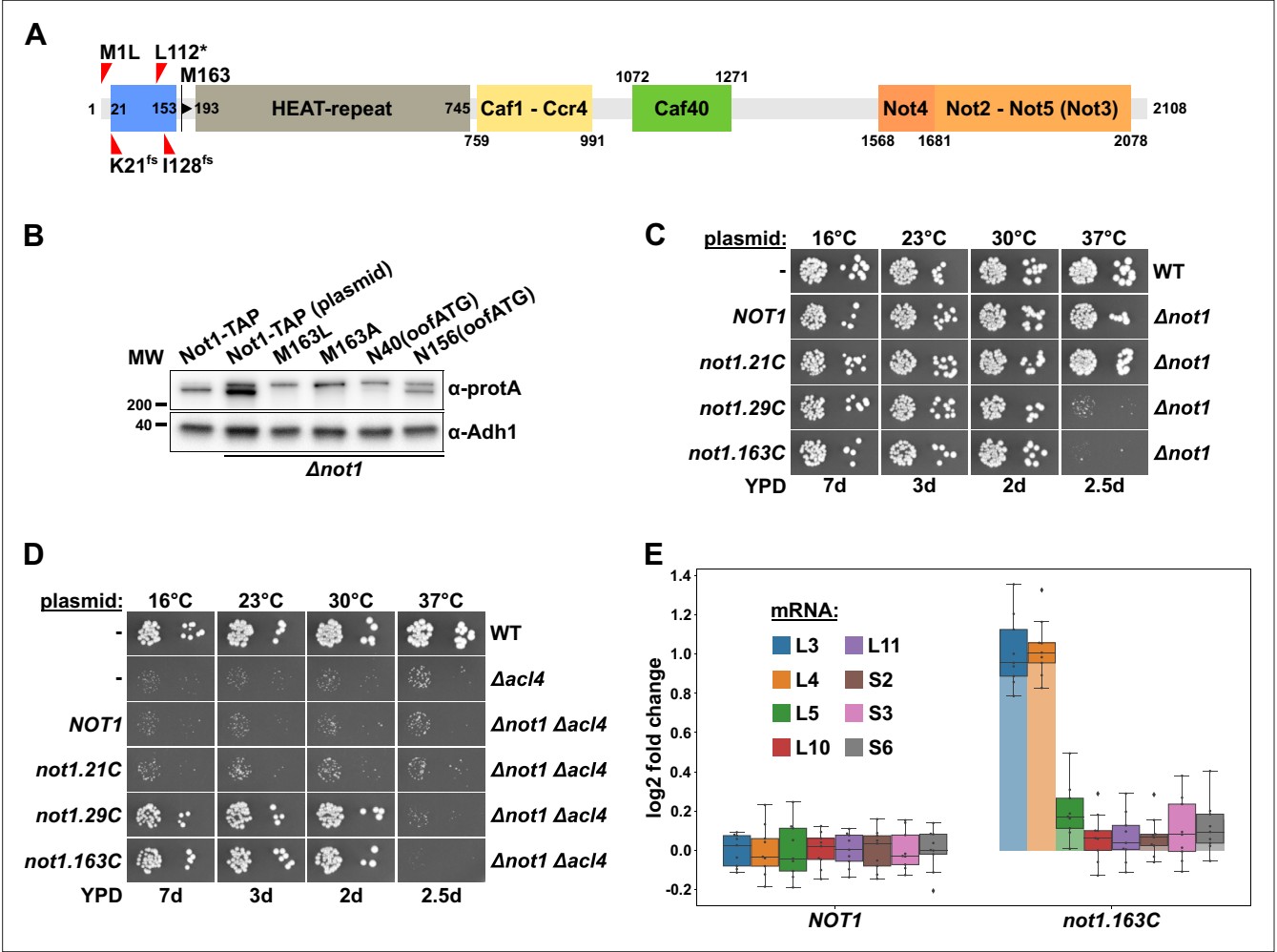

**Figure 2.** Absence of Not1's N-terminal domain suppresses the *Δacl4* growth defect and increases *RPL3* and *RPL4* mRNA levels. (**A**) Schematic representation of Not1 highlighting its domain organization and known binding sites of Ccr4-Not core components as revealed by diverse (co-)crystal structures (PDB: 4B8B and 4B8A [*Basquin et al., 2012*], 4CV5 [*Mathys et al., 2014*], 5AJD [*Bhaskar et al., 2015*], and 4BY6 [*Bhaskar et al., 2013*]). As shown in *Figure 3H*, the N-terminal Not1 segment encompassing amino acids 21–153 corresponds to the minimal Caf130-interacting domain (CaInD). Note that Ccr4 does not directly bind to Not1, it is recruited via its interaction with Caf1. The position and nature of the *Δacl4* suppressor mutations are indicated: M1L (ATG start codon changed to cTG), K21$^{fs}$ (AAA codon with deletion of one A, resulting in a frameshift), L112* (TTG codon changed to TaG stop codon), and I128$^{fs}$ (ATT codon with A deleted, resulting in a frameshift). M163 denotes the second methionine within Not1, it is encoded by the first occurring ATG trinucleotide after the start codon. (**B**) The shorter, major isoform of Not1 is generated by utilization of the ATG coding for M163 as the start codon. Total protein extracts, derived from cells expressing Not1-TAP, either from the genomic locus or from plasmid in a *Δnot1* strain, and the indicated variants, were analyzed by Western blotting using anti-protA and anti-Adh1 (loading control) antibodies. The N40(oofATG) and N156(oofATG) constructs contain an out-of-frame ATG (oofATG) owing to the silent mutagenesis of the N40 and N156 codons from AAC to AAt, which, together with the first position of the subsequent Asp-encoding codons, forms an ATG trinucleotide. (**C, D**) Growth phenotype of and suppression of the *Δacl4* growth defect by N-terminal deletion variants of Not1. Plasmids harboring full-length *NOT1* or the indicated *not1* deletion variants, expressed under the control of the *NOT1* promoter, were transformed into a *NOT1* shuffle strain (**C**) or a *NOT1/ACL4* double shuffle strain (**D**). After plasmid shuffling on 5-fluoroorotic acid (5-FOA)-containing plates, cells were restreaked on YPD plates and then, alongside a wild-type (WT) and *Δacl4* control strain, spotted in 10-fold serial dilution steps onto YPD plates. Note that the *not1.21C*, *not1.29C*, and *not1.163C* alleles express N-terminally truncated Not1 variants starting at amino acids 21, 29, and 163, respectively. (**E**) Absence of Not1's N-terminal domain increases *RPL3* and *RPL4* mRNA levels. Relative changes in mRNA levels between *Δnot1* cells complemented with either plasmid-borne *NOT1* or *not1.163C* were determined by qRT-PCR. Cells were grown in YPD medium at 30°C. The data shown were obtained with three independent *NOT1* shuffle strains (biological triplicates), in each case consisting of a technical triplicate, and they are represented as described in the legend to *Figure 1G*.

The online version of this article includes the following source data and figure supplement(s) for figure 2:

**Source data 1.** Original image files of the Western blots shown in *Figure 2B*, including a PDF file showing the full blots and indicating the cropped areas.

**Figure supplement 1.** Absence of other Ccr4-Not components does not suppress the *Δacl4* growth defect.

**Figure supplement 2.** Absence of general mRNA decay factors does not suppress the *Δacl4* growth defect.

We conclude that in *S. cerevisiae* Not1 is naturally synthesized as two distinct protein isoforms, which differ, due to a leaky scanning mechanism enabling the utilization of a downstream translation initiation site, by the presence (less abundant, full-length isoform) or absence (major isoform, starting with M163) of the N-terminal 162 amino acids.

The nature of the isolated *Δacl4*-suppressing *not1* alleles strongly suggested that the N-terminal 162 residues are a nonessential feature of Not1. Indeed, and in agreement with previous studies showing that Not1 variants with N-terminal deletions up to residue 394 or 753 support good growth at 30°C (*Maillet et al., 2000*; *Basquin et al., 2012*), the Not1 variant starting with M163 (163C construct; i.e., from residue 163 to the C-terminus) complemented the *Δnot1* null mutant to the wild-type extent from 16 to 30°C (*Figure 2C*); however, a ts phenotype could be observed at 37°C. Progressive mapping revealed that deletion of the first 28 residues (29C construct) still entailed poor growth at 37°C, whereas the deletion variant only lacking the N-terminal 20 residues (21C construct) permitted wild-type growth at all temperatures.

To demonstrate that absence of the N-terminal 162 amino acids enables, as predicted by the above-described findings, suppression of the *Δacl4* growth defect, we generated and transformed a *NOT1/ACL4* double shuffle strain with plasmids expressing wild-type Not1 or the three N-terminal deletion variants. Then, upon plasmid shuffling on 5-FOA-containing plates, the suppression capacity of the different constructs was determined by assessing growth on YPD plates (*Figure 2D*). As expected, no suppression of the *Δacl4* growth defect could be observed in *Δacl4/Δnot1* cells expressing either full-length Not1 or the fully functional Not1.21C variant. Conversely, expression of the Not1.29C or Not1.163C deletion variants restored wild-type growth of *Δacl4/Δnot1* cells at 16, 23, and 30°C but, in line with the above complementation assays (*Figure 2C*), not at 37°C. In accord with the efficient suppression of the *Δacl4* growth defect, the *RPL4* mRNA was, compared to its relative levels in the wild-type control, upregulated around twofold in *Δnot1* cells expressing the Not1.163C variant (*Figure 2E*). Likewise, as already observed before in cells lacking Caf130, Egd2, or both Btt1 and Egd1 (*Figure 1G*), the levels of the *RPL3* mRNA were also increased to a similar extent. However, none of the other tested RPG mRNAs (*RPL5*, *RPL10*, *RPL11*, *RPS2*, *RPS3*, and *RPS6*) exhibited similar changes in abundance as the *RPL3* and *RPL4* mRNAs. Taken together, these data show that the N-terminal 162 residues, which are specifically included in the minor, full-length Not1 isoform, are required both for growth at elevated temperature and for mediating the regulation of *RPL3* and *RPL4* mRNA levels.

Given that the Not1 scaffold of the Ccr4-Not complex is implicated in enabling negative regulation of the *RPL3* and *RPL4* mRNAs, an involvement of other Ccr4-Not components, especially those with established functions in mRNA degradation (the Caf1-Ccr4 deadenylase module) and coupling of translational repression with mRNA turnover (Not4 E3 ligase and Not2-Not5 module) (*Preissler et al., 2015*; *Alhusaini and Coller, 2016*; *Collart, 2016*; *Buschauer et al., 2020*), is highly likely. To test this by assessing suppression of the *Δacl4* growth defect, we first individually deleted the genes encoding these Ccr4-Not components in the W303 background. While the *Δcaf40* and *Δnot3* null mutants did not display any observable growth defects, the *Δcaf1*, *Δccr4*, and *Δnot4* null mutants exhibited pronounced sg phenotypes that were exacerbated at 37°C (*Figure 2—figure supplement 1C–G*). However, and in agreement with Not2 being required for the integrity of the Ccr4-Not complex (*Russell et al., 2002*), *Δnot2* and *Δnot5* mutant cells grew extremely slowly and were therefore excluded from the *Δacl4* suppression analysis (*Figure 2—figure supplement 1B*). The obtained null mutants were crossed with *ACL4* shuffle strains, tetrads were dissected, and then, after counter-selection against the *ACL4*-bearing *URA3* plasmid, the growth of cells derived from tetratype tetrads was assessed on YPD plates. However, absence of none of these Ccr4-Not components suppressed the growth defect of *Δacl4* cells at any of the tested temperatures, but, conversely, their absence synergistically enhanced, albeit to different extents, the sg phenotype of cells lacking Acl4 (*Figure 2—figure supplement 1C–G*). Based on this genetic analysis, we conclude that at least Caf40 and Not3 appear not to be required for the negative regulation of *RPL4* mRNA levels. At this stage, the plausible involvement of Not4 and especially the Caf1-Ccr4 deadenylase module cannot be discarded as any specific regulatory effect might be masked by more general effects of their absence on cytoplasmic mRNA decay and/or maintenance of proteostasis (*Panasenko and Collart, 2012*; *Halter et al., 2014*; *Preissler et al., 2015*; *Collart, 2016*). Recently, the N-terminal domain of Not5 (Not5-NTD) has been shown to mediate, via its binding to the ribosomal E-site, association of the Ccr4-Not complex

with translating 80S ribosomes lacking an accommodated tRNA in the A-site, thereby sensing and subjecting mRNAs with low codon optimality to degradation (*Buschauer et al., 2020*). Expression of a Not5 variant lacking the NTD (114C construct) in *Δnot5* cells did not result in any observable growth defect (*Figure 2—figure supplement 1H*), but was nonetheless neither suppressing the sg phenotype entailed by the absence of Acl4 nor leading to an increase in *RPL3* or *RPL4* mRNA levels (*Figure 2—figure supplement 1I and J*), suggesting that this mechanism is not part of the regulatory network controlling the abundance of the *RPL3* and *RPL4* mRNAs. In support of this, absence of Rps25 (eS25; *Δrps25a/Δrps25b*), which is a key determinant for Not5-NTD binding (*Buschauer et al., 2020*), neither suppressed the *Δacl4* growth defect (*Figure 2—figure supplement 1K*). Finally, no *Δacl4* suppression could be observed in the individual absence of the decapping activators Dhh1 and Pat1, which were shown to associate with Ccr4-Not via Not1 and Not3/5, respectively (*Maillet and Collart, 2002*; *Chen et al., 2014*; *Mathys et al., 2014*; *Alhusaini and Coller, 2016*), the major 5' -> 3' exonuclease Xrn1, or the exosome-assisting RNA helicase Ski2, which is required for cytoplasmic 3' -> 5' mRNA decay (*Parker, 2012*; *Figure 2—figure supplement 2A–D*).

## Caf130 connects Cal4 and Btt1 to Ccr4-Not by exclusively interacting with the full-length translational isoform of Not1

The common involvement in *RPL4* mRNA regulation, as well as their mutual interactions in large-scale yeast 2-hybrid (Y2H) and/or affinity purification approaches and the finding that Btt1 is associated with the Ccr4-Not complex in a Caf130-dependent manner (*Ito et al., 2001*; *Krogan et al., 2006*; *Cui et al., 2008*; *Yu et al., 2008*), indicated that Caf130, Cal4, and Btt1 may directly physically interact and be recruited through Caf130, likely via the N-terminal 162 amino acids of Not1, to the Ccr4-Not complex. To obtain evidence for this scenario, we first assessed the in vivo interactions of these components by GFP-Trap co-immunoprecipitation experiments. To this end, we constructed strains expressing distinct combinations of C-terminally GFP-tagged bait proteins and C-terminally TAP-tagged prey proteins. Upon rapid one-step affinity purification of the bait proteins from cell lysates using magnetic GFP-Trap beads, the prey proteins were detected by Western blot analysis with antibodies directed against the protein A moiety of the TAP tag, thus enabling the highly sensitive detection of co-precipitated prey proteins. In parallel, as specificity controls to evaluate the background binding of the prey proteins to the GFP-Trap beads, strains expressing nontagged bait proteins together with the individual TAP-tagged prey proteins were simultaneously analyzed. As expected, affinity purification of the full-length Not1-GFP bait resulted in the co-purification of Caf130 and Btt1; however, the major NAC β-subunit Egd1 and the NAC α-subunit Egd2 could not be detected above background levels (*Figure 3A*). Importantly, Cal4 was also enriched in the Not1-GFP immunoprecipitation, hence clearly establishing Cal4 as a novel accessory component of the Ccr4-Not complex. In agreement with the functional involvement of the very N-terminal part of Not1 in the negative regulation of *RPL3* and *RPL4* mRNA levels, the Not1.154C-GFP bait lacking the first 153 amino acids no longer co-purified Caf130 and Cal4, while it was still able to associate with core components of the Ccr4-Not complex, such as Not2, Not4, and Not5 (*Figure 3B*). Reciprocal experiments revealed specific interactions between (i) the Caf130-GFP bait and Btt1, Not1, and Cal4 (*Figure 3C*), (ii) the Cal4-GFP bait and Btt1, Not1, and Caf130 (*Figure 3D*), and (iii) the Btt1-GFP bait and Not1, Caf130, and Cal4 (*Figure 3E*). However, neither Egd1 nor Egd2 were detected above background in the Caf130-GFP and Cal4-GFP affinity purifications (*Figure 3C and D*), while the Btt1-GFP bait, as expected, co-purified Egd2 but not Egd1 (*Figure 3E*). Correspondingly, only the NAC β-subunit Egd1, but neither Caf130, Cal4, nor full-length Not1, was specifically co-precipitated with the Egd2-GFP bait (*Figure 3F*). Notably, a selective enrichment of the upper Not1-TAP band could be clearly discerned in the Caf130-GFP, Cal4-GFP, and Btt1-GFP affinity purifications (*Figure 3C–E*), thus further strengthening the notion that Caf130, Cal4, and Btt1 are specifically associated with the full-length Not1 isoform.

Next, we employed Y2H assays to untangle the interaction network between Not1, Caf130, Cal4, and Btt1. By testing the diverse distinct combinations of full-length proteins, we could reveal that Caf130 has the capacity to interact with Not1, Cal4, and Btt1 (*Figure 3G*); however, no Y2H interactions could be observed between the Cal4 bait and the Not1 or Btt1 preys and between the Btt1 bait and the Not1 or Cal4 preys (*Figure 3—figure supplement 1A and B*), strongly suggesting that Caf130 fulfills the role of a hub protein connecting, via its association with Not1, both Cal4 and Btt1 to the Ccr4-Not complex. In support of this, the Not1 bait exhibited some Y2H reporter activation,

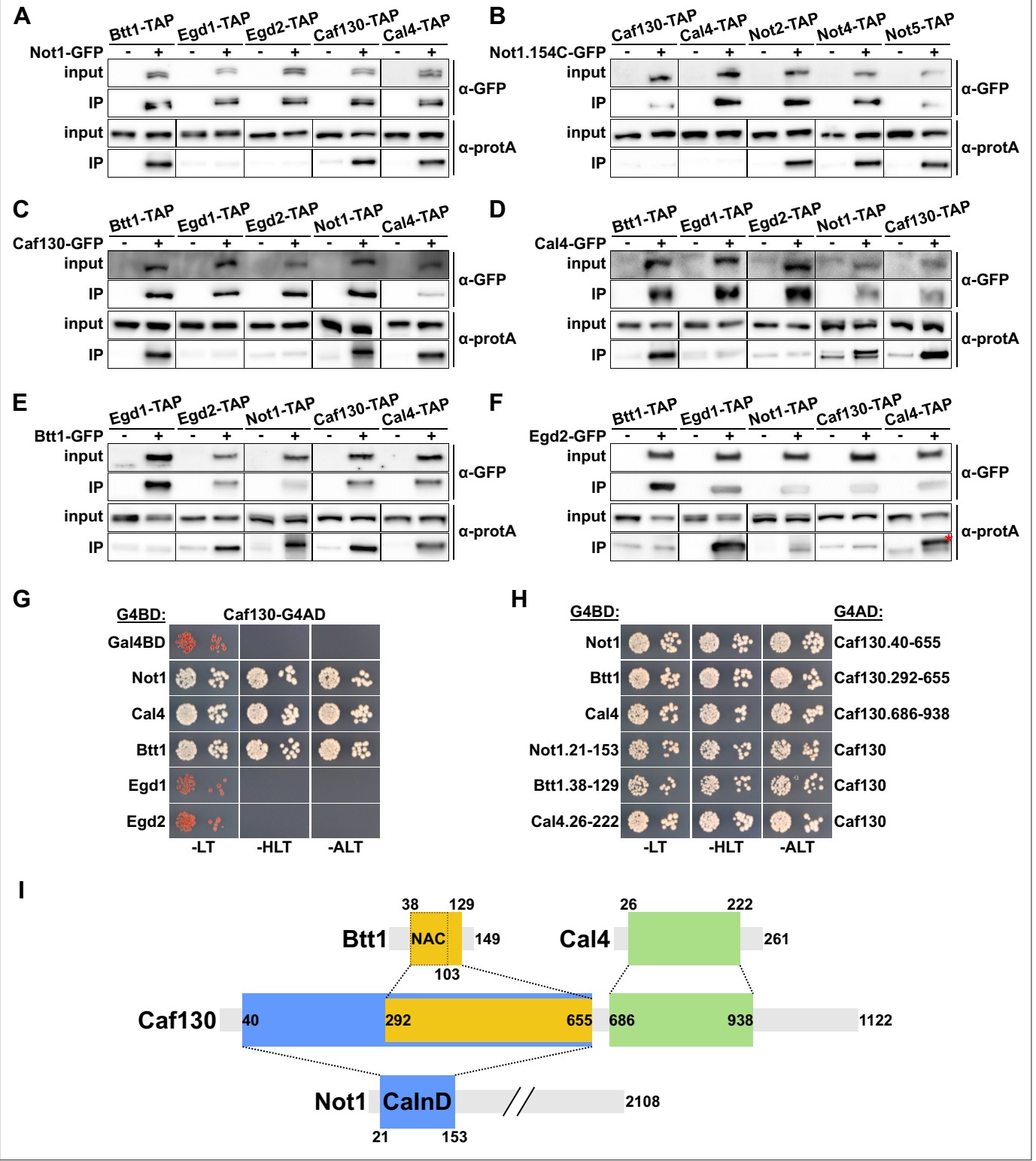

**Figure 3.** Caf130 connects Cal4 and Btt1 to Ccr4-Not by exclusively interacting with the full-length translational isoform of Not1. (**A–F**) Assessment of in vivo interactions by GFP-Trap co-immunoprecipitation. Cells expressing nontagged (-) or C-terminally GFP-tagged (+) versions of Not1 (**A**), Not1.154C (**B**), Caf130 (**C**), Cal4 (**D**), Btt1 (**E**), and Egd2 (**F**) together with the indicated C-terminally TAP-tagged prey proteins were grown in YPD medium at 30°C. All fusion proteins were expressed from their genomic locus, except GFP-tagged Not1 and Not1.154C as well as their nontagged counterparts, which were expressed from plasmid under the control of the *NOT1* promoter in Δ*not1* cells. Cell lysates (input; 1/1000 of IP input) and GFP-Trap affinity purifications (IP; 1/5 of complete IP) were analyzed by Western blotting using anti-GFP and anti-protA antibodies. Since Not1, Egd1, and Egd2 are

*Figure 3 continued on next page*

*Figure 3 continued*

expressed at higher levels, the inputs for detection of Not1-TAP were diluted twofold and those of Egd1-TAP and Egd2-TAP 20-fold to keep all Western signals in a similar range. Note that the band marked with a red asterisk corresponds to the Egd2-GFP bait protein, which is, due to its abundance in the IP, nonspecifically recognized by the anti-protA antibody. (**G, H**) Assessment of protein–protein interactions by yeast 2-hybrid (Y2H). (**G**) Caf130 interacts with Not1, Cal4, and Btt1. Plasmids expressing full-length Not1, Cal4, Btt1, Egd1, or Egd2, fused to the C-terminal Gal4 DNA-binding domain (G4BD), and full-length Caf130, fused to the C-terminal Gal4 activation domain (G4AD), were co-transformed into the Y2H reporter strain PJ69-4A. Cells were spotted in 10-fold serial dilution steps onto SC-Leu-Trp (-LT), SC-His-Leu-Trp (-HLT), and SC-Ade-Leu-Trp (-ALT) plates, which were incubated for 3 days at 30°C. (**H**) Minimal interaction surfaces mediating the binary Caf130-Not1, Caf130-Btt1, and Caf130-Cal4 association. Plasmids expressing the indicated C-terminally G4BD-tagged Not1, Cal4, or Btt1 and G4AD-tagged Caf130 full-length proteins or respective minimal interaction fragments thereof were co-transformed into the Y2H reporter strain PJ69-4A. (**I**) Schematic representation of the binary interactions and the determined minimal interaction surfaces. The respective minimal interaction surfaces, as determined by Y2H mapping, are highlighted by colored rectangles. The borders of the NAC domain, as defined in **Liu et al., 2010**, are also indicated. The Caf130-interacting domain of Not1 is abbreviated as CaInD.

The online version of this article includes the following source data and figure supplement(s) for figure 3:

**Source data 1.** Original image files of the Western blots shown in **Figure 3A and B**, including a PDF file showing the full blots and indicating the cropped areas.

**Source data 2.** Original image files of the Western blots shown in **Figure 3C**, including a PDF file showing the full blots and indicating the cropped areas.

**Source data 3.** Original image files of the Western blots shown in **Figure 3D**, including a PDF file showing the full blots and indicating the cropped areas.

**Source data 4.** Original image files of the Western blots shown in **Figure 3E**, including a PDF file showing the full blots and indicating the cropped areas.

**Source data 5.** Original image files of the Western blots shown in **Figure 3F**, including a PDF file showing the full blots and indicating the cropped areas.

**Figure supplement 1.** Caf130 interacts with Not1, Cal4, and Btt1.

**Figure supplement 2.** Mapping of the minimal interaction surfaces on Caf130, Not1, Cal4, and Btt1.

**Figure supplement 3.** Mapping of the Egd2-binding surface on Btt1.

albeit to a much lesser extent than in combination with the Caf130 prey, when combined with the Cal4 and Btt1 preys (**Figure 3—figure supplement 1C**), which, in light of the above findings, can readily be explained by Caf130 serving as a bridging molecule for these interactions. Moreover, as already indicated by the co-immunoprecipitation experiments, we did not detect any interactions between Egd1 or, respectively, Egd2 and Not1, Caf130, or Cal4 at 30°C (**Figure 3G**, **Figure 3—figure supplement 1D and E**). Interestingly, however, a Y2H interaction between Caf130 and both Egd1 and Egd2 could be observed at 16°C (**Figure 3—figure supplement 1F**), suggesting that the common NAC domain has an intrinsic capacity to interact with Caf130 and thus offering a potential explanation for the partially redundant contribution of Egd1 and Btt1 to the regulatory process. Subsequent Y2H mapping of the respective minimal interaction surfaces (**Figure 3H**, **Figure 3—figure supplement 2A–C**), as schematically summarized in **Figure 3I**, first revealed that Caf130 associates via (i) a large N-terminal portion (amino acids 40–655) with Not1, (ii) the C-terminal part thereof (amino acids 292–655) with Btt1, and (iii) a consecutive segment (amino acids 686–938) with Cal4. In agreement with the genetic and the GFP-Trap co-immunoprecipitation data (**Figure 2C and D** and **Figure 3A and B**), the minimal Not1 surface mediating Caf130 binding could be mapped to an N-terminal segment encompassing amino acids 21–153 (**Figure 3H**, **Figure 3—figure supplement 2A**), which we therefore termed the Caf130-interacting domain (CaInD). On Btt1, the minimal fragment for Caf130 binding comprised amino acids 38–129, corresponding to the NAC domain (amino acids 38–103) bearing a short C-terminal extension (**Figure 3H**, **Figure 3—figure supplement 2B**). In line with the reported NAC domain crystal structures of the human NACA-BTF3 heterodimer (**Liu et al., 2010**; **Wang et al., 2010**), both Btt1 and Egd1 interacted with Egd2's NAC domain, and the region covering the six predicted β-strands (amino acids 54–101) of Btt1's NAC domain was sufficient to mediate the association with Egd2 (**Figure 3—figure supplement 3A and B**). The finding that both Caf130 and Egd2 bind to the NAC domain of Btt1 corroborates, as already indicated by the lack of a detectable co-precipitation between these two (**Figure 3C and F**), a model in which Btt1 associates in a mutually exclusive manner with either Caf130 or Egd2. Finally, the minimal Caf130-binding surface on Cal4 was formed by amino acids 26–222 (**Figure 3H**, **Figure 3—figure supplement 2C**).

Taken together, the co-immunoprecipitation and Y2H interaction analyses establish Cal4 as a novel accessory component of the Ccr4-Not complex and reveal that Caf130, in its role as a scaffold protein, has the capacity to simultaneously interact with Not1, Btt1, and Cal4. Importantly, the deciphered physical interaction network correlates very well with the common function of these four proteins in regulating Rpl4 expression levels.

## The regulation-conferring signal is located within Rpl4 and overlaps with the Acl4-binding site

Interestingly, high-throughput sequencing indicated that one of the isolated *Δacl4* suppressors carried a mutation within *RPL4A*, hereafter referred to as the *rpl4a.W109C* allele, changing tryptophan 109 (TGG codon) to cysteine (TGT codon) in the 362-residue-long Rpl4a. The W109 residue is located at the C-terminal end of the long internal loop (amino acids 44–113), which mediates Acl4 binding (*Pillet et al., 2015*; *Stelter et al., 2015*; *Huber and Hoelz, 2017*). As revealed by the X-ray co-structure of *Chaetomium thermophilum* Acl4 and Rpl4 (*Huber and Hoelz, 2017*), the long internal loop under-goes large conformational changes upon Acl4 binding, including the reorientation of the W109 side chain from its loop-inward position in the ribosome-bound state to an outward configuration in which it is shielded by Acl4 (*Figure 4—figure supplement 1A and B*). Y2H assays showed that the Rpl4a.W109C protein still interacts, albeit to a lesser extent than wild-type Rpl4a, with Acl4 (*Figure 4—figure supplement 1C*), indicating that the W109 side chain is, however, not strictly required for this interaction. To confirm that the W109C substitution indeed suppresses the *Δacl4* growth defect, we first integrated the *rpl4a.W109C* allele, as well as the wild-type *RPL4A* control, at the genomic locus. This was necessary since expression of Rpl4a from a monocopy plasmid already efficiently restores the sg phenotype of *Δacl4* cells (*Pillet et al., 2015*). To evaluate the impact of the *rpl4a.W109C* mutation on yeast growth, the *rpl4a.W109C* allele was combined with the *Δrpl4b* null allele. As shown in *Figure 4A*, the strain exclusively expressing the Rpl4a.W109C protein grew almost as well as the *RPL4A/Δrpl4b* control strain (*Figure 4A*). Next, we combined the *rpl4a.W109C* allele with the *Δacl4* null allele, revealing a robust suppression of the *Δacl4* growth defect at all tested temperatures (*Figure 4B*). To assess whether mutation of further residues in proximity of W109 could also confer suppression, we tested the four previously described nonoverlapping, consecutive alanine substitution mutants (named BI, BII, BIII, and BIV; see *Figure 4C*), which no longer interact with Acl4 (*Pillet et al., 2015*). Again, these mutant alleles were integrated at the genomic *RPL4A* locus and their complementation and suppression capacity was determined by combining them with the *Δrpl4b* or *Δacl4* null mutation (*Figure 4A and B*). The BI mutations (F90A/N92A/M93A/C94A/R95A) conferred a strong sg phenotype and, accordingly, did not enable suppression of the *Δacl4* phenotype. Growth of cells expressing the variant harboring the BII mutations (R98A/M99A/F100A) was not substantially ameliorated, and almost no *Δacl4* suppression could be observed from 16 to 30°C; however, some growth improvement and weak suppression were apparent at 37°C. Conversely, the BIII mutations (P102A/T103A/K104A/T105A) permitted significantly better growth, especially at 16°C, at all tested temperatures except 37°C, and suppression, up to the BIII-inherent growth defect, could also be observed and was again particularly pronounced at 16°C. Similar to the W109C substitution, the BIV mutations (W106A/R107A/K108A/W109A), comprising an exchange of tryptophan 109 to alanine, only elicited a very mild growth defect and conferred robust suppression of the *Δacl4* growth defect at all tested temperatures (*Figure 4A and B*). Thus, the genetic analyses establish the W109 residue within the long internal loop region as a critical determinant for enabling negative regulation of Rpl4 expression.

Next, we assessed whether the observed suppression of the *Δacl4* growth defect by the *rpl4a.W109C* and BIV mutations coincided with a stabilization of their mRNAs. To this end, the wild-type and mutant-encoding *RPL4A* ORFs were fused at their 3' end to the yEGFP coding sequence and were expressed from a monocopy plasmid under the transcriptional control of the *ADH1* promoter in wild-type and *Δcaf130* cells. Then, the relative levels of the *RPL4A*-yEGFP fusion mRNAs were determined by qRT-PCR using a primer pair specifically amplifying a portion of the yEGFP coding sequence and compared between the wild-type and *Δcaf130* situation where regulation is either in place or disabled and *RPL4A* mRNA levels are therefore expected to be either minimal or maximal, respectively. Importantly, downregulation of the fusion mRNA containing wild-type *RPL4A*, when transcribed from the *ADH1* promoter, could be clearly observed in wild-type cells (*Figure 4D*), indicating that the altered

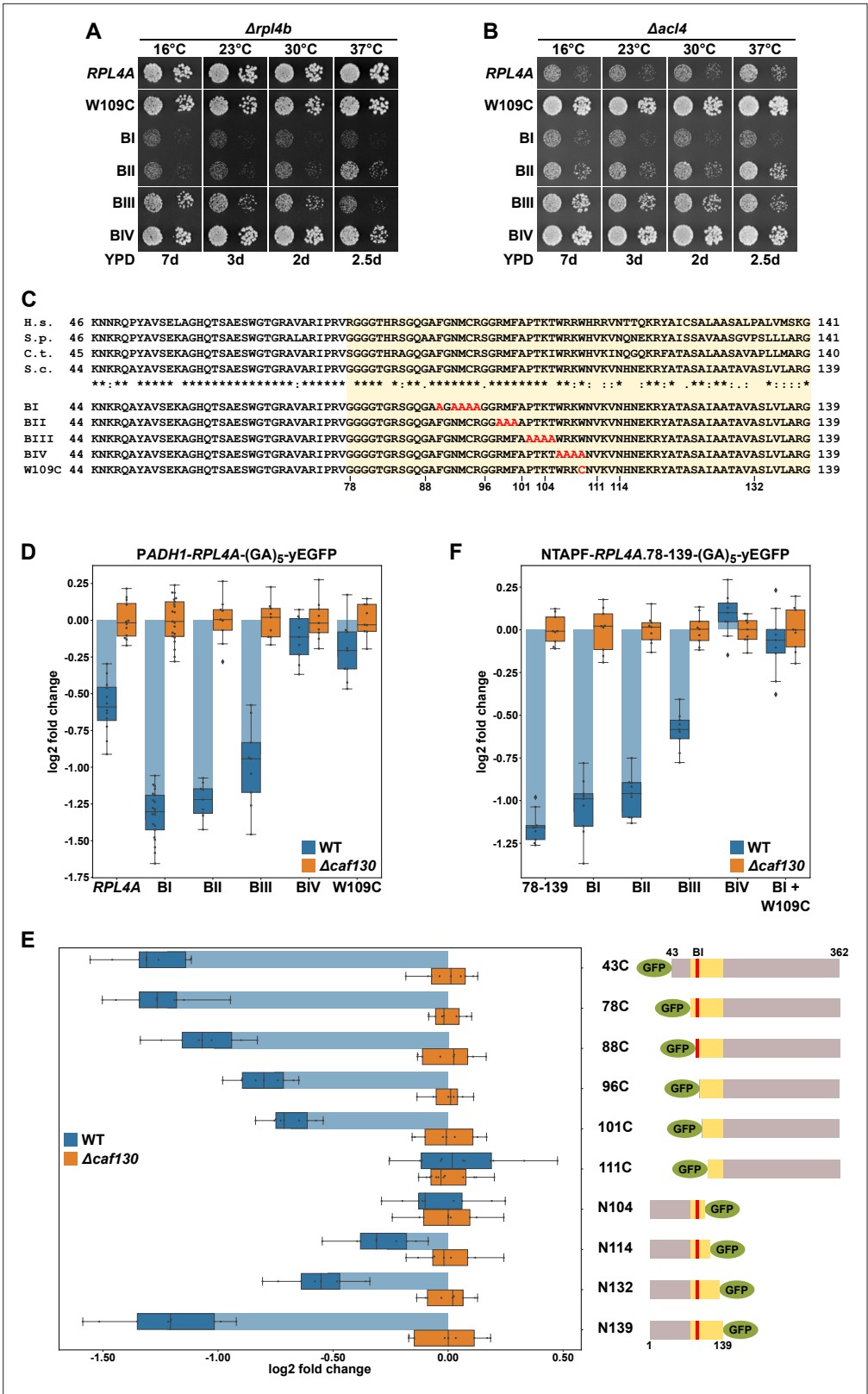

**Figure 4.** The Rpl4 protein harbors the regulation-conferring signal. (**A, B**) Suppression of the *Δacl4* growth defect by the *rpl4a.W109C* allele. Cells harboring wild-type (WT) *RPL4A* or the indicated *rpl4a* alleles, expressed from the genomic locus, in addition to either the deletion of *RPL4B* (*Δrpl4b*) (**A**) or *ACL4* (*Δacl4*) (**B**) were spotted in 10-fold serial dilution steps onto YPD plates. (**C**) Amino acid sequences of the long internal loop (amino acids

*Figure 4 continued on next page*

*Figure 4 continued*

44–113), extended to the C-terminal border of the minimal segment conferring full *RPL4A* mRNA regulation (amino acids 78–139; highlighted by a light yellow background color), of Rpl4 from different eukaryotic species (H.s., *Homo sapiens*; S.p., *Schizosaccharomyces pombe*; C.t., *Chaetomium thermophilum*; S.c., *Saccharomyces cerevisiae*). Conserved (*), strongly similar (:), and weakly similar (.) amino acids are indicated below the alignment. The nonoverlapping, consecutive alanine substitutions within this Rpl4a segment are depicted in the lower part: block-I mutant (BI): F90A/N92A/M93A/C94A/R95A; block-II mutant (BII): R98A/M99A/F100A; block-III mutant (BIII): P102A/T103A/K104A/T105A; and block-IV mutant (BIV): W106A/R107A/K108A/W109A. The W109C exchange is also indicated. (**D**) Negative regulation of *RPL4A* mRNA levels is strongly diminished by the *rpl4a.W109C* mutation. Levels of *RPL4A*-yEGFP fusion mRNAs were determined in WT (blue bars) or Δ*caf130* (orange bars) cells by qRT-PCR with a primer pair specifically amplifying a part of the yEGFP coding sequence fused to the 3′-end of the *RPL4A* ORF. Cells harboring *RPL4A* or the indicated *rpl4a* alleles, expressed from the *ADH1* promoter, on plasmid were grown at 30°C in SC-Leu medium. The data shown were obtained from at least three different WT and Δ*caf130* strains (biological replicates), in each case consisting of a technical triplicate. Changes in mRNA levels of each assayed *RPL4A* allele between WT (negative regulation on) and Δ*caf130* (negative regulation off) cells have been normalized to their maximal abundance in Δ*caf130* cells. The data are represented as described in the legend to *Figure 1G*. (**E**) Mapping of the minimal regulation-conferring region on *RPL4A*. Levels of fusion mRNAs containing different regions of the *RPL4A* coding sequence were determined in WT (blue bars) or Δ*caf130* (orange bars) cells by qRT-PCR. Cells expressing the indicated N-terminal deletion variants, fused to an N-terminal yEGFP tag, or C-terminal deletion variants, fused to a C-terminal yEGFP tag, from plasmid under the transcriptional control of the *ADH1* promoter were grown at 30°C in SC-Leu medium. To avoid any effect on mRNA levels of co-translational Acl4 binding to the nascent Rpl4a polypeptides, the BI mutations were introduced into those constructs comprising this region of the *RPL4A* coding sequence. The yEGFP-fused Rpl4a variants, encoded by the assayed constructs, are schematically represented. The Rpl4a segment encoded by the minimal regulation-conferring *RPL4A* region is highlighted in yellow and the position of the BI alanine substitutions by a red bar. The data shown were obtained from at least three different WT and Δ*caf130* strains (biological replicates), in each case consisting of a technical triplicate. (**F**) The *rpl4a.W109C* mutation within the minimal regulation-conferring region strongly diminishes negative regulation of *RPL4A* mRNA levels. Levels of fusion mRNAs were determined in WT (blue bars) or Δ*caf130* (orange bars) by qRT-PCR. Cells expressing the Rpl4a(78-139) fragment harboring the wild-type sequence or the indicated mutations, fused to an N-terminal TAP-Flag (NTAPF) and a C-terminal yEGFP tag, from plasmid under the transcriptional control of the *ADH1* promoter were grown at 30°C in SC-Leu medium. The data shown were obtained from three different WT and Δ*caf130* strains (biological triplicates), in each case consisting of a technical triplicate.

The online version of this article includes the following figure supplement(s) for figure 4:

**Figure supplement 1.** Residue W109 of Rpl4 is facing the inner surface of Acl4 and the W109C exchange reduces the interaction of Rpl4 with Acl4.

**Figure supplement 2.** Ongoing translation is required for efficient negative regulation.

promoter context and the addition of the yEGFP coding sequence do not fundamentally change the regulation-conferring process. Notably, the levels of the *RPL4A*-yEGFP fusion mRNA coding for the BI mutant protein were even more substantially decreased, strongly suggesting that co-translational capturing of nascent Rpl4a by Acl4 stabilizes the *RPL4A* mRNA. The levels of the mRNAs encoding the BII and BIII mutant variants were still lower than the one of the mRNA harboring wild-type *RPL4A*, but, compared to the BI-expressing mRNA, a slight gradual increase in their abundance could be noticed. Most importantly, and in line with the robust suppression of the Δ*acl4* growth defect, presence of either the BIV mutations or the W109C substitution restored the levels of their mRNAs in wild-type cells almost to the ones detected in Δ*caf130* cells.

Aiming to corroborate the importance of the above-identified residues for the regulatory process and to delineate, if possible, a minimal regulation-conferring region, we first constructed plasmids expressing progressively N- and C-terminally deleted Rpl4a variants, fused to an N- or C-terminal yEGFP moiety, respectively (as depicted in *Figure 4E*), under the transcriptional control of the *ADH1* promoter. To avoid any mRNA-stabilizing effect due to co-translational Acl4 binding, the BI mutations were introduced into all constructs comprising this region of the *RPL4A* coding sequence. Then, the plasmid constructs were transformed into wild-type and Δ*caf130* cells and the relative levels of the different fusion mRNAs were determined by qRT-PCR using, as above, a primer pair specifically amplifying a portion of the common yEGFP coding sequence. The levels of the yEGFP-*RPL4A* fusion mRNAs coding for Rpl4a deletion variants lacking the first 42 (denoted as 43C construct) or 77 amino

acids were, similarly to the *RPL4A*-yEGFP mRNA encoding full-length Rpl4a containing the BI mutations (*Figure 4D*), about 2.5-fold lower in wild-type compared to *Δcaf130* cells (*Figure 4E*). Further progressive mapping revealed a gradual increase in mRNA abundance when the encoded proteins were either devoid of the first 87, 95, or 100 amino acids; remarkably, the Rpl4a variant starting with amino acid 101 (101C construct) still conferred a significant, around 1.5-fold negative regulation to its mRNA. However, the fusion mRNA expressing the deletion variant lacking the first 110 amino acids was no longer subjected to regulation in wild-type cells; thus, clearly highlighting the importance of the short segment encompassing amino acids 101–110, which notably comprises the W109 residue. Accordingly, no regulation was imposed on its encoding fusion mRNA by the C-terminal deletion variant ending at amino acid 104 (N104 construct). Progressive extension of the C-terminal end of the encoded variants revealed that some mRNA regulation started to occur when Rpl4a ended at amino acid 114 and that, after a further subtle decrease in mRNA levels entailed by the Rpl4a.N132 protein, efficient regulation was reached again when the encoded Rpl4a was extended up to amino acid 139 (*Figure 4E*). Next, we addressed whether the inferred minimal regulation-conferring region (amino acids 78–139) was sufficient to enforce, when placed in a heterologous context, a decrease in mRNA levels in wild-type cells. To this end, we generated a plasmid-based construct expressing Rpl4a residues 78–139 from the *ADH1* promoter as a fusion protein that is flanked by an N-terminal TAP-Flag tag (NTAPF) and, for the determination of the mRNA levels by qRT-PCR, a C-terminal yEGFP moiety. Moreover, the BI, BII, BIII, and BIV mutations, as well as a combination of the W109C substitution with the BI mutations, were introduced into the coding sequence in order to assess whether these alterations had the same effect within the minimal region as in the context of full-length *RPL4A* (*Figure 4D*). Importantly, the presence of either the BIV mutations or the W109C substitution resulted, when compared to the similarly regulated wild-type construct or the one containing only the BI mutations, in an increase in the abundance of the respective mRNAs up to their levels in *Δcaf130* cells (*Figure 4F*). Moreover, a slight increase in mRNA levels could be observed in the presence of the BII and, more evidently, of the BIII mutations, pointing once more to a minor contribution of the residues that are altered by the BIII mutations toward the negative regulation of its encoding mRNA.

Taken together, mapping of the regulation-conferring region on Rpl4a revealed that a segment encompassing amino acids 78–139 is sufficient to have a negative impact on the abundance of the encoding mRNA. Within this region, the tryptophan 109 residue, whose mutation to cysteine enables robust suppression of the *Δacl4* growth defect, appears to be a critical determinant for mediating the negative regulation of *RPL4* mRNA levels. Notably, the W109 residue, which is located near the C-terminal end of the Acl4-binding site, is shielded by Acl4, and, moreover, mutations that abolish the interaction with Acl4 promote a further reduction of *RPL4* mRNA levels. It is therefore highly likely that co-translational capturing of Rpl4 by Acl4 stabilizes the *RPL4* mRNA (see below), possibly by precluding the recognition of the nascent Rpl4 segment around the W109 residue by the regulatory machinery. In line with ongoing translation being required for efficient negative regulation, addition of the translation elongation inhibitor cycloheximide abrogated the difference in abundance between the above reporter mRNAs containing the BI or BIV mutations within the minimal regulation-conferring coding sequence (*Figure 4—figure supplement 2*).

## The regulation-conferring Rpl3 segment is adjacent to the Rrb1-binding site

Given that the same machinery, with the exception of Cal4, is involved in the negative regulation of *RPL3* mRNA levels and considering that Rpl3 is also co-translationally captured by a dedicated chaperone, the essential Rrb1 (*Pausch et al., 2015*), we next wished to explore whether the underlying principles of both regulation events might be similar. In particular, we suspected that the regulation-conferring region might overlap with or be in the immediate proximity of the Rrb1-binding site, which we had previously mapped to the N-terminal 15 residues of Rpl3 (*Pausch et al., 2015*). To map the regulation-conferring region, we constructed monocopy plasmids expressing wild-type Rpl3 as well as N- and/or C-terminal truncation variants thereof, fused to a C-terminal yEGFP moiety, under the transcriptional control of the *ADH1* promoter. Then, upon transformation into wild-type and *Δcaf130* cells, with the latter providing a benchmark for the maximal abundance of each transcript, total RNA was extracted from exponentially growing cells and the relative levels of the *RPL3*-yEGFP fusion mRNAs were determined by qRT-PCR. In this experimental setup, the levels of the fusion mRNA harboring

full-length *RPL3* were only downregulated by around 1.25-fold in wild-type cells (*Figure 5A*); thus, negative regulation was less efficient than in the case of the endogenous *RPL3* mRNA (*Figure 1G*). Notably, however, the abundance of the mRNA encoding a deletion variant lacking the first seven residues (8C construct), which is no longer capable of interacting with Rrb1 (*Pausch et al., 2015*; *Figure 5—figure supplement 1*), was reduced more than threefold, indicating that absence of Rrb1 binding to nascent Rpl3 has an mRNA-destabilizing effect. Analyses of further N-terminal deletion variants revealed a minor increase in mRNA abundance when the encoded protein lacked the first 11 amino acids (12C construct) and a more prominent increase, resulting in around 2.2-fold lower mRNA levels, when the first 14 residues (15C construct) were missing (*Figure 5A*). Strikingly, removal of the first 17 amino acids (18C construct) from the encoded protein raised the abundance of the mRNA almost up to its levels in Δ*caf130* cells, suggesting an important contribution of a very short segment, comprising residues 12 to 17, to the negative regulation. Mapping of the C-terminal border revealed that the fusion mRNA expressing the first 52 amino acids (N52 construct) was considerably more regulated than the full-length *RPL3*-yEGFP mRNA. However, further refinement by testing even shorter, C-terminally truncated variants was not possible since their expression, presumably owing to the titration of Rrb1 (*Pausch et al., 2015*), conferred a strong sg phenotype to wild-type cells (*Figure 5— figure supplement 2B*). Therefore, we generated constructs expressing different C-terminally deleted Rpl3 variants that were simultaneously lacking the first seven amino acids. Compared to the fusion mRNA coding for the Rpl3.N52 variant, the abundance of the mRNA encoding the Rpl3 fragment spanning residues 8–52 (8–52 construct) was even further diminished, exhibiting a more than 2.5-fold reduction in wild-type cells compared to Δ*caf130* cells (*Figure 5A*), thus, confirming the notion that co-translational recognition of the N-terminal Rpl3 residues by Rrb1 positively affects mRNA levels. The extent of negative regulation was only marginally decreased when the encoded Rpl3 variant ended at amino acid 48 (8–48 construct), but a strong increase in mRNA levels could be observed when the expressed Rpl3 variant lacked an additional four C-terminal residues (8–44 construct). In conclusion, the above data show that the minimal regulation-conferring region required for robust negative regulation of the *RPL3* mRNA is contained within the N-terminal part of Rpl3, from residue 8 to 52, and attribute a potentially crucial involvement to a short segment between residues 11 and 18.

To assess the contribution of discrete residues within the minimal regulation-conferring region with maximum sensitivity, as achieved by introducing the BI mutations in the case of Rpl4, without removing any N-terminal residues but still disabling Rrb1 binding, we first had to identify the residues that are mandatory for mediating the interaction with Rrb1. Given that the first seven amino acids of Rpl3 are required for Rrb1 association (*Pausch et al., 2015*), we focused the mutational analysis on residues within this short segment (H3, R4, K5, and Y6) and additionally included the conserved R10 and H11 residues (*Figure 5B*). Gratifyingly, the H3E, K5E, and Y6E mutations, both in the context of full-length Rpl3 or when comprised in the C-terminally truncated Rpl3.N52 variant, abolished the Y2H interaction with Rrb1 (*Figure 5—figure supplement 1*). Moreover, these mutants were unable to complement the lethality of Δ*rpl3* cells (*Figure 5—figure supplement 2A*). On the other hand, the R10E/H11E and R10A/H11A substitutions did not affect the Y2H interaction with Rrb1, but nonetheless, presumably owing to an important role of these two residues in rRNA binding, pre-60S assembly, or functioning of the ribosome, they resulted in extremely weak growth of Δ*rpl3* cells (*Figure 5—figure supplements 1 and 2A*). Importantly, presence of the H3E mutation, which represents the most N-terminal exchange abolishing Rrb1 binding, led to a similar decrease in *RPL3*-yEGFP mRNA levels as removal of the first seven residues did (*Figure 5A*). Hence, we chose to introduce the H3E mutation into Rpl3.N52 for unveiling the contribution of selected residues, within the above-determined minimal region (amino acids 8–52), to the negative regulation of the encoding mRNA. To facilitate the task, we simultaneously changed 2–3 neighboring residues, especially focusing on bulky hydrophobic and positively charged amino acids, to alanine (*Figure 5B*). Before assessing the mRNA levels, we evaluated the generated Rpl3 variants, in the absence of the H3E mutation, with respect to their capability to associate with Rrb1 and sustain growth of Δ*rpl3* cells. In the context of the N-terminal 52 amino acids, none of the introduced mutations affected the Y2H interaction with Rrb1 (*Figure 5—figure supplement 1*). In the context of full-length Rpl3, however, the H13A/L14A and F46A/L47A mutations reduced or respectively abolished the interaction with Rrb1, while all other tested mutants associated with Rrb1 to a similar extent as wild-type Rpl3. Given that concurrent alanine substitution of F46 and L47, which are situated at the beginning of the first β-strand in the center of Rpl3's two-lobed globular domain

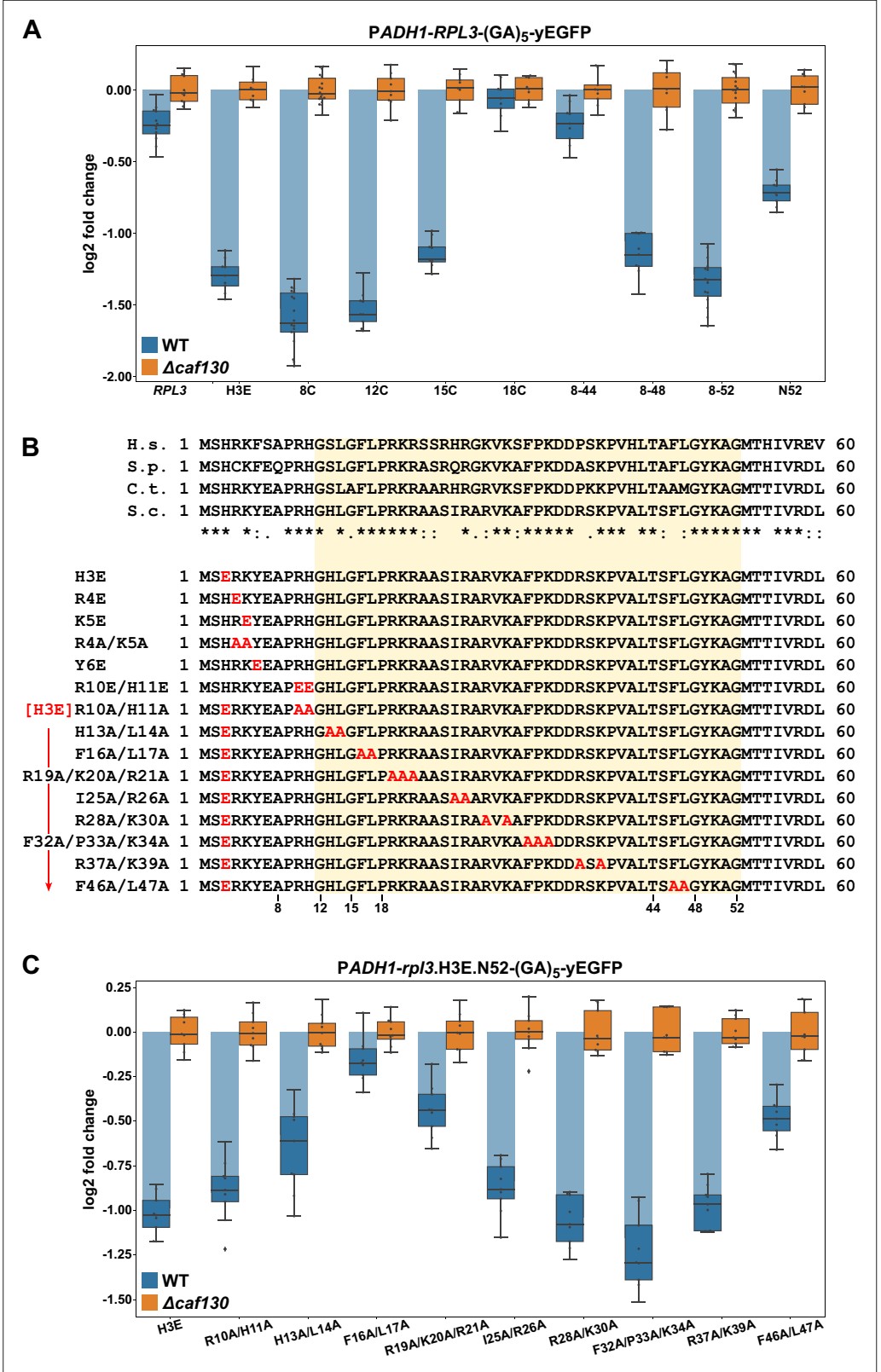

**Figure 5.** The regulation-conferring Rpl3 segment is adjacent to the Rrb1-binding site. (**A**) Mapping of the minimal regulation-conferring region on *RPL3*. Levels of fusion mRNAs containing different regions of the *RPL3* coding sequence were determined in wild-type (WT; blue bars) or Δ*caf130* (orange bars) cells by qRT-PCR with a primer pair specifically amplifying a part of the yEGFP coding sequence. Cells expressing full-length Rpl3 or the indicated

*Figure 5 continued on next page*

*Figure 5 continued*

substitution and deletion variants, fused to a C-terminal yEGFP tag, from plasmid under the control of the *ADH1* promoter were grown at 30°C in SC-Leu medium. The data shown were obtained from three different WT and *Δcaf130* strains (biological replicates; note that some strains were used more than once), in each case consisting of a technical triplicate. The data are represented as described in the legend to **Figure 4D**. (**B**) Amino acid sequences of the N-terminal region of Rpl3, containing the minimal Rrb1-interacting region (amino acids 1–15; **Pausch et al., 2015**) and extended to the C-terminal border of the minimal segment conferring full *RPL3* mRNA regulation (amino acids 12–52; highlighted by a light yellow background color), from different eukaryotic species (H.s., *H. sapiens*; S.p., *S. pombe*; C.t., *C. thermophilum*; S.c., *S. cerevisiae*). Conserved (*), strongly similar (:), and weakly similar (.) amino acids are indicated below the alignment. The glutamate and alanine substitutions, contained in the Rpl3 variants used in this study, within the N-terminal region of Rpl3 are depicted in the lower part. (**C**) Residues F16 and L17 are main determinants for efficient negative regulation of *RPL3* mRNA levels. Levels of fusion mRNAs were determined in WT (blue bars) or *Δcaf130* (orange bars) cells expressing the Rpl3.N52 fragment harboring the indicated mutations, fused to a C-terminal yEGFP tag, from plasmid under the transcriptional control of the *ADH1* promoter. To avoid any effect on mRNA levels of co-translational Rrb1 binding to the nascent Rpl3 polypeptides, the H3E mutation was introduced into all assayed constructs. The data shown were obtained from three different WT and *Δcaf130* strains (biological triplicates), in each case consisting of a technical triplicate.

The online version of this article includes the following figure supplement(s) for figure 5:

**Figure supplement 1.** Effect of mutations within Rpl3's N-terminal region on the Y2H interaction with Rrb1.

**Figure supplement 2.** The *rpl3.F16A/L17A* allele fully complements the absence of endogenous *RPL3* and suppresses the lethality of *Δrrb1* cells.

(**Figure 5—figure supplement 2C**), also abolished growth of *Δrpl3* cells (**Figure 5—figure supplement 2A**), the combination of these two mutations likely affects the productive folding of full-length Rpl3. Importantly, most of the other generated *rpl3* mutants did not display any apparent growth defect, only the H13A/L14A and the R28A/K30A mutations conferred an sg phenotype at all tested temperatures or moderately impaired growth at 37°C, respectively (**Figure 5—figure supplement 2A**). After having established their impact on Rrb1 binding and growth, we assessed the effect of the different alanine substitutions on the abundance of the fusion mRNAs encoding these C-terminally yEGFP-tagged Rpl3.N52 variants bearing the H3E exchange. Compared to the control containing only the H3E mutation, which reduced transcript levels by more than twofold in wild-type cells, the most prominent increase in mRNA abundance could be observed by the additional presence of the F16A/L17A substitutions (**Figure 5C**). Moreover, a clear diminution of negative regulation, resulting in a less than 1.5-fold downregulation of mRNA levels in wild-type cells, was brought about by the R19A/K20A/R21A and F46A/L47A mutations. Next, we wondered whether deregulated expression of Rpl3 was sufficient to restore growth of cells lacking the essential Rrb1. In contrast to the robust suppression of the *Δacl4* growth defect by endogenously expressed W109C- or BIV-mutant Rpl4a variants, expression of Rpl3.F16A/L17A from monocopy plasmid only enabled weak growth in the absence of Rrb1 (**Figure 5—figure supplement 2D**), suggesting that the essential role of Rrb1 extends beyond being a passive Rpl3 binder and likely includes other aspects, such as promoting the safe transfer and efficient assembly of Rpl3 into early pre-60S particles.

Taken together, we have mapped the regulation-conferring region to amino acids 8–52 of Rpl3 and identified residues therein, especially phenylalanine 16 and/or leucine 17, that serve as necessary determinants for efficient negative regulation of *RPL3* mRNA levels. These two residues are directly adjacent to the minimal Rrb1-binding site consisting of the N-terminal 15 amino acids (**Pausch et al., 2015**), and, noteworthily, the F16A/L17A double substitution enables normal growth and does not appear to affect the interaction with Rrb1. Importantly, the above-described data now permit to conclude that similar principles apply to the negative regulation of *RPL3* and *RPL4* mRNA levels. Besides basically involving the same regulatory machinery, maximal regulation requires in both cases an additional segment of around 30 amino acids after the identified, critically important Rpl3 (F16/L17) or Rpl4 (W109) residues, suggesting that an auxiliary, yet to be unveiled feature contributes to the regulation process (see Discussion). Moreover, the crucial role of individual residues provides compelling evidence that nascent Rpl3 and Rpl4 harbor the signal eliciting the negative regulation of their own mRNA levels. Finally, the immediate proximity or overlap of the Rrb1- or Acl4-binding site with residues that are needed for potent regulation advocates a model in which co-translational

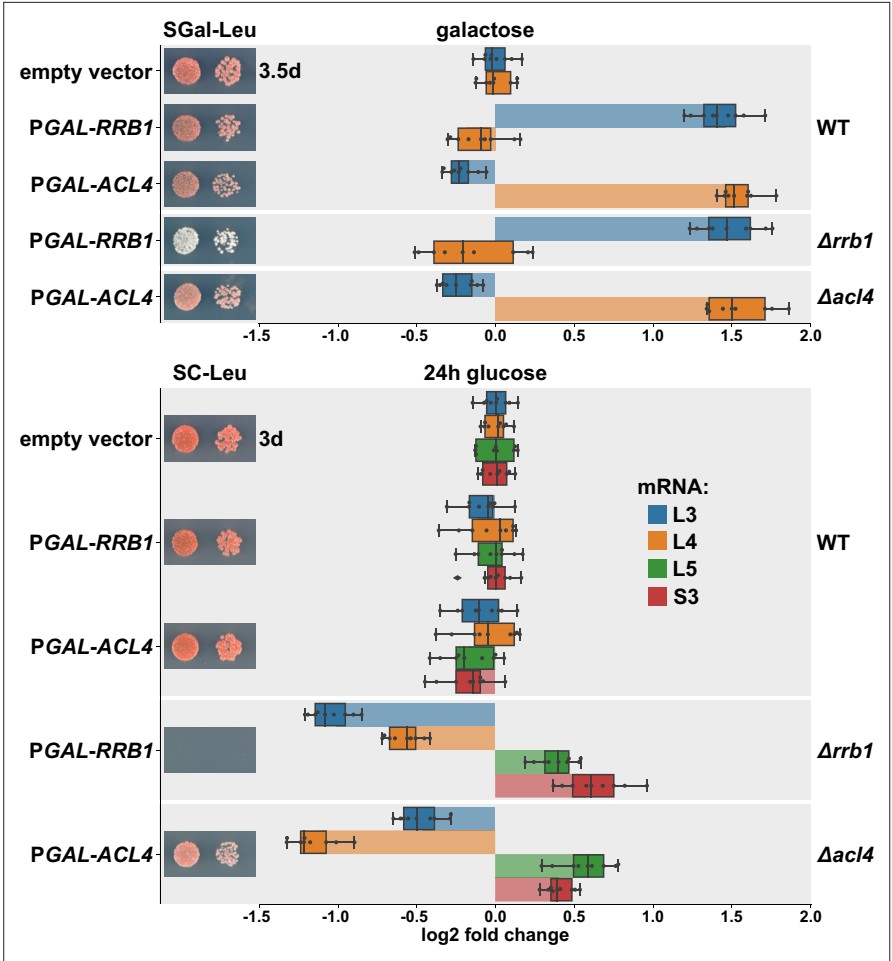

**Figure 6.** Overexpression of Rrb1 and Acl4 increases *RPL3* and *RPL4* mRNA levels. Wild-type (WT), *RRB1* shuffle (*Δrrb1*), and *Δacl4* cells were transformed with an empty vector or plasmids expressing either Rrb1 or Acl4 under the control of the inducible *GAL1-10* promoter. Relative levels of the *RPL3*, *RPL4*, *RPL5*, and *RPS3* mRNAs were determined by qRT-PCR using total RNA extracted from log-phase cells grown in SGal-Leu medium (galactose; upper panel) or shifted for 24 hr to SC-Leu medium (glucose; lower panel). The relative changes in mRNA levels between the different conditions (Rrb1 and Acl4 overexpression or depletion in WT, *Δrrb1*, or *Δacl4* cells) have been normalized to the abundance of each assayed mRNA in WT cells transformed with the empty vector and grown in the same medium. The data shown were obtained from three different WT, *RRB1* shuffle, and *Δacl4* strains (biological triplicates), in each case consisting of a technical triplicate, and they are represented as described in the legend to *Figure 1G*. In addition, the transformed cells were spotted in 10-fold serial dilution steps onto SGal-Leu (galactose) or SC-Leu (glucose) plates, which were incubated at 30°C.

capturing of Rpl3 or Rpl4 by its respective dedicated chaperone would preclude their recognition by the regulatory machinery.

## Overexpression of Rrb1 and Acl4 increases *RPL3* and *RPL4* mRNA levels

Next, we wished to obtain more direct evidence for a positive effect of Rrb1 or Acl4 binding to nascent Rpl3 or Rpl4, respectively, on the abundance of the encoding mRNAs. To this end, we expressed the dedicated chaperones Rrb1 and Acl4 in wild-type cells or in cells either lacking the genomic copy of *RRB1* (*Δrrb1*) or *ACL4* (*Δacl4*) from a monocopy plasmid under the control of the galactose-inducible *GAL1-10* promoter and assessed the levels of the endogenous *RPL3* and *RPL4* mRNAs by qRT-PCR. When grown at 30°C in liquid SGal-Leu medium, overexpression of Rrb1 resulted in a more than 2.5-fold increase in *RPL3* mRNA levels both in wild-type and *Δrrb1* cells (*Figure 6*), whereas a slight decrease in *RPL4* mRNA abundance could be observed. Likewise, overexpression of Acl4 led to a

similarly robust increase in *RPL4* mRNA levels while, at the same time, the abundance of the *RPL3* mRNA was marginally negatively affected. Conversely, depletion of either Rrb1 or Acl4, which as expected entailed either a lethal or an sg phenotype, by growing cells for 24 hr in glucose-containing medium resulted in a more than twofold decrease in *RPL3* or *RPL4* mRNA levels, respectively (*Figure 6*). These findings are consistent with the above observations that mutational inactivation of Acl4 or Rrb1 binding by the BI mutations or the H3E substitution, respectively, augmented the negative regulation of their mRNAs (*Figures 4D and 5A*). Moreover, and also in in agreement with the observed reduction in *Δacl4* cells (*Figure 1G*), Acl4-depleted cells exhibited an almost 1.5-fold decrease in *RPL3* mRNA levels (*Figure 6*). Similarly, *RPL4* mRNA abundance was reduced to a comparable extent upon Rrb1 depletion. However, the levels of other assessed mRNAs, such as the ones encoding Rpl5 or Rps3, were found to be moderately upregulated upon Rrb1 or Acl4 depletion. We presume that this mutual reduction of the other mRNA being regulated by the same machinery might be due a decreased rate of early pre-60S assembly and the concomitant sequestration of Rrb1 or Acl4, which are only available in limiting amounts, by nonincorporated, excess Rpl3 or Rpl4 arising from Acl4 or Rrb1 depletion, respectively. Taken together, we conclude that the availability of the dedicated chaperone for binding to its nascent r-protein client is a crucial parameter for determining the stability of the corresponding mRNA.

## Deregulated expression of Rpl3 and Rpl4 induces their aggregation and abolishes growth in the absence of the E3 ubiquitin ligase Tom1

What could be the physiological reason for the tight regulation of Rpl3 and Rpl4 expression levels and the coupling of the regulatory process to the availability of their dedicated chaperones Rrb1 and Acl4? A previous study of the Deshaies laboratory revealed that aggregation of many r-proteins, including Rpl3 and Rpl4, is largely increased in cells lacking the E3 ubiquitin ligase Tom1 (*Sung et al., 2016a*), and, moreover, different reports have shown that perturbations of ribosome assembly, essentially leading to an accumulation of newly synthesized, nonassembled r-proteins, negatively affect cellular proteostasis (*Sung et al., 2016a*; *Albert et al., 2019*; *Martín-Villanueva et al., 2019*; *Tye et al., 2019*). To assess genetically the impact of excess Rpl3 or Rpl4, we individually overexpressed these two r-proteins from a multicopy plasmid under the control of the inducible *GAL1-10* promoter both in wild-type and *Δtom1* cells. While only a minor effect on growth of wild-type cells could be discerned, overexpression of Rpl3 or Rpl4a in *Δtom1* cells resulted in a more severe growth defect than overexpression of Rpl26 (*Figure 7—figure supplement 1A*), which was previously shown to be degraded by the proteasome upon ubiquitination by Tom1 (*Sung et al., 2016a*; *Sung et al., 2016b*). To evaluate the effect of the Rpl3 and Rpl4a mutant variants that efficiently reduce the negative regulation of their encoding mRNAs, these had therefore to be more moderately overexpressed, again under the transcriptional control of the *GAL1-10* promoter, from monocopy plasmids. Strikingly, overexpression of the BIV and W109C Rpl4a variants as well as the F16A/L17A and R19A/K20A/R21A Rpl3 variants exclusively and strongly compromised, albeit to different extents, the growth of *Δtom1* cells, while overexpression of wild-type Rpl4a and Rpl3 only marginally affected growth (*Figure 7—figure supplement 1B and C*). To exclude that in the case of the overexpressed Rpl3 mutants the observed effects are due to titration of Rrb1, we additionally added the H3E mutation, which, as shown above, abolishes the interaction with Rrb1 (*Figure 5—figure supplement 1*). Notably, presence of the H3E mutation substantially exacerbated the impact of the Rpl3 variants harboring the F16A/L17A and R19A/K20A/R21A substitutions on growth of *Δtom1* cells (*Figure 7—figure supplement 1D*). Therefore, and considering the contribution of Acl4 to Rpl4's soluble expression (*Pillet et al., 2015*), it appears that association of the respective dedicated chaperone has a positive influence on an intrinsically difficult property of newly synthesized Rpl3 and Rpl4, which could consist in their aggregation propensity.

To explore this possibility, we next assessed whether the overexpressed Rpl3 (F16A/L17A and R19A/K20A/R21A, either alone or combined with H3E) and Rpl4a (BIV and W109C) variants, fused to a C-terminal 2xHA tag, would exhibit aggregation in *Δtom1* cells at 30°C. To this end, we shifted wild-type and *Δtom1* cells containing the different monocopy plasmids, pre-grown in liquid medium with raffinose as carbon source, for 4 hr to galactose-containing medium and revealed the inducibly expressed proteins in the total extract and the insoluble pellet fraction by Western analysis using anti-HA antibodies. In agreement with Rpl4 being ubiquitinated in vitro by Tom1 (*Sung et al., 2016a*),

the abundance of wild-type Rpl4a-2xHA was clearly increased in *Δtom1* cells when compared to its levels in wild-type cells (*Figure 7—figure supplement 2A*). In accord with their deregulated expression, the BIV and W109C Rpl4a variants were more abundant than wild-type Rpl4a both in wild-type and *Δtom1* cells. In good correlation with the observed expression levels, the two Rpl4a variants, which could already be detected to some extent in the insoluble fraction of wild-type cells, exhibited a higher occurrence than wild-type Rpl4a in aggregates of *Δtom1* cells. The two Rpl3 variants (F16A/L17A and R19A/K20A/R21A), despite being similarly abundant as wild-type Rpl3 in the total extracts, were considerably enriched in the insoluble fraction of wild-type cells (*Figure 7—figure supplement 2B*). Likewise, while the levels of wild-type Rpl3 were comparable in wild-type and *Δtom1* cells, more Rpl3 was present in the aggregate fraction of cells lacking Tom1. Compared to wild-type Rpl3, the abundance and, even more markedly, the insolubility of the two Rpl3 mutant proteins, especially of the F16A/L17A variant, were substantially increased in the absence of Tom1. Presence of the H3E mutation strongly reduced the amounts of wild-type and mutant Rpl3 in total extracts of wild-type cells; nevertheless, and in contrast to Rpl3.H3E, in particular the H3E/F16A/L17A variant exhibited a notable degree of aggregation. Remarkably, absence of Tom1 restored the levels of wild-type and mutant Rpl3 containing the H3E mutation and resulted in their prominent occurrence in the insoluble fraction. Altogether, the above findings provide evidence that Tom1-mediated clearance of excess Rpl3 and Rpl4 is required to efficiently prevent their aggregation. Moreover, given that wild-type Rpl3 and Rpl4a exert a dosage-dependent negative effect on the growth of *Δtom1* cells and also considering that only Rpl3 and Rpl4a variants inducibly expressed in a deregulated fashion, thus exhibiting higher abundance than their wild-type counterparts in the insoluble fraction, severely affect the growth of cells lacking Tom1, it is reasonable to assume that the detrimental impact on cell growth only sets in once a certain threshold of aggregation has been exceeded.

To gain additional insight into the nature and location of the aggregation process, we examined the fate of the above Rpl4a variants (BIV and W109C) and of the two Rpl3 mutant proteins (F16A/L17A and H3E/F16A/L17A) exhibiting the highest aggregation propensity by fluorescence microscopy. To this end, we transformed monocopy plasmids expressing wild-type and mutant Rpl3 and Rpl4a, fused to a C-terminal yeast codon-optimized mNeonGreen (yOmNG), from the *GAL1-10* promoter into wild-type and *Δtom1* cells, additionally bearing a plasmid providing the nucleolar marker protein Nop58-yEmCherry. Cells were first pre-grown at 30°C in liquid medium with raffinose as carbon source and then shifted for 4 hr to galactose-containing medium. Wild-type Rpl3 and Rpl4a showed, when expressed in wild-type cells, almost exclusively cytoplasmic localization and exhibited only in a fraction of *Δtom1* cells (less than 20%) nucleolar accumulation or enrichment in nucle(ol)ar dots (*Figure 7A*, *Figure 7—figure supplement 2C*). Conversely, the mutant Rpl3 and Rpl4a variants displayed in most *Δtom1* cells a strong fluorescence signal in the nucle(ol)ar compartment. As in the case of the wild-type proteins, we again observed different types of localization patterns, ranging from a rather diffuse nucleolar enrichment, sometimes expanding to the adjacent nucleoplasm, to the appearance of one to several bright nuclear dots or blob-like structures. Given that the fluorescence signal intensity is highest in the latter two morphological states, we presume that these actually correspond to aggregates of excess Rpl3 and Rpl4a, which initially, when still less abundant, can diffusely distribute within the nucleolar phase. In line with the finding that the presence of the H3E mutation enhances the negative impact of Rpl3.F16A/L17A overexpression on growth of *Δtom1* cells (*Figure 7—figure supplement 1D*), the fraction of cells exhibiting extensive nuclear aggregation was higher upon Rpl3.H3E/F16A/L17A expression (*Figure 7—figure supplement 2C*). Taken together, we conclude that the induced overexpression of Rpl3 or Rpl4 leads to their aggregation within the nucle(ol)ar compartment of cells lacking the E3 ubiquitin ligase Tom1.

Having shown that exogenously overexpressed Rpl3 and Rpl4 exhibit aggregation, we next wished to address the physiological effect of their moderate, constitutive surplus expression elicited by inactivation of the negative regulatory network. Notably, the absence of Caf130, Not1's N-terminal CaInD domain, or either of the two NAC subunits, which all cause the deregulated expression of both Rpl3 and Rpl4, conferred synthetic lethality to cells lacking Tom1 (*Figure 7B*, *Figure 7—figure supplement 3B–G*). Even more remarkably, cells simultaneously lacking Cal4, which only increases *RPL4* but not *RPL3* mRNA levels, and Tom1 were also inviable (*Figure 7B*, *Figure 7—figure supplement 3A*), suggesting that deregulated expression of endogenous Rpl4 is already sufficient to confer lethality when excess r-proteins are not degraded by Tom1-dependent clearance. In agreement with excess

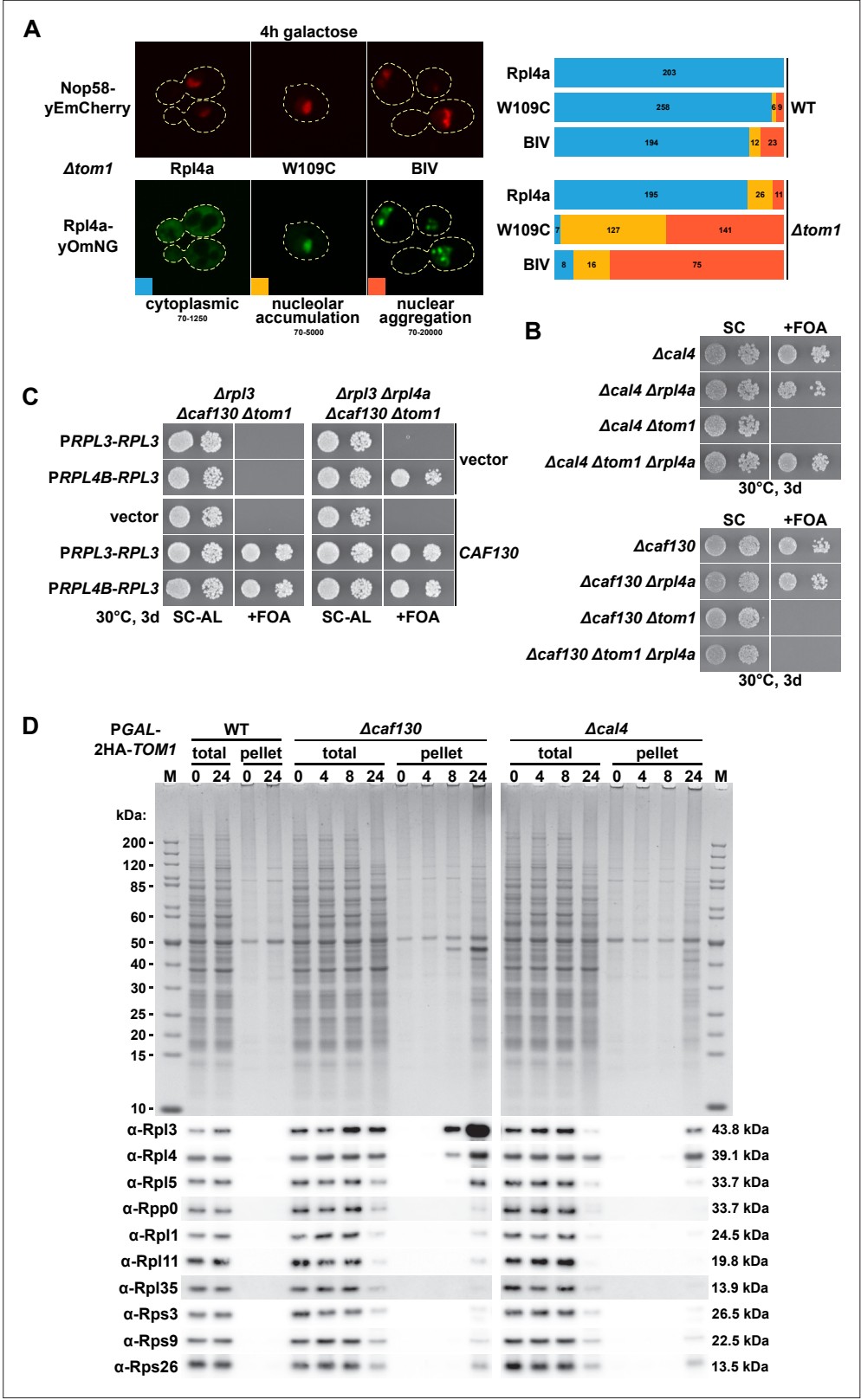

**Figure 7.** Deregulated expression of Rpl3 and Rpl4 induces their aggregation and abolishes growth in the absence of Tom1. (**A**) Overexpressed Rpl4a variants, exhibiting deregulated expression, accumulate in the nucleolus and aggregate in the nucleus in the absence of Tom1. Wild-type (WT) and *Δtom1* strains were co-transformed with plasmids expressing the indicated Rpl4a variants, C-terminally fused to a yeast codon-optimized

*Figure 7 continued on next page*

*Figure 7 continued*

mNeonGreen (yOmNG), under the control of the inducible *GAL1-10* promoter and a plasmid expressing Nop58-yEmCherry to indicate the subcellular position of the nucleolus. Cells were grown at 30°C in SRaf-Leu (raffinose) medium, and expression of the Rpl4a variants was induced for 4 hr with 2% galactose. The left panel shows representative examples of the three types of observed localizations (cytoplasmic, nucleolar accumulation, and nuclear aggregation). The images shown were acquired from *Δtom1* cells expressing wild-type Rpl4a or the two indicated Rpl4a variants and are displayed according to the indicated 16-bit brightness level ranges (min-max); note that the cytoplasmic signal, due to these parameter choices, is not well visible in the examples highlighting the nucleolar accumulation and nuclear aggregation. The right panel shows proportional bar graphs based on the number of counted cells displaying each of the three typical localizations (blue: cytoplasmic; yellow: nucleolar accumulation; red: nuclear aggregation). (**B**) Reduced expression of Rpl4 suppresses the lethality of *Δcal4/Δtom1* but not of *Δcaf130/Δtom1* cells. The indicated single, double, and triple deletion strains, all derived from tetratype tetrads, were spotted in 10-fold serial dilution steps onto SC and SC + fluoroorotic acid (FOA) (+FOA) plates. (**C**) Reduced expression of both Rpl3 and Rpl4 efficiently suppresses the lethality of *Δcaf130/Δtom1* cells. Empty vector (YCplac111) or plasmids harboring *RPL3*, expressed either from the *RPL3* or *RPL4B* promoter, and empty vector (pASZ11) or a plasmid containing *CAF130*, expressed from the *ADH1* promoter, were co-transformed into *RPL3/CAF130* (*Δrpl3/Δcaf130*) double shuffle strains additionally bearing chromosomal deletions of *TOM1* (*Δtom1*; left panel) or both *TOM1* and *RPL4A* (*Δtom1/Δrpl4a*; right panel). Transformants were restreaked on SC-Ade-Leu plates, and cells were then spotted in 10-fold serial dilution steps onto SC-Ade-Leu (SC-AL) and SC + FOA-Ade-Leu (+FOA) plates. (**D**) Depletion of Tom1 in *Δcaf130* or *Δcal4* cells leads to the aggregation of Rpl3 and/or Rpl4, thereby perturbing overall cellular proteostasis. WT, *Δcaf130*, or *Δcal4* cells, expressing N-terminally 2xHA-tagged Tom1 under the transcriptional control of the *GAL1* promoter from the genomic locus (P*GAL*-2HA-*TOM1*), were grown at 30°C in YPGal medium and then shifted for up to 24 hr to YPD medium. Cells were harvested after the indicated times of growth in YPD medium (0, 4, 8, or 24 hr). The total extracts (total) and the insoluble pellet fractions (pellet) were analyzed by SDS-PAGE and Coomassie staining (upper panel) and by Western blotting using the indicated antibodies (lower panel).

The online version of this article includes the following source data and figure supplement(s) for figure 7:

**Source data 1.** Original image files of the Coomassie-stained gel and the Western blots shown in *Figure 7D*, including a PDF file showing the full gel and blots and indicating the cropped areas.

**Figure supplement 1.** Overexpression of Rpl3 and Rpl4 variants affects the growth of *Δtom1* cells.

**Figure supplement 2.** Overexpressed Rpl3 and Rpl4 variants aggregate in *Δtom1* cells.

**Figure supplement 2—source data 1.** Original image files of the Western blots shown in *Figure 7—figure supplement 2A*, including a PDF file showing the full blots and indicating the cropped areas.

**Figure supplement 2—source data 2.** Original image files of the Western blots shown in *Figure 7—figure supplement 2B*, including a PDF file showing the full blots and indicating the cropped areas.

**Figure supplement 3.** Absence of individual components of the regulatory machinery confers lethality to cells lacking Tom1.

**Figure supplement 4.** Identification of aggregated proteins in *Δcaf130* cells upon genetic depletion of Tom1.

**Figure supplement 4—source data 1.** Original image file of the Coomassie-stained gel shown in *Figure 7—figure supplement 4*, including a PDF file showing the full gel and indicating the cropped area.

Rpl4 being directly responsible for the synergistic growth defect, deregulated Rpl4a expression, this time enabled by the presence of the BIV mutations in the genomic *RPL4A* copy, did not support growth upon genetic depletion of Tom1 (*Figure 7—figure supplement 3H*). Accordingly, lowering the levels of synthesized Rpl4, by deleting the *RPL4A* gene, efficiently suppressed the synthetic lethality of *Δcal4/Δtom1* cells (*Figure 7B*). Providing Rpl4 exclusively from the *RPL4B* locus, however, was not sufficient to restore growth of *Δcaf130/Δtom1* cells (*Figure 7B*), suggesting that not only excess supply of Rpl4 but also of Rpl3 is individually detrimental for cells lacking Tom1. In line with this notion, expression of the Rpl3.F16A/L17A variant, under the transcriptional control of its own promoter from a multicopy plasmid, abolished growth of Tom1-depleted cells (*Figure 7—figure supplement 1*). Even more importantly, only simultaneously reducing the abundance of Rpl4 and Rpl3 by deleting *RPL4A* and expressing Rpl3 from the weaker *RPL4B* promoter (*Zeevi et al., 2011*; *Knight et al., 2014*), but not solely lowering Rpl3 levels, permitted efficient growth of *Δcaf130/Δtom1* cells (*Figure 7C*). Together, the genetic data convincingly demonstrate that the moderate constant surplus supply, around twofold at the mRNA level, of either Rpl3 or Rpl4 is sufficient to perturb growth and possibly also proteostasis of cells lacking Tom1. To explore whether deregulated expression of Rpl3

and/or Rpl4 indeed promotes their aggregation, we assessed their occurrence in the insoluble fraction of Δcaf130 and Δcal4 cells upon genetic depletion of Tom1. Strikingly, Rpl3 appeared already after 8 hr in glucose-containing medium in the insoluble fraction of Δcaf130/P*GAL*-2xHA-*TOM1* cells, and Western analysis also revealed some accumulation of Rpl4 at this time point (*Figure 7D*). After 24 hr of Tom1 depletion, a massive aggregation of Rpl3 and, albeit to a lesser extent, of Rpl4 could be observed. Concomitantly, many additional proteins showed up in the insoluble fraction, including, as indicated by Western analysis, several r-proteins (*Figure 7D*), suggesting that aggregation of Rpl3 and Rpl4 leads to an extensive perturbation of cellular proteostasis. Notably, the abundance of the other directly tested r-proteins was substantially decreased in the total extracts after 24 hr of Tom1 depletion, presumably reflecting, as recently described (*Albert et al., 2019*), the decreased transcription of their encoding genes as a result of lower Ifh1 promoter occupancy in order to alleviate the proteotoxic burden. Mass spectrometry (MS) analysis of the major distinct gel bands confirmed the high prevalence of almost all r-proteins in the insoluble fraction and additionally revealed the presence of a broad range of different proteins, including ribosome biogenesis factors, proteasome subunits, general chaperones, and translation factors (*Figure 7—figure supplement 4*, *Supplementary file 8*). As expected, depleting Tom1 for 24 hr in Δcal4 cells resulted, to a similar extent as observed above in Δcaf130 cells, in extensive aggregation of Rpl4 and the concomitant presence of many additional proteins in the insoluble fraction (*Figure 7D*). We note that the deregulated expression of Rpl3 and/or Rpl4, in the absence of Tom1-dependent clearance of their excess occurrence, promotes their aggregation and entails a loss of overall proteostasis, which very likely accounts for the observed synthetic lethality of Δcaf130/Δtom1 and also of Δcal4/Δtom1 cells.

Taken together, we conclude that the two dedicated chaperones Rrb1 and Acl4 intimately cooperate with the regulatory machinery to provide optimal levels of assembly-competent Rpl3 and Rpl4. By perfectly balancing their de novo synthesis, pre-60S assembly can be sustained at the highest possible rate without requiring the Tom1-mediated degradation of excess Rpl3 and Rpl4, which

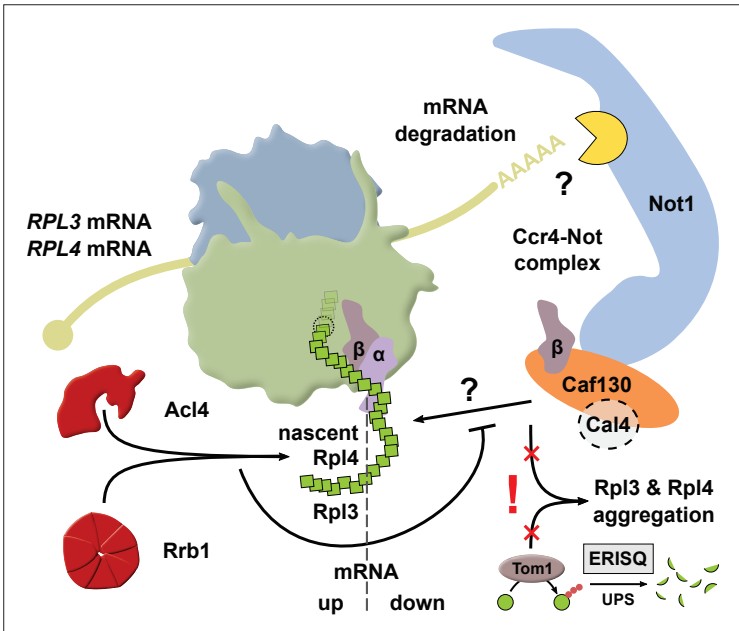

**Figure 8.** Simplified model showing how availability of the dedicated chaperone Rrb1 or Acl4 and the here uncovered regulatory network cooperate to balance Rpl3 and Rpl4 expression by co-translationally regulating *RPL3* and *RPL4* mRNA levels. The question marks indicate that it remains to be determined how nascent Rpl3 or Rpl4 are recognized by the regulatory machinery and how this leads to the degradation, presumably involving a component of the Ccr4-Not complex, of the *RPL3* or *RPL4* mRNAs. Also included in the model and highlighted by an exclamation mark is the finding that surplus production of Rpl3 and/or Rpl4, for example, elicited by inactivation of the regulatory machinery, may lead to their aggregation when cells lack the E3 ubiquitin ligase Tom1, which is required for mediating the degradation of excess r-proteins by the ubiquitin proteasome system (UPS) via the so-called ERISQ (excess ribosomal protein quality control) pathway. The α and β subunit of the nascent polypeptide-associated complex (NAC) are denoted as α and β, respectively. For more details, see Discussion.

would, as a last resort, be necessary to avoid their aggregation and, ultimately, a potentially deleterious collapse of cellular proteostasis. Importantly, the above-described data also strongly suggest that the main, physiologically relevant targets of the regulatory machinery are the *RPL3* and *RPL4A/B* mRNAs.

## Discussion

In this study, we have unveiled a novel, fascinating mechanism enabling the tight co-translational regulation of r-protein expression that is of physiological importance for the maintenance of cellular proteostasis. On the basis of our data and the current state of knowledge, we propose the following model for how the de novo synthesis of Rpl3 and Rpl4 is fine-tuned to meet the demands of ribosome assembly and, at the same time, protect cells from the potentially detrimental effects of their surplus production (*Figure 8*). Under normal conditions, that is, when ribosome assembly proceeds at an optimal rate, the dedicated chaperones Rrb1 and Acl4 are available in sufficient amounts to capture their nascent r-protein client Rpl3 or Rpl4, respectively, as the specific chaperone-binding segment emerges from the exit tunnel on the surface of the 60S r-subunit, thereby leading to a stabilization of the *RPL3* or *RPL4* mRNA that is in the process of being translated. Conversely, when ribosome biogenesis occurs at reduced pace and Rpl3 or Rpl4 cannot get efficiently integrated into the developing pre-60S particles, Rrb1 and Acl4 get sequestered by their unassembled r-protein partner in the nucleus and, therefore, can no longer bind to nascent Rpl3 or Rpl4 in a timely manner. In this case, the presumed, co-translational interaction of NAC with the regulation-conferring segment on these two r-proteins persists long enough to enable the recruitment of or the transfer to the regulatory machinery, which, likely via the Caf1-Ccr4 deadenylase module of the Caf130-associated Ccr4-Not core complex, promotes the degradation of the physically connected *RPL3* or *RPL4* mRNA. The finding that overexpression of Rrb1 or Acl4 leads to a further, specific increase in *RPL3* or *RPL4* mRNA levels strongly suggests that the two dedicated chaperones are actually present in somewhat limiting amounts and that, even under normal growth conditions, there is a constant competition between Rrb1 or Acl4 and the regulatory machinery for binding to nascent Rpl3 or Rpl4 (*Figure 6*), thus, conferring high sensitivity to the regulatory process. Accordingly, the functional utility of dedicated chaperones can be extended to the purpose of serving as molecular rheostats that, in the case of Rrb1 and Acl4, continuously sense the status of early pre-60S assembly by surveying the levels of free Rpl3 or Rpl4, respectively, and thereby coordinate the production of new Rpl3 and Rpl4 with their actual consumption during biogenesis of 60S r-subunits. The need for such a tight regulation becomes apparent when Rpl3 and/or Rpl4 expression is deregulated, such as in the absence of Caf130 or Cal4, and cells cannot clear these excessively produced r-proteins via their Tom1-mediated ubiquitination (ERISQ pathway) and subsequent proteasomal degradation. In this setting, Rpl3 and/or Rpl4 undergo massive aggregation, which likely accounts for the observed collapse of overall proteostasis and the inability to sustain cell growth (*Figure 7B and D*). We conclude that, depending on the cell's proteostatic state, the meticulous adjustment of the abundance of unassembled Rpl3 and/or Rpl4, achieved by a properly functioning interplay between the regulatory machinery and the dedicated chaperones Rrb1 and Acl4, may become essential to maintain cellular proteostasis. Moreover, the here-described autoregulatory feedback loop constitutes together with the ERISQ pathway a robust buffering system to prevent cells from experiencing the harmful impact of excess Rpl3 and/or Rpl4.

As mentioned in the Introduction, the complementary action of Ifh1 and Sfp1, which are predominantly required for activation of either category I and II (Ifh1) or category III (Sfp1) RPG promoters, ensures the co-regulated expression of all RPGs under most conditions (*Zencir et al., 2020*; *Shore et al., 2021*). Ifh1, owing to its Utp22-dependent sequestration in the CURI complex, is also employed, by sensing the 90S assembly status, to coordinate the transcriptional output of most RPGs with that of the 35S pre-rRNA (*Albert et al., 2016*). In addition, in response to a ribosome assembly stress, leading to the aggregation of unassembled r-proteins, Ifh1 gets rapidly displaced from RPG promoters and appears to accumulate in an insoluble nucle(ol)ar fraction (*Albert et al., 2019*); the concomitant decrease in Ifh1-dependent RPG transcription then helps to alleviate the proteostatic stress by reducing the production of new r-proteins. Notably, the *RPL3* and *RPL4A/B* genes contain category III promoters, and they are the RPGs whose efficient transcription, while being rather insensitive to Ifh1 depletion or the ribosome assembly stress response (RASTR), shows the highest Sfp1 dependence (*Albert et al., 2019*; *Zencir et al., 2020*). Accordingly, the here-described co-translational regulation

of *RPL3* and *RPL4* mRNA levels represents an elegant mechanism to specifically reduce the de novo synthesis of Rpl3 and Rpl4 when their unassembled levels exceed, due to a perturbation of ribosome assembly, the buffering capacity of the dedicated chaperones Rrb1 and Acl4. While the above would suggest a special relevance for rapidly responding to certain stress conditions, our data indicate that the regulatory mechanism also continuously operates under normal growth conditions. In line with a constant adjustment of their transcript levels via a regulated degradation pathway, the *RPL3* and *RPL4A/B* transcripts are among the five RPG mRNAs exhibiting markedly shorter half-lives than all other RPG mRNAs (*Wang et al., 2002*). What could be the reason for their different transcriptional regulation and the need to tightly control the levels of unassembled Rpl3 and Rpl4? Both Rpl3 and Rpl4, by associating shortly before or after the generation of the 27SA$_2$ pre-rRNA, are among the earliest assembling large subunit r-proteins, and they fulfill a central role for the compaction and/or stabilization of the earliest pre-60S particles (*Rosado et al., 2007*; *Pöll et al., 2009*; *Gamalinda et al., 2014*; *Pillet et al., 2015*; *Joret et al., 2018*). Therefore, to sustain optimal rates of 60S production and avoid the costs and impact of abortive pre-60S assembly, it is necessary to warrant a sufficient supply of assembly-competent Rpl3 and Rpl4. The temporary nuclear storage of Rpl3 and Rpl4 in complex with their dedicated chaperone not only provides the required buffering capacity to rapidly respond to short-term increases in assembly demand but also the means to relay and directly connect the status of early pre-60S assembly to the rate of the two r-proteins' de novo synthesis. Compared to a purely transcription-based adaptation of protein levels, the uncovered regulatory mechanism, owing to the constant supply of new *RPL3* and *RPL4A/B* mRNAs that are then either translated or subjected to regulated degradation, has the evident advantage of enabling a more rapid response. We conclude that the accurate functioning of this regulatory system is instrumental to avoid the impact of both the surplus and insufficient supply of Rpl3 and/or Rpl4.

## Potential mechanism of substrate recognition and mRNA degradation

While our study has identified several of the involved regulatory components (NAC, Caf130, Cal4, and Not1) as well as the regulation-conferring segment on the two nascent r-proteins and some key residues therein, the mechanism and selectivity of the substrate recognition process and the component(s) mediating mRNA degradation remain to be determined. Considering the mandatory requirement of a physical connection with Not1 for negative regulation of *RPL3* and *RPL4* mRNA levels to occur, an involvement of one of the associated core components of the Ccr4-Not complex can be supposed. Since it is well established that the Ccr4-Not complex plays an important role in the decay of cytoplasmic mRNAs, a process that is initiated by deadenylation of the poly(A) tail (*Parker, 2012*), we consider it highly likely that its Caf1-Ccr4 deadenylase module promotes degradation of the *RPL3* and *RPL4* mRNA. Notably, the utilization of a defined segment of the encoded nascent polypeptide constitutes, to the best of our knowledge, an unprecedented mechanism of recruitment of the Ccr4-Not complex to a substrate mRNA, which is more conventionally achieved through interactions with either the poly(A)-binding protein or rather general as well as specific RNA-binding proteins (*Parker, 2012*; *Wahle and Winkler, 2013*; *Bresson and Tollervey, 2018*), but, as recently shown, can also occur via the accommodation of the Not5-NTD in the ribosomal E-site of mRNAs displaying low codon optimality (*Buschauer et al., 2020*). Regardless of the recruitment mechanism, sensing of the mRNA's translation status appears in many cases to be an important aspect for enabling selective mRNA degradation.

How are nascent Rpl3 and Rpl4, in the absence of Rrb1 or Acl4 binding, recognized as signals for the selective recruitment of the Ccr4-Not complex in order to initiate the degradation of the encoding mRNAs? The strict requirement of both NAC subunits for negative regulation indicates that heterodimeric NAC, owing to its established role as a ribosome-associated chaperone that binds nascent chains (*Deuerling et al., 2019*), is involved in the substrate recognition process. Given that the N-terminal tail of NAC-β can insert, as seen in the cryo-EM structure of a reconstituted *Caenorhabditis elegans* NAC-60S complex, up to the constriction point of the polypeptide exit tunnel to sense and bind to nascent chains (*Gamerdinger et al., 2019*), we assume that the yeast NAC-β subunits may be responsible for establishing the initial contact with the N-terminal part of the regulation-conferring segment of nascent Rpl3 and Rpl4. Conspicuously, in both cases the residues identified as being crucial for conferring regulation, F16/L17 and W109, respectively, are followed by a segment of around 30 amino acids that is also required to confer maximal regulation. Considering that around 25–30 amino

acids are generally buried within the exit tunnel (*Bhushan et al., 2010*; *Wilson et al., 2016*; *Döring et al., 2017*), we speculate that the residues following these key residues, especially those potentially extending from the constriction point of the tunnel to the peptidyl transferase center, may have a propensity for stalling the nascent chain in the exit tunnel. The immediate proximity or even partial overlap of the Rrb1- or Acl4-binding site and the key residues of the regulation-conferring segment suggests that timely association with the dedicated chaperone may preclude NAC from interacting with nascent Rpl3 or Rpl4 and allow translation to proceed. Conversely, if these are not swiftly enough captured by their dedicated chaperone, NAC could sense and bind to the regulation-conferring segment, an event that presumably fortifies stalling and, thereby, further decreases the speed of translation. This would provide the necessary time window to judiciously decide about the fate of the stalled nascent ribosome-nascent chain complex (RNC) and its associated mRNA. If the dedicated chaperone associates sufficiently fast with its fully or partially exposed binding site on nascent Rpl3 or Rpl4, the concomitant displacement of NAC may generate a pulling force that might be necessary to overcome stalling, and the precocious degradation of the encoding mRNA can be avoided. On the other hand, when Rrb1 or Acl4 are not available in a high enough concentration in the cytoplasm, the probability of channeling the stalled RNCs to the regulated mRNA degradation pathway increases over time. Considering the high selectivity of the regulatory process and that NAC-β utilizes its NAC domain to interact either with NAC-α or Caf130, it is reasonable to postulate that Caf130-associated NAC-β then takes over the nascent Rpl3 and Rpl4 substrates. To efficiently channel the selected mRNAs to regulated degradation and confer directionality to the process, the interaction of NAC-β with Caf130 is expected to increase the affinity for the substrate, possibly through the formation of a dedicated substrate-binding surface together with Caf130 and/or Not1's CaInD domain. At present, it is not clear why Cal4 only participates in the regulation of the *RPL4* mRNA, but, given its robust association with Caf130, we assume that Cal4 either makes an essential contribution to substrate recognition or is required for conferring the necessary strength to RNC stalling such that the associated mRNA cannot evade its degradation.

The above-described scenario would fit well with the recently proposed role of NAC as a triage factor that promotes the faithful transfer of nascent chains to the proper targeting machinery or chaperone-assisted folding pathway (*Hsieh et al., 2020*). Moreover, our findings are reminiscent of the previous observation that mammalian mRNAs encoding proteins whose signal sequence is inefficiently recognized by the signal recognition particle (SRP) are, albeit by a yet to be determined mechanism, selectively degraded (*Karamyshev et al., 2014*). In analogy to the SRP-recognized sequences of secretory proteins, the binding sites on r-proteins, especially the N-terminally located segment on Rpl3, that enable the co-translational recruitment of dedicated chaperones could also be viewed as highly specific signal sequences. In the special case of nascent Rpl3 and Rpl4, the timely association with Rrb1 or Acl4 not only enables their fail-safe production as assembly-competent r-proteins but also prevents the degradation of the encoding mRNAs. Clearly, future experiments will be required to challenge the presented model, unveil the exact nature of the uncovered regulatory mechanism, and reveal whether it also exists in evolutionary more complex eukaryotes.

## Implications of perturbed r-protein homeostasis for developmental disorders

Notably, our study highlights the r-proteins Rpl3 and Rpl4 as potential drivers of cellular protein aggregation. Our findings therefore further reinforce the notion that the aggregation of unassembled r-proteins represents a threat to the maintenance of cellular proteostasis, which, as shown by previous studies, yeast cells try first to avoid by clearing excess r-proteins via Tom1-mediated degradation (ERISQ pathway) and then to resolve by activating a stress response pathway, referred to as RASTR or RPAS, that increases the protein folding, disaggregation, and degradation capacity and decreases Ifh1-dependent transcription of RPGs (*Sung et al., 2016a*; *Albert et al., 2019*; *Tye et al., 2019*). Importantly, the reduced supply of single r-proteins, resulting in the accumulation of orphan r-proteins, has also recently been shown to cause proteotoxic stress and cell elimination in *Drosophila* (*Baumgartner et al., 2021*; *Recasens-Alvarez et al., 2021*). Moreover, it is well established that mutations in around 20 different RPGs, mostly leading to haploinsufficiency of individual r-proteins, result in the development of a ribosomopathy called Diamond-Blackfan anemia (DBA) (*Narla and Ebert, 2010*; *Danilova and Gazda, 2015*; *Da Costa et al., 2018*; *Aspesi and Ellis, 2019*), whose

defining characteristics include reduced proliferation and increased apoptosis of erythroid progenitor cells, raising the possibility that proteotoxic stress could contribute to the manifestation of DBA (*Recasens-Alvarez et al., 2021*). Further, several unassembled r-proteins, especially RPL5 and RPL11 in the context of the 5S RNP but also RPL4 and RPL26, whose yeast counterparts are established Tom1 targets (*Sung et al., 2016a*; *Sung et al., 2016b*), interact with the E3 ubiquitin ligase MDM2 and thereby inhibit the ubiquitination and degradation of the apoptosis-promoting transcription factor p53 (*Bursac et al., 2014*; *Pelava et al., 2016*). In the case of RPL4, both its overexpression and depletion, the latter in an RPL5- and RPL11-dependent manner, lead to p53 stabilization (*He et al., 2016*). Moreover, a possible connection between RPL3 and RPL4 variants and DBA is suggested by the identification of missense mutations both in *RPL3* (one DBA patient; His11 to Arg substitution, unknown significance for disease manifestation) and *RPL4* (one individual with DBA-like phenotypes; Val-Leu insertion between Ala58 and Gly59) (*Gazda et al., 2012*; *Jongmans et al., 2018*). Besides in DBA, increased p53 activity also plays a pivotal role in eliciting tissue-specific defects in a variety of different developmental syndromes (*Bowen and Attardi, 2019*). In this respect, it is worth mentioning that genetic changes in *HUWE1*, encoding the ortholog of Tom1, are associated with multiple neurodevelopmental disorders, prominently including X-linked intellectual disability (*Giles and Grill, 2020*), and that reduced HUWE1 levels, due to a disease-causing mutation, increase p53 signaling (*Aprigliano et al., 2021*). Moreover, individuals with mutations in *CNOT1* exhibit a broad range of neurodevelopmental phenotypes, most consistently resulting in intellectual disability, development, speech, and motor delay, and hypotonia (*Vissers et al., 2020*). Taking into account the findings of our study and the above considerations, perturbed proteostasis, elicited by unassembled r-proteins such as RPL3 and RPL4, could not only contribute to the development of DBA but possibly also influence the aging process and be of relevance to the etiology of diverse developmental disorders and even neurodegenerative diseases of protein aggregation (*Kaushik and Cuervo, 2015*; *Szybińska and Leśniak, 2017*; *Maor-Nof et al., 2021*).

## Materials and methods
### Yeast strains, genetic methods, and plasmids

The *S. cerevisiae* strains used in this study (listed in *Supplementary file 1*) are derivatives of W303 (*Thomas and Rothstein, 1989*). For Y2H analyses, the reporter strain PJ69-4A was used (*James et al., 1996*). Deletion disruption, C-terminal tagging at the genomic locus, and N-terminal 2xHA-tagging of *TOM1* under the transcriptional control of the *GAL1* promoter were performed as described (*Longtine et al., 1998*; *Janke et al., 2004*). Strains harboring combinations of different gene deletions, tagged alleles, and/or the conditional 2xHA-*TOM1* allele were generated by crossing and, upon sporulation of the diploids, tetrad dissection using a Singer MSM System series 200 micromanipulator (Singer Instruments, Roadwater, UK). To generate strains harboring *rpl4a* alleles at the genomic locus, a two-step allele replacement strategy was employed (*Klöckner et al., 2009*). Briefly, wild-type *RPL4A* and the *rpl4a.W109C*, *rpl4a.BI-mt*, *rpl4a.BII-mt*, *rpl4a.BIII-mt*, and *rpl4.BIV-mt* alleles, excised from plasmid and bearing the native *RPL4A* promoter and terminator regions, were integrated into haploid *rpl4a*::klURA3NPT2 mutant cells (YBP143 and YBP144) by homologous recombination, and correctness of the allele replacement was verified by PCR and sequencing. To combine these alleles with either the *Δrpl4b* or *Δacl4* null mutation, strains harboring the integrated *RPL4A* wild-type and *rpl4a* mutant alleles were either crossed with a *Δrpl4a/Δrpl4b* pHT4467Δ-*RPL4A* (YBP34 or YBP52) or a *Δrpl4a/Δacl4* YCplac33-*ACL4* (YBP98 or YBP104) strain, and, upon sporulation and tetrad dissection, haploid spore clones with the correct genotype were selected. Preparation of media, yeast transformation, and genetic manipulations was done according to established procedures. All recombinant DNA techniques were according to established procedures using *Escherichia coli* DH5α for cloning and plasmid propagation. All cloned DNA fragments generated by PCR amplification were verified by sequencing. More information on the plasmids, which are listed in *Supplementary file 2*, is available upon request.

## Isolation of Δacl4 suppressors and identification of candidate mutations by high-throughput sequencing

Spontaneous suppressors of the Δacl4 sg phenotype were isolated from nine different Δacl4 null mutant (acl4::HIS3MX4: YKL697, YKL698, YKL700, and YKL701; acl4::natNT2: YKL703, YKL704, YKL707, YKL708, and YBP255) and two different Δacl4/Δrpl4a double mutant (acl4::natNT2/rpl4a::HIS3MX4: YBP98 and YBP104) strains by growing serial dilutions or restreaks of these strains on YPD plates at 16, 23, and 30°C. After a further restreak round on YPD plates, 53 suppressor strains were retained and their genomic DNA was isolated. Upon high-throughput sequencing of the mutant genomes, clear candidate mutations could be identified by bioinformatics analyses for 48 independent suppressor strains. The 47 different mutations mapped to only four different genes, namely, CAF130 (35 different mutations), YJR011C/CAL4 (7), NOT1 (4), and RPL4A (1) (see **Supplementary file 3**).

Genomic DNA was extracted from cell pellets containing 20 $OD_{600}$ units of the parental control strains and the suppressor mutants, which had been grown in YPD to an $OD_{600}$ of around 1, exactly as previously described (**Thoms et al., 2018**). To estimate the integrity of the isolated genomic DNA, 2.5 µl of the preparation was migrated on a 1% agarose gel. The concentration of the genomic DNA was determined with the Qubit dsDNA BR Assay Kit (Invitrogen, Carlsbad, USA) on a Qubit 2.0 fluorimeter (Invitrogen).

Libraries were generated from 1 µg of genomic DNA, and high-throughput sequencing was performed on a HiSeq 3000 instrument (Illumina, San Diego, USA). Library preparation and Illumina sequencing were carried out by the Next Generation Sequencing (NGS) Platform of the University of Bern (Switzerland). The raw reads (paired-end reads of 150 bp) were processed according to the following procedure. After performing a quality check with FastQC v0.11.7 (fastqc: https://www.bioinformatics.babraham.ac.uk/projects/fastqc/), all the reads were filtered for quality (min 20) and truncated to 100 bp with Sickle v1.29 (**Joshi and Fass, 2011**) and then mapped with BWA-MEM v0.7.10 (**Li and Durbin, 2010**) to the S. cerevisiae reference genome R64-1-1.90 downloaded from Ensembl (**Yates et al., 2020**). The SAM files were sorted and converted to BAM files with SAMtools v1.2 (**Li, 2011**). Single-nucleotide variants (SNVs), as well as small insertions and deletions (Indels), were called with SAMtools and BCFtools v1.2 (**Li, 2011**). Variant annotation was added with SnpEff v4.3 (**Cingolani et al., 2012b**); then, variants were filtered with SnpSift (**Cingolani et al., 2012a**) to keep homozygous variants that are not found in the parental control strain and that are not 'synonymous' or 'intergenic,' leading to an annotated and curated Variant Call Format (VCF) file. Results were viewed and validated with the Integrative Genomics Viewer (IGV) software (**Thorvaldsdóttir et al., 2013**). The raw reads have been deposited at the European Nucleotide Archive (ENA) under the study accession number PRJEB45852.

## RNA extraction

For RNA-seq and the determination of mRNA levels by real-time qRT-PCR (experiments shown in **Figure 1G**), total RNA was extracted by the hot acid-phenol method (**Ausubel et al., 1994**). Briefly, yeast cells were exponentially grown in YPD medium at 30°C and 10 ml of each culture was harvested by centrifugation. Cells were then washed once in ice-cold water and frozen in liquid nitrogen. The cell pellets were resuspended in 400 µl of TES solution (10 mM Tris-HCl pH 7.5, 10 mM EDTA, and 0.5% SDS) and 400 µl of acid phenol was added. The tubes were vigorously vortexed and incubated at 65°C for 45 min with occasional vortexing. The extraction mix was cooled down on ice for 5 min, and the upper aqueous phase was recovered after centrifugation (5 min, 13,500 rpm, 4°C). Following a second acid phenol (400 µl) and a chloroform (400 µl) extraction, the aqueous phase was transferred into a new tube and the RNA was precipitated by the addition of 40 µl of 3 M sodium acetate pH 5.3 and 1 ml of ice-cold ethanol. After centrifugation (5 min, 13,500 rpm, 4°C), the RNA pellet was washed with 900 µl of ice-cold 70% ethanol, collected by centrifugation, briefly air-dried, and resuspended in 100 µl nano-pure water. RNA concentrations were determined using a NanoDrop 1000 spectrophotometer (Thermo Fisher Scientific, Waltham, USA). Then, 5 µg of total RNA was treated with DNase (DNA-free Kit for DNase Treatment & Removal; Invitrogen).

For most other qRT-PCR experiments, total RNA was prepared from around 6 $OD_{600}$ of yeast cells, which were grown in YPD medium or the appropriate synthetic medium (to maintain transformed plasmids) and harvested at an $OD_{600}$ of around 0.6, by the formamide-EDTA extraction method (**Shedlovskiy et al., 2017**). Briefly, frozen cell pellets were resuspended in 350 µl of FAE solution (98%

formamide and 10 mM EDTA), vortexed, and incubated at 65°C for 10 min in a thermoshaker set to 1200 rpm. The extraction mix was centrifuged (5 min, 13,500 rpm, 4°C) and, to avoid taking unbroken, pelleted cells, only 300 µl of the clear supernatant were transferred into a new tube. Then, RNA was precipitated by the addition of 40 µl of 3 M sodium acetate pH 5.3 and 1 ml of ice-cold ethanol. After centrifugation (5 min, 13,500 rpm, 4°C), the RNA pellet was washed with 900 µl of ice-cold 70% ethanol, collected by centrifugation, briefly air-dried, and resuspended in 200–300 µl nano-pure water. RNA concentrations were determined using a NanoDrop 1000 or a NanoDrop One spectrophotometer (Thermo Fisher Scientific).

For assessing the effects of TORC1 inhibition on mRNA levels (*Figure 1—figure supplement 2B*), cells were grown in YPD medium at 30°C to an $OD_{600}$ of around 0.6 and treated for 20 min with 200 ng/ml rapamycin (LC Laboratories, Woburn, USA). For assessing whether ongoing translation is required for negative regulation (*Figure 4—figure supplement 2*), cells were cultured at 30°C in SGal-Leu medium and treated for different durations with 200 µg/ml cycloheximide (C7698; Sigma-Aldrich, St. Louis, USA). In both cases, cells were rapidly harvested by mixing 700 µl of culture with 700 µl of ice-cold ethanol and immediate freezing in liquid nitrogen. Frozen samples were stored at –80°C and, after thawing, centrifuged for 5 min at 13,500 rpm. Cell pellets were washed once with 100 µl $dH_2O$ and resuspended in 50 µl FAE solution. Total RNA was prepared by the formamide-EDTA extraction method as described above, except that RNA was precipitated from 40 µl of clear supernatant upon addition of 5 µl 3 M sodium acetate pH 5.3 and 100 µl ice-cold ethanol and that the air-dried RNA pellet was resuspended in 30 µl nano-pure water.

## Determination of mRNA levels by qRT-PCR

The isolated RNAs were diluted to 5 ng/µl and 9 µl (45 ng of total RNA) of these dilutions were used to prepare the reaction mixes for the real-time qRT-PCR using the KAPA SYBR FAST One-Step Universal kit (Roche, Basel, Switzerland) according to the manufacturer's instructions. Reaction mixes (60 µl) consisting of 30 µl KAPA SYBR FAST qPCR Master Mix (2×), 1.2 µl KAPA RT Mix (50×), 1.2 µl 5′ primer (10 µM), 1.2 µl 3′ primer (10 µM), 17.4 µl $dH_2O$, and 9 µl of diluted RNA were prepared and 18 µl thereof were then transferred into three PCR strip tubes and served as technical replicates. Real-time qRT-PCRs were run in a Rotor-Gene Q real-time PCR cycler (QIAGEN, Hilden, Germany) using the following program: 5 min at 42°C (reverse transcription), 3 min at 95°C (initial denaturation and enzyme activation), 3 s at 95°C (denaturation), 20 s at 60°C (annealing, elongation, and fluorescence data acquisition), 40 cycles. The raw data were analyzed with the LinRegPCR program (*Ruijter et al., 2009*). The following oligonucleotide pairs were used for the specific amplification of DNA fragments, corresponding to the *RPL3*, *RPL4*, *RPL5*, *RPL10*, *RPL11*, *RPS2*, *RPS3*, *RPS6*, *ACT1*, and *UBC6* mRNAs or the yEGFP fusion protein encoding mRNAs, from the input cDNAs:

> RPL3-I-forward: 5′-ACTCCACCAGTTGTCGTTGTTGGT-3′
> RPL3-I-reverse: 5′-TGTTCAGCCCAGACGGTGGTC-3′ (amplicon size 86 base pairs [bp])
> RPL4-I-forward: 5′-ACCTCCGCTGAATCCTGGGGT-3′
> RPL4-I-reverse: 5′-ACCGGTACCACCACCACCAA-3′ (amplicon size 72 bp)
> RPL5-I-forward: 5′-TAGCTGCTGCCTACTCCCACGA-3′
> RPL5-I-reverse: 5′-GCAGCAGCCCAGTTGGTCAAA-3′ (amplicon size 70 bp)
> RPL10-I-forward: 5′-TGTCTTGTGCCGGTGCGGAT-3′
> RPL10-I-reverse: 5′-TGTCGACACGAGCGGCCAAA-3′ (amplicon size 84 bp)
> RPL11-I-forward: 5′-ACACTGTCAGAACTTTCGGT-3′
> RPL11-I-reverse: 5′-TTTCTTCAGCCTTTGGACCT-3′ (amplicon size 81 bp)
> RPS2-forward: 5′-AGGGATGGGTTCCAGTTACC-3′
> RPS2-reverse: 5′-TGGCAAAGAGTGCAAGAAGA-3′ (amplicon size 89 bp)
> RPS3-I-forward: 5′-GCTGCTTACGGTGTCGTCAGAT-3′
> RPS3-I-reverse: 5′-AGCCTTAGCTCTGGCAGCTCTT-3′ (amplicon size 96 bp)
> RPS6-forward: 5′-CAAGGCTCCAAAGATCCAAA-3′
> RPS6-reverse: 5′-TGAGCGTTTCTGACCTTCAA-3′ (amplicon size 87 bp)
> ACT1-forward: 5′-TCCAAGCCGTTTTGTCCTTG-3′
> ACT1-reverse: 5′-TGAGTAACACCATCACCGGA-3′ (amplicon size 76 bp)
> UBC6-forward: 5′-ACAAAGGCTGCGAAGGAAAA-3′
> UBC6-reverse: 5′-TGTTCAGCGCGTATTCTGTC-3′ (amplicon size 74 bp)
> yEGFP-II-forward: 5′-TCACTGGTGTTGTCCCAATT-3′
> yEGFP-II-reverse: 5′-ACCTTCACCGGAGACAGAAA-3′ (amplicon size 77 bp)

The log2 of the N0 values calculated by the LinRegPCR program were used for further calculations. First, for each biological sample the three technical replicate values obtained for the gene of interest were normalized against the *UBC6* values; to capture the maximal technical variation, the lowest *UBC6* value was subtracted from the highest value, the highest *UBC6* value from the lowest value, and the median *UBC6* value from the median value. Then the *UBC6*-normalized values were normalized across the entire experiment to the average of the reference sample(s) (either wild-type or, for mapping of the regulation-conferring element, Δcaf130 cells). All single values are shown as dots in the box and whisker plots, which were generated using the seaborn Python data visualization library (*Waskom, 2021*). In the case of the experiment addressing the requirement of ongoing translation for negative regulation (time course of galactose induction and cycloheximide treatment; *Figure 4— figure supplement 2*), the log2-transformed N0 values of the two different reporter mRNAs at each different time point were normalized to the respective value of the BI-containing reporter mRNA at time point 0. The data from five independent experiments, obtained with four different wild-type strains and in each case consisting of technical triplicates, were plotted using the default settings of the lineplot function of the seaborn Python data visualization library.

## Chromatin immunoprecipitation (ChIP)

ChIP experiments were essentially performed as previously described (*Zencir et al., 2020*). Briefly, wild-type and Δcaf130 mutant strains were grown in YPD medium at 30°C to an $OD_{600}$ of around 0.6. To inhibit TORC1, rapamycin was added to a final concentration of 200 ng/ml and cultures were incubated for an additional 20 min. Upon crosslinking with 1% formaldehyde for 10 min and quenching with 125 mM glycine for 5 min at 30°C, cells were harvested by centrifugation, washed once with 1 ml ice-cold water, and frozen in liquid nitrogen. Cells were resuspended in 400 µl ChIP lysis buffer (50 mM HEPES-Na pH 7.5, 150 mM NaCl, 1 mM EDTA, 1% NP-40, 0.1% sodium deoxycholate, and 1 mM PMSF) and cell extracts were obtained by glass bead lysis with a Precellys 24 homogenizer (Bertin Technologies, Montigny-le-Bretonneux, France) set at 5000 rpm using a 3 × 30 s lysis cycle with 30 s breaks in between at 4°C. Lysates were transferred to new tubes. Then, for complete extract recovery, 150 µl of lysis buffer was used to rinse the glass beads and combined with the already transferred lysate. Cell lysates were centrifuged at 4°C for 30 min at 13,500 rpm and 300 µl ChIP lysis buffer was added to the pellets. Chromatin was fragmented by sonication (6 × 15 s with power set to 50 and keeping samples between sonication cycles on ice) using a microtip-equipped SLPe sonifier (Branson Ultrasonics, Brookfield, USA). The sonicated samples were centrifuged at 7000 rpm for 15 min at 4°C, and the supernatants were transferred to a new tube. To specifically ChIP initiating RNA Pol II, the supernatants were incubated with 0.75 µl (at 1 µg/µl) of a polyclonal rabbit antibody recognizing the S5-phosphorylated heptapeptide repeats in the C-terminal domain of Rpo21/Rpb1 (ab5131; Abcam, Cambridge, UK) for 1 hr on a rotating wheel at 4°C. Meanwhile, magnetic beads (30 µl of slurry per ChIP) coupled to anti-rabbit IgG (Dynabeads M-280 Sheep Anti-Rabbit IgG; Invitrogen) were washed once with 1 ml ChIP lysis buffer and blocked for around 45 min by incubation with 1 ml ChIP lysis buffer containing 3% BSA. Upon removal of the blocking solution, samples were added to the blocked beads and incubated for 2 hr at 4°C. Then, beads were washed twice with 1 ml AT1 buffer (50 mM HEPES-Na pH 7.5, 150 mM NaCl, 1 mM EDTA, and 0.03% SDS), once with 1 ml AT2 buffer (50 mM HEPES-Na pH 7.5, 1 M NaCl, and 1 mM EDTA), once with 1 ml AT3 buffer (20 mM Tris-HCl pH 7.5, 250 mM LiCl, 1 mM EDTA, 0.5% NP-40, and 0.5% sodium deoxycholate), and twice with 1 ml TE buffer (10 mM Tris-HCl pH 8 and 1 mM EDTA pH 8). To elute immunoprecipitated chromatin fragments, beads were incubated for 10 min at 65°C in 95 µl TES buffer (10 mM Tris-HCl pH 8, 1 mM EDTA, and 1% SDS). Upon transfer of eluates to 1.5 ml safe-lock tubes, crosslinks were reversed by overnight incubation at 65°C in a thermoshaker set to 1200 rpm. Then, residual RNA was digested for 10 min at 37°C upon addition of 5 µl RNase A solution (R6148 (20 mg/ml RNase A; Sigma-Aldrich)). DNA fragments were purified using the GenElute PCR Clean-Up Kit (Sigma-Aldrich) and eluted in 50 µl of elution buffer (10 mM Tris-HCl pH 8.5).

The isolated DNA fragments were diluted 50-fold in $dH_2O$, and 9 µl of these dilutions were used to prepare the reaction mixes for the qPCR, which was performed and analyzed as described above, except that the KAPA RT Mix was not added to the reaction mixes and that the reverse transcription step was omitted. For normalization of RNA Pol II occupancy around the TSS of the assessed genes, occupancy around the TSS of *SNR52* was used as a reference. The following oligonucleotide pairs

were used for the specific amplification of DNA fragments in the proximity of the TSS of *RPL3*, *RPL4A*, *RPL4B*, *RPL30*, *RPL39*, *RPS20*, *ACT1*, *ADH1*, *UBC6*, and *SNR52*:

> RPL3-forward: 5′-TTCATTTCGGTTTTGTCATCTC-3′
> RPL3-reverse: 5′-TAAATGACCGTGACGTGGTG-3′ (amplicon size 95 bp)
> RPL4A-forward: 5′-TCACATTTCTTTTAGCCTCGCA-3′
> RPL4A-reverse: 5′-CGATGATTGTTCTTGGGATATTGC-3′ (amplicon size 79 bp)
> RPL4B-forward: 5′-TGAAGATCATGAATACGTTACACTACT-3′
> RPL4B-reverse: 5′-GTAAATGACTATGAATATGTAAGCGATTG-3′ (amplicon size 90 bp)
> RPL30-forward: 5′-CAGACCGGGAGTGTTTAAGAACC-3′
> RPL30-reverse: 5′-CCATGAAAGAATAAAGCGAAA-3′ (amplicon size 105 bp)
> RPL39-forward: 5′-TGAAAATTCGAAAAAGACAAGC-3′
> RPL39-reverse: 5′-TTGCTATTGATCAGTTCAGCATC-3′ (amplicon size 105 bp)
> RPS20-forward: 5′-TTGAAACTCCTACAAGAAAGCAAG-3′
> RPS20-reverse: 5′-TGTTGTTGTTCTTGTTCTTCAACC-3′ (amplicon size 98 bp)
> ACT1-forward: 5′-TGTGTAAAGCCGGTTTTGCC-3′
> ACT1-reverse: 5′-TGACCCATACCGACCATGAT-3′ (amplicon size 100 bp)
> ADH1-forward: 5′-ACCAAGCATACAATCAACTATCTCA-3′
> ADH1-reverse: 5′-CAACTTACCGTGGGATTCGT-3′ (amplicon size 88 bp)
> UBC6-forward: 5′-CTACAAAGCAGGCTCACAAGA-3′
> UBC6-reverse: 5′-GTTGGGGCGAGCAAGAATAT-3′ (amplicon size 86 bp)
> SNR52-forward: 5′-TGAATGACATTAGCGTGAACA-3′
> SNR52-reverse: 5′-TCAGAAGGAAGGCAACATAAG-3′ (amplicon size 82 bp)

## RNA-seq

Total RNA was extracted by the hot acid-phenol method (see above), and DNase-treated RNA samples were sent to the NGS platform of the University of Bern. The quality of the samples was assessed by their analysis on a Fragment Analyzer (Advanced Analytical Technologies Inc, Ankeny, USA). The libraries were prepared according to the TruSeq Stranded mRNA Sample Preparation Guide (Illumina). Briefly, polyA-containing mRNAs were purified, fragmented, reverse transcribed, and amplified to generate the libraries, which were subjected to high-throughput sequencing on a HiSeq 3000 instrument (Illumina).

In a preliminary step, the reads from multiple lanes were combined into single files. Quality control was performed with FastQC v0.11.7 (fastqc: https://www.bioinformatics.babraham.ac.uk/projects/fastqc/), revealing excellent quality reads for all samples; hence, no cleaning was applied. The yeast genome R64-1-1.90, downloaded from Ensembl (Saccharomyces_cerevisiae.R64-1-1.dna.toplevel.fa; *Yates et al., 2020*), was indexed for STAR v2.5.0b (*Dobin et al., 2013*). Then the reads from step 1 were remapped to genes for each sample with STAR using the annotation information from Ensembl (Saccharomyces_cerevisiae.R64-1-1.90.gtf). The final table of counts was obtained by merging the individual tables with Unix commands. Since several genes of interest have paralogs (e.g., *RPL4A* and *RPL4B*), we used the parameter "`--outFilterMultimapNmax` 2" to allow for two possible locations of each read.

The read counts (*Supplementary file 5*; for metadata, see *Supplementary file 6*) were then analyzed using the R library DESeq2, version 1.30.1 (*Love et al., 2014*). Since the mating types of the strains were not taken into consideration during the design of the experiment and can be imbalanced between the triplicate of each investigated genotype, a first analysis was conducted on all samples to determine the genes that were significantly changed (padj<0.05) due to the mating type of the strains (*MAT**a*** versus *MATα*). 26 genes were identified and removed for a second analysis where each mutant triplicate was compared to the wild-type triplicate. For details about the R script, see *Supplementary file 7*. The raw reads have been deposited at the ENA under the study accession number PRJEB45852.

## Preparation of total yeast protein extracts and Western analysis

Total yeast protein extracts were prepared as previously described (*Yaffe and Schatz, 1984*). Cultures were grown to an $OD_{600}$ of around 0.8, and protein extracts were prepared from an equivalent of 1 $OD_{600}$ of cells. Western blot analysis was carried out according to standard protocols. The following primary antibodies were used in this study: mouse monoclonal anti-GFP (1:2000; Roche), anti-HA

(clone 16B12, 1:3000; BioLegend, San Diego, USA), and anti-Rpl3 (1:5000; J. Warner, Albert Einstein College of Medicine, New York, USA); rabbit polyclonal anti-Adh1 (1:50,000; obtained from the laboratory of C. De Virgilio, University of Fribourg, Fribourg, Switzerland), anti-Rpl1 (1:5000; obtained from the laboratory of J. de la Cruz, University of Sevilla, Sevilla, Spain [*Petitjean et al., 1995*]), anti-Rpl4 (1:10,000; L. Lindahl, University of Baltimore, Baltimore, USA), anti-Rpl5 (1:15,000; S. R. Valentini, São Paulo State University, Araraquara, Brazil [*Zanelli et al., 2006*]), anti-Rpl11 (1:5000; L. Lindahl), anti-Rpl35 (1:5000; M. Seedorf, ZMBH, University of Heidelberg, Heidelberg, Germany [*Frey et al., 2001*]), anti-Rpp0 (1:5000; S. R. Valentini [*Zanelli et al., 2006*]), anti-Rps3 (1:20,000; M. Seedorf [*Frey et al., 2001*]), anti-Rps9 (1:10,000; L. Lindahl), and anti-Tsr2/Rps26 (1:3000; V. G. Panse, University of Zürich, Zürich, Switzerland [*Schütz et al., 2014*]). Secondary goat anti-mouse or anti-rabbit horseradish peroxidase-conjugated antibodies (Bio-Rad, Hercules, USA) were used at a dilution of 1:10,000. For detection of TAP-tagged proteins, the peroxidase anti-peroxidase (PAP) Soluble Complex antibody produced in rabbit (Sigma-Aldrich) was used at a dilution of 1:20,000. Immobilized protein-antibody complexes were visualized by using enhanced chemiluminescence detection kits (WesternBright Quantum and Sirius; Advansta, San Jose, USA) and an Azure c500 imaging system (Azure Biosystems, Dublin, USA). Images were processed with ImageJ (*Schneider et al., 2012*).

## GFP-Trap co-immunoprecipitation

Yeast cells were grown at 30°C in 200 ml of YPD medium to an $OD_{600}$ of around 0.8. Cells were washed in ice-cold $dH_2O$ and resuspended in 400 µl of lysis buffer (50 mM Tris-HCl pH 7.5, 100 mM NaCl, 1.5 mM $MgCl_2$, 5% glycerol, 0.1% NP-40, 1 mM PMSF, and SIGMA*FAST* EDTA-free protease inhibitor cocktail [Sigma-Aldrich]). Cell extracts were obtained by glass bead lysis with a Precellys 24 homogenizer (Bertin Technologies) set at 5000 rpm using a 3 × 30 s lysis cycle with 30 s breaks in between at 4°C. Cell lysates were clarified by centrifugation at 4°C for 10 min at 13,500 rpm. GFP-Trap Magnetic Agarose beads (Chromotek, Planegg-Martinsried, Germany) were blocked by incubation with wild-type yeast cell lysates (1 $A_{260}$ unit per µl of bead slurry) for 1 hr. For affinity purification, 20 µl of blocked GFP-Trap bead slurry were incubated with 100 $A_{260}$ units of cell lysate in a total volume of 650 µl for 2 hr at 4°C. Beads were then washed nine times with 600 µl lysis buffer and finally boiled for 5 min in 50 µl of 3× SDS sample buffer to elute the bound proteins. For Western analysis, 0.1 $A_{260}$ units of cell lysate (input) and one-fifth of the affinity purification (IP) were separated on Bolt 4–12% Bis-Tris Plus 15-well gels (Invitrogen), run in Bolt 1× MOPS SDS running buffer (Novex, Carlsbad, USA), and subsequently transferred onto Amersham Protran nitrocellulose membranes (Cytiva, Marlborough, USA), which were incubated with anti-GFP antibodies and the PAP Soluble Complex antibody to detect the GFP-tagged bait and the TAP-tagged prey proteins, respectively.

## Y2H interaction analysis

For Y2H-interaction assays, plasmids expressing bait proteins, fused to the Gal4 DNA-binding domain (G4BD), and prey proteins, fused to the Gal4 activation domain (G4AD), were co-transformed into reporter strain PJ69-4A. Y2H interactions were documented by spotting representative transformants in 10-fold serial dilution steps onto SC-Leu-Trp, SC-Leu-Trp-His (*HIS3* reporter), and SC-Leu-Trp-Ade (*ADE2* reporter) plates, which were incubated for 3 days at 30°C. Growth on SC-Leu-Trp-His plates is indicative of a weak/moderate interaction, whereas only relatively strong interactions permit growth on SC-Leu-Trp-Ade plates.

## Protein aggregation assay

Yeast cells expressing N-terminally 2xHA-tagged Tom1 under the transcriptional control of the *GAL1* promoter from the genomic locus and additionally harboring deletions of *CAF130* (*Δcaf130*) or *CAL4* (*Δcal4*) were grown at 30°C in 50 ml of YPGal medium and then shifted for up to 24 hr to YPD medium (*Figure 7D*). Wild-type or *Δtom1* cells, transformed with plasmids expressing different C-terminally 2xHA-tagged variants of Rpl3 or Rpl4a under the control of the *GAL1-10* promoter, were grown at 30°C in 50 ml of SRaf-Leu (raffinose) medium to an $OD_{600}$ of around 0.4, and expression of the Rpl3 and Rpl4a variants was induced for 4 hr with 2% galactose (*Figure 7—figure supplement 2A and B*). Cells were harvested and resuspended in 400 µl of lysis buffer (20 mM Na-phosphate pH 6.8, 1 mM EDTA, 0.1% Tween-20, 1 mM DTT [freshly added], and 1 mM PMSF [freshly added]). Cell extracts were obtained by glass bead lysis with a Precellys 24 homogenizer (Bertin Technologies) set at 5000 rpm

using a 3 × 30 s lysis cycle with 30 s breaks in between at 4°C. Cell lysates were clarified by centrifugation at 4°C for 20 min at 2500 rpm. Aliquots of 0.5 $A_{260}$ units of clarified total extracts were diluted into a final volume of 100 μl of 3× SDS sample buffer (0.005 $A_{260}$ units per μl). To pellet aggregated proteins, 10 $A_{260}$ units of clarified cell extracts were centrifuged at 4°C for 20 min at 13,500 rpm. The pellets were then washed three times by resuspension in 900 μl of wash buffer (20 mM Na-phosphate pH 6.8, 500 mM NaCl, and 2% NP-40). The final insoluble pellets were resuspended and boiled in 100 μl of 3× SDS sample buffer (corresponding to 0.1 $A_{260}$ units per μl of clarified total extract input). For Coomassie staining and Western analysis, 5 μl of total extract (0.025 $A_{260}$ units) and insoluble pellet (0.5 $A_{260}$ units) were separated on NuPAGE 4–12% Bis-Tris 26-well Midi gels (Invitrogen), run in NuPAGE 1× MES SDS running buffer (Novex).

To determine the identity of the aggregated proteins, the insoluble pellet fractions (2 $A_{260}$ units) were separated on NuPAGE 4–12% Bis-Tris 15-well gels (Invitrogen), run in NuPAGE 1× MOPS SDS running buffer (Novex), and subsequently stained with Brilliant Blue G Colloidal Coomassie (Sigma-Aldrich). Proteins contained in Coomassie-stained bands were digested in-gel with trypsin and identified, upon mass spectrometric analysis of the obtained peptides, using the MaxQuant software package (*Tyanova et al., 2016*).

## Fluorescence microscopy

Wild-type or *Δtom1* cells were transformed with plasmids expressing, under the control of the *GAL1-10* promoter, the different Rpl3 and Rpl4 variants fused to a C-terminal mNeonGreen (*Shaner et al., 2013*; Allele Biotechnology, San Diego, USA), codon-optimized for expression in yeast and, hence, referred to as yeast-optimized mNeonGreen (yOmNG). Transformed cells were grown at 30°C in 20 ml of SRaf-Leu to an $OD_{600}$ of around 0.25 and expression of the yOmNG-tagged Rpl3 and Rpl4a variants was induced for 4 hr with 2% galactose. Live yeast cells were imaged by fluorescence microscopy using a VisiScope CSU-W1 spinning disk confocal microscope (Visitron Systems GmbH, Puchheim, Germany). Nop58-yEmCherry, expressed from plasmid under the control of the cognate promoter, was used as a nucleolar marker. The ImageJ software was used to process the images. Cells displaying one of the three types of observed localizations (cytoplasmic, nucleolar accumulation, and nuclear aggregation) of the mNeonGreen-tagged Rpl3 or Rpl4 variants were counted manually on z-projected maximum intensity images, while the shown examples (*Figure 7A*, *Figure 7—figure supplement 2C*) correspond to a selected slice derived from the full z-stacked image.

## Sequence alignments and analysis of 3D structures

Multiple sequence alignments of orthologous proteins were generated in the ClustalW output format with T-Coffee using the default settings of the EBI website interface (*Notredame et al., 2000*). Analysis and image preparation of three-dimensional structures, downloaded from the PDB archive, were carried out with the PyMOL (PyMOL Molecular Graphics System; http://pymol.org/) software. The coordinates of the following structures were used: *S. cerevisiae* 80S ribosome (PDB 4V88; *Ben-Shem et al., 2011*) and *C. thermophilum* Acl4-Rpl4 complex (PDB 5TQB; *Huber and Hoelz, 2017*).

## Acknowledgements

We thank J de la Cruz, C De Virgilio, L Lindahl, VG Panse, M Seedorf, SR Valentini, and J Warner for the kind gift of antibodies, B Albert and JA Ripperger for valuable advice concerning the ChIP experiment, and J de la Cruz and B Pertschy for fruitful discussions. We gratefully acknowledge B Egger and F Meyenhofer of the Bioimage Core Facility of the University of Fribourg for their support and assistance. This work was supported by the Swiss National Science Foundation (31003A_156764, 31003A_175547, and 310030_204801 to DK), the Novartis Foundation for Medical-Biological Research (14C154 to DK), and the Canton of Fribourg.

# Additional information

## Funding

| Funder | Grant reference number | Author |
|---|---|---|
| Schweizerischer Nationalfonds zur Förderung der Wissenschaftlichen Forschung | 31003A_156764 | Dieter Kressler |
| Schweizerischer Nationalfonds zur Förderung der Wissenschaftlichen Forschung | 31003A_175547 | Dieter Kressler |
| Schweizerischer Nationalfonds zur Förderung der Wissenschaftlichen Forschung | 310030_204801 | Dieter Kressler |
| Novartis Stiftung für Medizinisch-Biologische Forschung | 14C154 | Dieter Kressler |

The funders had no role in study design, data collection and interpretation, or the decision to submit the work for publication.

## Author contributions

Benjamin Pillet, Conceptualization, Formal analysis, Funding acquisition, Investigation, Methodology, Project administration, Resources, Software, Supervision, Validation, Visualization, Writing - original draft, Writing – review and editing; Alfonso Méndez-Godoy, Guillaume Murat, Sébastien Favre, Investigation; Michael Stumpe, Formal analysis, Investigation, Software; Laurent Falquet, Data curation, Formal analysis, Investigation, Methodology, Resources, Software, Writing – review and editing; Dieter Kressler, Conceptualization, Formal analysis, Funding acquisition, Investigation, Methodology, Project administration, Resources, Supervision, Validation, Visualization, Writing - original draft, Writing – review and editing

## Author ORCIDs

Benjamin Pillet http://orcid.org/0000-0002-7313-4304
Michael Stumpe http://orcid.org/0000-0002-9443-9326
Dieter Kressler http://orcid.org/0000-0003-4855-3563

## Decision letter and Author response

Decision letter https://doi.org/10.7554/eLife.74255.sa1
Author response https://doi.org/10.7554/eLife.74255.sa2

# Additional files

## Supplementary files

- Transparent reporting form
- Supplementary file 1. Yeast strains used in this study.
- Supplementary file 2. Plasmids used in this study.
- Supplementary file 3. Identified Δacl4 and Δacl4/Δrpl4a suppressor mutations.
- Supplementary file 4. Results of RNA-Seq.
- Supplementary file 5. Raw read counts of RNA-Seq.
- Supplementary file 6. Metadata of RNA-Seq.
- Supplementary file 7. R script for analysis of RNA-Seq raw read counts.
- Supplementary file 8. Identified proteins in aggregates of Δcaf130/PGAL-2xHA-TOM1 cells.

## Data availability

The raw reads of the sequenced Δacl4 and Δacl4/Δrpl4a suppressor genomes and of the differential gene expression analysis (RNA-Seq) have been deposited at the European Nucleotide Archive under the study accession number PRJEB45852.

The following dataset was generated:

| Author(s) | Year | Dataset title | Dataset URL | Database and Identifier |
|---|---|---|---|---|
| Pillet B, Méndez-Godoy A, Murat G, Favre S, Stumpe M, Falquet L, Kressler D | 2021 | Dedicated chaperones coordinate co-translational regulation of ribosomal protein production with ribosome assembly to preserve proteostasis | https://www.ebi.ac.uk/ena/browser/view/PRJEB45852 | European Nucleotide Archive (ENA), PRJEB45852 |

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
