## [Editor Report]

The work describes an exciting new mechanism for how r-proteins are produced in the correct abundances. Specifically, the authors find that the co-translational recognition of Rpl3/4 by their respective chaperones maintains the stability of RPL3 and RPL4 mRNAs. This mechanism is reminiscent of mechanisms of translation regulation in yeast mitochondria where oxidative phosphorylation complex assembly factors similarly regulate RNA stability and translation to ensure subunits are not produced in excess.

---

## [Decision Letter]

**Decision letter after peer review:**

Thank you for submitting your article "Dedicated chaperones coordinate co-translational regulation of ribosomal protein production with ribosome assembly to preserve proteostasis" for consideration by *eLife*. Your article has been reviewed by 3 peer reviewers, and the evaluation has been overseen by a Reviewing Editor and David Ron as the Senior Editor. The following individuals involved in review of your submission have agreed to reveal their identity: Eugene Valkov (Reviewer #2); David Shore (Reviewer #3); David Tollervey (Reviewer #4).

Essential revisions:

1) The authors should consider how they might shorten the text throughout without sacrificing any information. There are many spots where deletions or rephrasing could actually improve clarity. Additionally, the authors could either delete the penultimate Results section ("Possible conservation…") completely or reduce it to a short paragraph since it contains no new functional data. Finally, some of the analysis could be tightened up to shorten the manuscript as well.

2) The RNA-seq experiment in Figure 1 does not distinguish between effects on transcription and mRNA stability. It is important to measure transcription more directly in these strains, for example by RNA Pol2 ChIP-seq, NET-seq or TT-seq. Or RNA Pol2 ChIP-qPCR at RPL3/4 and perhaps at one or two RP genes that appear to be highly regulated at initiation (category I or II) for comparison. This should allow the authors to determine the extent to which regulation of all r-protein gene expression in the strains they test is either at the level of transcription initiation or mRNA stability (production versus degradation). This is particularly relevant given that recent studies have shown that RPL3/4 genes are subjected to minimal regulation at the transcriptional level following stress compared to most other r-protein genes. In sum, transcription should be more directly analyzed in the key mutant strains and in wild-type cells under a few stress conditions (e.g. heat shock, TORC1 inhibition, nutrient deprivation, ribosome assembly defects, etc.).

3) The authors need to address directly the relationship between the system they describe here and the observations in Buschauer et al. Science, 2020 (and references therein) regarding the involvement of Ccr4-Not in monitoring codon optimality at translating ribosomes. This could be done through a clearer discussion of these points but would be better addressed by testing the role of the Not5 N-terminal domain in RPL3/4 mRNA regulation and, perhaps using data in Buschauer et al., determine whether ribosome stalling occurs around the regulatory regions they identify.

*Reviewer #3 (Recommendations for the authors):*

One of my major concerns is that the manuscript is too long, with the writing often verbose and very difficult to follow. I will make some specific suggestions below but would urge the authors to consider how they might shorten the text throughout without sacrificing any information. There are many spots where deletions or rephrasing could actually improve clarity. I would also suggest that the authors either delete the penultimate Results section ("Possible conservation…") completely or reduce it to a short paragraph since it contains no new functional data.

Regarding the experiments themselves, I have the following major criticisms/suggestions:

(1) The RNA-seq experiment in Figure 1 does not distinguish between effects on transcription and mRNA stability. It would thus be useful to measure transcription more directly in these strains, either by NET-seq or RNA Pol2 ChIP-seq. This should allow the authors to determine the extent to which regulation of all r-protein gene expression in the strains they test is either at the level of transcription initiation or mRNA stability (production versus degradation). This is particularly relevant given that recent studies have shown that RPL3/4 genes are subjected to minimal regulation at the transcriptional level following stress compared to most other r-protein genes (see below).

(2) The authors should consider auxin-induced degron (AID) experiments to address the role of Caf1, Ccr4, and Not4, using RNA-seq or RT-PCR to measure the RPL3/4 mRNA levels following auxin addition, with appropriate controls.

(3) To solidify the claim that Caf130 bridges the interaction between Btt1 and Not1, the pull-down experiments could be performed in a caf130Δ strain.

(4) On pg. 13 line 29, I would suggest that the authors delete "nascent" and rephrase the title of this section since it implies that regulation occurs exclusively at the ribosome, which is not actually demonstrated here. I would agree that this is very likely to be the case, but I would suggest that the authors make clear that they don't know for sure, unless, of course, they have a strong argument that this has to be true.

(5) The argument (pg. 15 line 18) that the regulatory signal is on the Rpl4 protein and not its mRNA is not very strong, in my opinion. It might be interesting to see if cycloheximide treatment blocks the regulation, perhaps following acute Acl4 depletion by AID.

(6) The authors need to address directly the relationship between the system they describe here and the observations in Buschauer et al. Science, 2020 (and references therein) regarding the involvement of Ccr4-Not in monitoring codon optimality at translating ribosomes. They should test the role of the Not5 N-terminal domain in RPL3/4 mRNA regulation and, perhaps using data in Buschauer et al., determine whether ribosome stalling occurs around the regulatory regions they identify.

(7) Related to the point above, the authors could significantly improve their study by either testing the role of NAC subunits in codon optimality monitoring by Ccr4-Not (related to Buschauer et al. Figure 5) or searching for the putative targets of the Rpl3/4 regulatory regions (which they speculate could be Btt1). Both would be even better. A key point that they need to address is whether or not the RPL3/4 regulatory system they describe here is in fact controlled by codon optimality, which might itself be exacerbated by stress, rather than through a specific protein-protein interaction involving the r-protein regulatory regions and/or interactions with the chaperones. It is less clear to me how the latter might be regulated by stress.

(8) The sentence pg. 26 line 3 doesn't make sense to me ("even though") and I think, as mentioned above, that the authors miss an important point here. Zencir et al. show that RPL3/4 and the other category III genes are much less down-regulated following different forms of stress than are the majority of genes (category I and II), and suggest that this may be due to the fact that they are coupled to specialized chaperone proteins. The authors' findings suggest that RPL3/4 mRNAs are indeed strongly regulated upon stress (though they don't actually examine stress conditions in this study!), but at a post-transcriptional level, namely mRNA stability. This (potential) novel finding should be directly tested by examing RPL3/4 mRNA stability under various stress conditions (e.g. heat shock, TORC1 inhibition, nutrient deprivation, ribosome assembly defects, etc.).

(9) Related to the above, the authors never test experimentally the notion that Ccr4-Not regulation of RPL3/4 mRNAs is part of the ribosome assembly stress response described by Albert et al. (2019) and Tye et al. (2019). The authors focus primarily on the relationship between chaperone levels and RPL3/4 mRNA levels without really addressing the possible physiological role of Ccr4-Not mRNA regulation with respect to Rpl3/4. A key feature of this response is that it is abrogated by cycloheximide treatment, differentiating it from some other stress responses. Dampening of RPL3/4 mRNA regulation by cycloheximide would also provide support for the notion that the polypeptide and not the RNA of the regulatory region is responsible for regulation.

Additional comments and suggestions:

(1) In the Abstract, pg. 2 line 8 "prevents" should probably be changed to "reduces", since they show that even in conditions of normal chaperone levels there is still degradation.

(2) Pg. 6, lines 13-16: simplify to something like "To gain further insight into this observation we isolated a large number of acl4Δ and acl4Δ/rpl4Δ suppressors and identified the causative mutations by whole-genome sequencing".

(3) Not clear how causative mutations were identified without pooling and sequencing of sup+ and sup- isolates from a single backcross (pooled linkage analysis). Were there really no other polymorphisms in the suppressor strains.

(4) It would be useful to show a cartoon of the NAC and Ccr4-Not complexes in Figure 1 to help the reader to follow the subsequent analysis.

(5) In section "RPL4 mRNA levels are increased…" (pgs. 7-8) it would be interesting to know, as pointed out above, which of these effects are at the level of mRNA production versus degradation or stabilization. The effect of cal4Δ is hardly small, with both up and down-regulated genes. Why is that?

(6) The sentences on pg. 25, lines 11-14 and 27-31, are awkward and need to be rewritten for clarity ("only present in insufficient abundance"…?; too many subordinate clauses in the latter sentence).

*Reviewer #4 (Recommendations for the authors):*

1. Figure 1: In the tree plots – indicating the zero with a dashed line might be helpful. This type of data is often presented as a volcano plot – with significance (adjusted P values) rather than counts on the axis. I quite like this version, however, in the absence of P-values for named gene classes in the graph, the authors should consider stating these for the most important changes, especially when marginal (eg Rpl3 and Rpl4 in ∆btt1, ∆egd1).

2. P12, P13; Figure 3: "Btt1 appears to exclusively interact with either Caf130 or Egd2, but not simultaneously with both, since neither Caf130 nor Egd2 could co-purify each other…." Could this be an efficiency issue? I did not see any indication of the pairwise coprecipitation efficiency but according to the legend ~200 fold more precipitate was loaded relative to input, so the stoichiometry is unclear. If – for example – 10% of each protein is in complex with Btt1, only 1% would coprecipitate.

3. Figure 7S4: The general approach of using MS to assess the proteins present in the pellet fraction is good. However, the version used, involving the cutting of visible gel bands is surprising. By definition, most proteins detected during depletion are absent from the 0 h sample, since the corresponding regions were not analysed. A labelling based method, eg SILAC or iCAT, allowing quantitation of relative recovery for all proteins might have been better. The data presented are, however, sufficient to support the major conclusions, that protein aggregation is increased and multiple proteins are recruited.

4. Figure 8: What does the exclamation mark indicate?

5. In the PDF, commas separating thousands in numbers are inverted.

6. A great deal of data are presented, but even allowing for this the text is very long, with frequent repetitions in the text and again in the figure legends and Discussion. At over 8 pages, the Discussion seems particularly long. A result is that only afficionados may make it thought to the end.

---

## [Author Response]

Essential revisions:1) The authors should consider how they might shorten the text throughout without sacrificing any information. There are many spots where deletions or rephrasing could actually improve clarity. Additionally, the authors could either delete the penultimate Results section ("Possible conservation…") completely or reduce it to a short paragraph since it contains no new functional data. Finally, some of the analysis could be tightened up to shorten the manuscript as well.

As recommended, we did our best to shorten and improve the clarity of the text. Especially, we have tried to remove all unnecessary repetitions of conclusions, which are later covered in the Discussion, in the Results section and of technical descriptions that are indicated on several occasions. In agreement with the valuable specific suggestion, we have deleted the entire Discussion paragraph “Possible conservation of the regulatory process”. Even though we discuss therein for the first time, also based on structural alignments of only recently available structure predictions, pertinent commonalities and differences between the yeast and human proteins associating with Not’1 N-terminal domain, we agree that this extensive analysis is not particularly relevant for the functional understanding of the uncovered regulatory mechanism.

2) The RNA-seq experiment in Figure 1 does not distinguish between effects on transcription and mRNA stability. It is important to measure transcription more directly in these strains, for example by RNA Pol2 ChIP-seq, NET-seq or TT-seq. Or RNA Pol2 ChIP-qPCR at RPL3/4 and perhaps at one or two RP genes that appear to be highly regulated at initiation (category I or II) for comparison. This should allow the authors to determine the extent to which regulation of all r-protein gene expression in the strains they test is either at the level of transcription initiation or mRNA stability (production versus degradation). This is particularly relevant given that recent studies have shown that RPL3/4 genes are subjected to minimal regulation at the transcriptional level following stress compared to most other r-protein genes. In sum, transcription should be more directly analyzed in the key mutant strains and in wild-type cells under a few stress conditions (e.g. heat shock, TORC1 inhibition, nutrient deprivation, ribosome assembly defects, etc.).

To measure possible effects on transcription, we have performed the suggested ChIP-qPCR experiments with wild-type and *∆caf130* mutant cells. To specifically ChIP initiating RNA Pol II, we employed, as commonly done (Albert et al. *eLife* 2019, PMID: 31124783; Zencir et al. Nucleic Acids Res. 2020, PMID: 33084907), an antibody (Abcam ab5131) recognizing the S5-phosphorylated heptapeptide repeats in the C-terminal domain of Rpo21/Rpb1. Then, we assessed occupancy of RNA Pol II around the transcription start site (TSS) of several RPGs, containing either category I (*RPL30* and *RPL39*), II (*RPS20*), or III (*RPL3*, *RPL4A*, and *RPL4B*) promoters, by qPCR, using occupancy around the TSS of *SNR52* as a reference for normalization. This analysis revealed that absence of Caf130 does not lead to an increased association of initiating RNA Pol II on the *RPL3* and *RPL4A/B* promoters (see new Figure 1—figure supplement 2A), strongly supporting the notion that the observed increase in *RPL3* and *RPL4* mRNA levels is due their increased stability when the uncovered regulatory mechanism has been inactivated. Moreover, we have also tested the effects of TORC1 inhibition by rapamycin treatment, which resulted in a similarly reduced transcription of all tested RPGs both in wild-type and *∆caf130* cells (Figure 1—figure supplement 2A). Nevertheless, the *RPL3* and *RPL4* mRNA levels remained around two-fold higher in cells lacking Caf130, suggesting that the *RPL3* and *RPL4* mRNAs, even when present at lower abundance, are still subjected to negative regulation under these conditions in wild-type cells (Figure 1—figure supplement 2B). The new text describing these findings can be found on page 7 (starting on line 34) and page 8 (ending on line 8) of the revised manuscript.

The finding that transcription initiation is not increased at the *RPL3* and *RPL4A/B* promoters in the absence of Caf130 was not unexpected as we had already shown in the submitted manuscript that inactivation of the regulatory machinery also leads to increased mRNA levels when *RPL3* and *RPL4A*, or the minimal regulation-conferring regions thereof, are transcribed from the *ADH1* promoter. Moreover, the finding that one single (leading to the W109C substitution in Rpl4a) or only a few (leading to the F16A/L17A substitutions in Rpl3) nucleotide change(s) within the coding sequence of the minimal regulation-conferring region increases the mRNA levels of these reporter constructs in wild-type cells provides additional strong evidence that regulation does not occur at the transcriptional stage.

3) The authors need to address directly the relationship between the system they describe here and the observations in Buschauer et al. Science, 2020 (and references therein) regarding the involvement of Ccr4-Not in monitoring codon optimality at translating ribosomes. This could be done through a clearer discussion of these points but would be better addressed by testing the role of the Not5 N-terminal domain in RPL3/4 mRNA regulation and, perhaps using data in Buschauer et al., determine whether ribosome stalling occurs around the regulatory regions they identify.

In the submitted manuscript, we had already described in the Results section that absence of Not5’s N-terminal domain (Figure 2—figure supplement 1I), which does not result in any observable growth defect (Figure 2—figure supplement 1H), as well as deletion of Rps25 (Figure 2—figure supplement 1J) do not suppress the growth defect of *∆acl4* cells (see page 11, lines 8-17). Moreover, we had mentioned in the Discussion that accommodation of the Not5-NTD in the ribosomal E-site of mRNAs displaying low A-site codon optimality represents one of the mechanisms of Ccr4-Not complex recruitment to a substrate mRNA (see page 27, lines 29-31).

To further corroborate that the mechanism reported by Buschauer et al. is not involved, we have, as suggested, assessed the effect of the absence of Not5’s N-terminal domain on *RPL3* and *RPL4* mRNA levels. As expected from the genetic data, expression of the Not5 variant lacking the N-terminal domain in *∆not5* cells does not lead to an increase in *RPL3* or *RPL4* mRNA levels (see new Figure 2—figure supplement 1J in the revised manuscript). Moreover, we have also introduced this additional information in the text (see page 11, lines 16-21 of the revised manuscript).

As previously shown by the Coller laboratory (Presnyak et al. Cell 2015, PMID: 25768907), most RPG mRNAs display an optimal codon content of more than 80%. Moreover, a stretch of several consecutive rare codons appears to be required to cause a dramatic decrease in mRNA stability, as revealed by the insertion of ten non-optimal codons into otherwise stable mRNA reporter constructs (see Hu et al. Nature 2009, PMID: 19701183; Sweet et al. PLoS Biol. 2012, PMID: 22719226). As suggested, we have assessed, according to the data provided in Figure 5C of the Buschauer et al. publication, the occurrence and distribution of Not4-enriched A-site codons and non-optimal codons, respectively, within the minimal regulation-conferring regions of *RPL3* (codons encoding residues 1-52) and *RPL4A* (codons encoding residues 78-139) (see ). (Author response image 1)

**Author response image 1. sa2fig1:** 

This analysis revealed that these *RPL3* and *RPL4A* regions contain 86.54% and 93.55% Not4-IP non-enriched A-site codons, respectively, and exhibit an optimal codon content of 80.77% and 88.71%, respectively. Importantly, there are not any evident stretches of either Not4-enriched A-site codons or non-optimal codons. In this context it is worth mentioning that the absence of Not5’s N-terminal domain does not increase the half-life of a *HIS3* reporter mRNA displaying 50% codon optimality (see Figure 5E in the Buschauer et al. publication); thus, making its involvement in targeting these *RPL3* and *RPL4* mRNA regions, which display an even higher codon optimality, to degradation highly unlikely.As our data clearly rule out an involvement of Not5’s N-terminal domain, we do not believe that it is necessary to further discuss, as proposed, the mechanism described by Buschauer et al., whose physiological relevance (i.e. the effect on specific mRNAs) has not been investigated in that study, in the context of our findings.

Reviewer #3 (Recommendations for the authors):One of my major concerns is that the manuscript is too long, with the writing often verbose and very difficult to follow. I will make some specific suggestions below but would urge the authors to consider how they might shorten the text throughout without sacrificing any information. There are many spots where deletions or rephrasing could actually improve clarity. I would also suggest that the authors either delete the penultimate Results section ("Possible conservation…") completely or reduce it to a short paragraph since it contains no new functional data.

See our answer to Essential revision point 1.

Regarding the experiments themselves, I have the following major criticisms/suggestions:(1) The RNA-seq experiment in Figure 1 does not distinguish between effects on transcription and mRNA stability. It would thus be useful to measure transcription more directly in these strains, either by NET-seq or RNA Pol2 ChIP-seq. This should allow the authors to determine the extent to which regulation of all r-protein gene expression in the strains they test is either at the level of transcription initiation or mRNA stability (production versus degradation). This is particularly relevant given that recent studies have shown that RPL3/4 genes are subjected to minimal regulation at the transcriptional level following stress compared to most other r-protein genes (see below).

See our answer to Essential revision point 2.

(2) The authors should consider auxin-induced degron (AID) experiments to address the role of Caf1, Ccr4, and Not4, using RNA-seq or RT-PCR to measure the RPL3/4 mRNA levels following auxin addition, with appropriate controls.

We have planned to thoroughly address the plausible involvement of the Caf1-Ccr4 deadenylase module in initiating degradation of the regulated *RPL3* and *RPL4* mRNAs as a follow-up to this first manuscript. However, the outcome and interpretation of these experiments may be hampered by the anticipated pleiotropic effects entailed by depletion of Caf1, Ccr4, and Not4 on general cytoplasmic mRNA decay and/or maintenance of proteostasis. In this sense, it may indeed be helpful to consider using, as proposed, a more rapid depletion system than the conventional one based on glucose-mediated repression of genes that are under the transcriptional control of the *GAL1-10* promoter. As auxin has recently been shown to inhibit TORC1 (Nicastro et al. PLoS Genet. 2021, PMID: 33690632), whose activity is required for efficient transcription of all RPGs, experiments using the auxin-induced degron (AID) depletion system need to be properly controlled and may, in the worst case, turn out to be inept for detecting around two-fold changes in *RPL3* and *RPL4* mRNA levels. As an alternative to assessing *RPL3* and *RPL4* mRNA levels upon depletion of these Ccr4-Not components, we have also planned to do this in *∆not1* cells co-expressing the major Not1 isoform (Not1.163C), lacking the N-terminal domain and sustaining the essential function of the Ccr4-Not complex, and the minor, full-length Not1 isoform (Not1.M163A) additionally containing deletions or point mutations that specifically abrogate Caf1-Ccr4 or Not4 binding. This setting would allow to also assess, besides mRNA levels, the capacity of these Not1 variants to complement *in trans* suppression of the *∆acl4* growth defect or synthetic lethality with the *∆tom1* null allele.

(3) To solidify the claim that Caf130 bridges the interaction between Btt1 and Not1, the pull-down experiments could be performed in a caf130Δ strain.

In our opinion, the shown Y2H and pull-down experiments collectively provide pretty solid evidence that Caf130 bridges the interaction between Btt1 and Not1 (see Figure 3A-H and Figure 3—figure supplements 1,2). Moreover, the proposed experiment has already been reported in a previous study and the result supports our conclusion (see Figure 7 in Cui et al. Mol. Genet. Genomics 2008, PMID: 18214544). We have provided this information and referred to the above article in the submitted manuscript (page 11, lines 26-31): “… and the finding that Btt1 is associated with the Ccr4-Not complex in a Caf130-dependent manner …”.

(4) On pg. 13 line 29, I would suggest that the authors delete "nascent" and rephrase the title of this section since it implies that regulation occurs exclusively at the ribosome, which is not actually demonstrated here. I would agree that this is very likely to be the case, but I would suggest that the authors make clear that they don't know for sure, unless, of course, they have a strong argument that this has to be true.

As suggested, we have removed “nascent” from this section title.

(5) The argument (pg. 15 line 18) that the regulatory signal is on the Rpl4 protein and not its mRNA is not very strong, in my opinion. It might be interesting to see if cycloheximide treatment blocks the regulation, perhaps following acute Acl4 depletion by AID.

While we believe that the entity of our data provide rather strong evidence for the claim that the regulatory signal is part of the Rpl3 or Rpl4 protein (see below), we agree that it might be premature to draw these conclusions at this stage of the manuscript. We have therefore omitted this sentence (page 15, lines 15-18 of the submitted manuscript) in the revised manuscript.

According to our opinion, several lines of evidence provide convincing support for the argument that the regulatory signal is harbored by the Rpl3 and Rpl4 protein: (1) A single amino acid exchange (W109C, one nucleotide substitution) within full-length Rpl4a confers *∆acl4* suppression and up-regulates levels of the *rpl4a.W109C*-yEGFP fusion mRNA (Figure 4D); (2) The W109C substitution within the minimal regulation-conferring segment (encompassing amino acids 78-139 of Rpl4a) is sufficient to reduce, even when Acl4 binding is disabled by the BI mutations, negative regulation of the encoding mRNA (Figure 4F); (3) Overexpression of Acl4, which we had previously shown to capture Rpl4 in a co-translational manner (Pillet et al. PLoS Genetics 2015) and whose binding site overlaps with the identified key residues mediating regulation (W109 and possibly also residues W106, R107, and K108 that are substituted alongside W109 with alanines in the BIV mutant) (see Acl4-Rpl4 co-structure in Huber and Hoelz Nat. Commun. 2017, PMID: 28148929), results in increased *RPL4* mRNA levels (Figure 6); (4) Similar principles apply to the regulation of RPL3 mRNA levels. For example, two amino acid changes (F16A/L17A, six nucleotide substitutions), which are immediately following the minimal Rrb1 binding site (see Pausch et al. Nat. Commun. 2015 and Figure 5—figure supplement 1), reduce negative regulation of the mRNA reporter encoding residues 1-52 of Rpl3 (see Figure 5C). Overexpression of Rrb1 or presence of the H3E mutation, which abolishes Rrb1 binding, increases or, respectively, decreases the levels of the *RPL3* mRNA (Figure 6 and Figure 5A).

While we are absolutely open for alternative models and have also explicitly stated in the Discussion that “Clearly, future experiments will be required to challenge the presented model and unveil the exact nature of the uncovered regulatory mechanism.” (see page 29, lines 11-12 of the submitted manuscript and page 28, lines 34-36 of the revised manuscript), the proposed co-translational regulation of *RPL3* and *RPL4* mRNA levels, involving a competition between the dedicated chaperone and the regulatory machinery for binding to nascent Rpl3 or Rpl4, represents for the moment the most plausible model. Assuming that the regulatory signal would be on the mRNA, how would it elicit regulation? As our data show that Not5’s N-terminal domain is not involved, selective recruitment of the Ccr4-Not complex to the *RPL3* or *RPL4* mRNA owing to low A-site codon optimality is highly unlikely (see also our answer to Essential revision point 3). In any case, it is questionable whether one ‘tryptophan’ codon (W109), in the context of otherwise optimal codons, or the combination of an optimal ‘phenylalanine’ and ‘leucine’ codon (F16/L17) could selectively impose regulation to the encoding mRNAs. Moreover, given the close proximity or partial overlap of the dedicated chaperone binding site and the regulatory signal, a positive effect of Rrb1 or Acl4 binding on the stability of the *RPL3* or *RPL4* mRNA cannot be reconciled with a regulatory event involving decoding of the above-mentioned codons as the binding site of the dedicated chaperone would have not at all (Rpl3: residues 1-15) or only partially (Rpl4: residues 43-114) exited from the ribosome. Therefore, it is also difficult to imagine how a potential secondary structure, involving the bases of these codons, could act as a regulatory signal that blocks ribosome movement on these mRNAs. Similarly, how would mutation of these codons reduce recognition by the regulatory machinery, especially considering that the involved ribosome-associated NAC has a well-established role in binding to nascent chains?

Concerning the suggestion to address by cycloheximide treatment whether regulation depends on ongoing translation, see our answer to point 9.

(6) The authors need to address directly the relationship between the system they describe here and the observations in Buschauer et al. Science, 2020 (and references therein) regarding the involvement of Ccr4-Not in monitoring codon optimality at translating ribosomes. They should test the role of the Not5 N-terminal domain in RPL3/4 mRNA regulation and, perhaps using data in Buschauer et al., determine whether ribosome stalling occurs around the regulatory regions they identify.

See our answer to Essential revision point 3.

(7) Related to the point above, the authors could significantly improve their study by either testing the role of NAC subunits in codon optimality monitoring by Ccr4-Not (related to Buschauer et al. Figure 5) or searching for the putative targets of the Rpl3/4 regulatory regions (which they speculate could be Btt1). Both would be even better. A key point that they need to address is whether or not the RPL3/4 regulatory system they describe here is in fact controlled by codon optimality, which might itself be exacerbated by stress, rather than through a specific protein-protein interaction involving the r-protein regulatory regions and/or interactions with the chaperones. It is less clear to me how the latter might be regulated by stress.

As pointed out in the Discussion, we firmly believe that co-translational recognition of nascent Rpl3 or Rpl4 by their dedicated chaperones Rrb1 and Acl4, respectively, represents a plausible and elegant mechanism to rapidly relay and directly couple the status of pre-60S assembly to the rate of Rpl3 or Rpl4 de novo synthesis. By continuously sensing the levels of unassembled Rpl3 and Rpl4, which can be temporarily stored in the nucleus in complex with their dedicated chaperone, the availability of free Rrb1 and Acl4 to capture their nascent r-protein partner determines the fate of the encoding *RPL3* and *RPL4* mRNAs and, thereby, the amount of newly synthesized Rpl3 or Rpl4. Accordingly, only changes in unassembled Rpl3 or Rpl4 that exceed the binding capacity of Rrb1 or Acl4, owing to a specific or more global perturbation of ribosome assembly (stress), will lead to a rapid and strong downregulation of *RPL3* or *RPL4* mRNA levels. In other words, any ribosome assembly defect or stress leading to reduced incorporation of Rpl3 or Rpl4 would lead to a titration of free Rrb1 or Acl4 and their nuclear sequestration in complex with their unassembled r-protein client. As Rrb1 or Acl4 overexpression and inactivation of their binding (H3E mutation in Rpl3 or BI mutations in Rpl4) either positively or negatively affect the abundance of the *RPL3* or *RPL4* mRNAs (Figure 6 and Figures 5A and 4D), it can be inferred that the regulatory mechanism also continuously operates under normal growth conditions. Therefore, such a tight regulation of unassembled Rpl3 or Rpl4 levels is ideally suited to both warrant optimal rates of 60S production, thus avoiding the costs and impact of abortive pre-60S assembly, and prevent cells form the potentially detrimental effect of Rpl3 or Rpl4 aggregation. Evidently, the proposed co-translational regulation mechanism has the advantage of enabling a more rapid response than a transcription-based regulation of protein levels.

We agree with this reviewer that it would be interesting to understand how the regulatory signal on Rpl3 or Rpl4 is deciphered by the regulatory machinery. As already replied above (see answer to Essential revision point 3), our data indicate that the here-described regulatory system does not involve monitoring of codon optimality. Rather, as stated in the Discussion, we envisage that NAC binding to nascent Rpl3 or Rpl4, presumably around the identified key residues (F16/L17 in Rpl3 and W109 in Rpl4), may fortify a stalling activity that is present in the distal region of the minimal regulation-conferring segments. Slowing down translation would then provide the necessary time window to enable the recruitment of the Caf130-associated Ccr4-Not complex and initiate the degradation of the *RPL3* and *RPL4* mRNAs. Despite significant efforts since the discovery of NAC in 1994, only very little is still known about how NAC recognizes and affects the fate of nascent polypeptides (reviewed in Deuerling et al. Cold Spring Harb. Perspect. Biol. 2019, PMID: 30833456). It appears that capturing the weak and transient interactions between NAC and its nascent, physiological substrates on translating ribosomes it technically very challenging. Addressing this issue would probably require the biochemical and structural analysis of robustly stalled nascent polypeptides displaying different lengths of the regulatory signal at the solvent side of the exit tunnel. Considering the enormous challenge and complexity of such experiments, we believe that this task should be part of a thorough follow-up study aimed at deciphering the molecular details of the regulatory mechanism that we have unveiled in this study.

(8) The sentence pg. 26 line 3 doesn't make sense to me ("even though") and I think, as mentioned above, that the authors miss an important point here. Zencir et al. show that RPL3/4 and the other category III genes are much less down-regulated following different forms of stress than are the majority of genes (category I and II), and suggest that this may be due to the fact that they are coupled to specialized chaperone proteins. The authors' findings suggest that RPL3/4 mRNAs are indeed strongly regulated upon stress (though they don't actually examine stress conditions in this study!), but at a post-transcriptional level, namely mRNA stability. This (potential) novel finding should be directly tested by examing RPL3/4 mRNA stability under various stress conditions (e.g. heat shock, TORC1 inhibition, nutrient deprivation, ribosome assembly defects, etc.).

We agree that the “even though” in this sentence is somehow confusing. We have therefore removed it and rephrased the sentence to: “… the complementary action of Ifh1 and Sfp1, which are predominantly required for activation ….”.

We have stated in the Discussion of the submitted manuscript that the *RPL3* and *RPL4A/B* genes are the RPGs whose transcription, while being rather insensitive to Ifh1 depletion or the ribosome assembly stress response (RASTR), shows the highest Sfp1 dependence (Zencir et al. Nucleic Acids Res. 2020; Albert et al. *eLife* 2019) (see page 26, lines 12-15 of the revised manuscript). Zencir et al. have therefore proposed that dedicated chaperones could compensate for the lack of transcriptional downregulation of *RPL3* and *RPL4* upon RASTR. Our study has now revealed how the dedicated chaperones Rrb1 and Acl4 are involved in a sophisticated co-translational regulation mechanism that, by regulating the stability of the *RPL3* and *RPL4* mRNAs, adjusts the expression levels of Rpl3 and Rpl4 to their actual consumption during ribosome assembly. In this context, it is noteworthy that none of the other r-proteins encoded by mRNAs that are transcribed from category III promoters have so far been reported to associate with a dedicated chaperone. Hence, adaptation of their protein levels during stress conditions that do not affect Sfp1 activity is expected to occur *via* alternative mechanisms. Notably, amongst these Rps22, Rps28, and Rpl1 have been implicated in the regulation of *RPS22B*, *RPS28B*, and *RPL1B* mRNA levels (Roy et al. Commun. Biol. 2020, PMID: 33311538; He et al. Mol. Cell. Biol. 2014, PMID: 24492965).

As already mentioned above (see answer to point 7), the proposed regulatory system appears to continuously operate even under normal growth conditions, but would also be ideally suited to rapidly respond to any stress leading to the accumulation of unassembled Rpl3 or Rpl4 and, thereby, to the titration of Rrb1 or Acl4. We have now shown that the regulatory machinery, as assessed in *∆caf130* mutant cells, does not play a role in transcription initiation at the promoters of the *RPL3* and *RPL4A/B* genes (see answer to Essential revision point 2). Our new data also show that rapamycin treatment reduces transcription initiation at all tested RPG promoters to a similar extent in wild-type and *∆caf130* mutant cells (see new Figure 1—figure supplement 2A). Accordingly, the levels of these RPG mRNAs are lower in rapamycin-treated cells, but an around two-fold higher abundance of the *RPL3* and *RPL4* mRNAs can still be observed in *∆caf130* mutant cells (see new Figure 1—figure supplement 2B), suggesting that regulation still occurs under this ‘stress’ condition. However, we believe that addressing whether regulation remains active under the proposed variety of stress conditions is beyond the scope of this study.

(9) Related to the above, the authors never test experimentally the notion that Ccr4-Not regulation of RPL3/4 mRNAs is part of the ribosome assembly stress response described by Albert et al. (2019) and Tye et al. (2019). The authors focus primarily on the relationship between chaperone levels and RPL3/4 mRNA levels without really addressing the possible physiological role of Ccr4-Not mRNA regulation with respect to Rpl3/4. A key feature of this response is that it is abrogated by cycloheximide treatment, differentiating it from some other stress responses. Dampening of RPL3/4 mRNA regulation by cycloheximide would also provide support for the notion that the polypeptide and not the RNA of the regulatory region is responsible for regulation.

In our experiments, we observe that regulation of *RPL3* and *RPL4* mRNA levels occurs under normal growth conditions (growth at 30ºC in rich YPD medium), as revealed by their increased abundance in the absence of individual components of the regulatory machinery. Importantly, all these mutants (*∆caf130*, *∆cal4*, *∆btt1/∆egd1*, *∆egd2*, and *not1.163C*) do not display any growth defects at 30ºC; hence, regulation appears to occur in the absence of perturbed ribosome assembly or other cellular stresses. Moreover, all these mutants exhibit synthetic lethality with the *∆tom1* allele and, except *∆cal4*, also a temperature-sensitive growth phenotype, strongly suggesting that tight regulation of Rpl3 and/or Rpl4 levels becomes essential for cell survival under proteostatic stress conditions. Accordingly, it appears that the different processes (the here-described co-translational regulation of *RPL3* and *RPL4* mRNA levels, RASTR/RPAS, and ERISQ) continuously cooperate to prevent proteotoxic stress arising from the aggregation of excessively abundant, unassembled r-proteins. An important readout of RASTR is the downregulation of Ifh1-dependent RPG transcription. However, as we now show in the revised manuscript, cells experiencing deregulated expression of Rpl3 and/or Rpl4 (*∆caf130* mutant grown at 30ºC) do not exhibit reduced transcription at Ifh1-dependent category I (*RPL30* and *RPL39*) and II (*RPS20*) RPG promoters (see also answer to Essential revision point 2); thus, strengthening the notion that co-translational regulation of *RPL3* and *RPL4* mRNA levels occurs independently of RASTR. However, it is reasonable to assume that these two processes are intimately interconnected. As already mentioned above (see answers to point 7 and 8), any perturbation of ribosome assembly that would be sufficiently severe to lead to the accumulation of unassembled Rpl3 or Rpl4 and, hence, to the limited availability of free Rrb1 or Acl4 is expected to increase negative regulation of *RPL3* or *RPL4* mRNA levels. Reciprocally, deregulated expression of Rpl3 and/or Rpl4 in cells simultaneously experiencing proteostatic stress (ERISQ inactivation or growth at 37ºC) is expected, owing to the massive aggregation of Rpl3 and/or Rpl4 and the entailed insolubility of many other r-proteins (Figure 7D), to result in the activation of RASTR and the transcriptional downregulation of Ifh1-dependent RPG transcription. It will be interesting to learn whether aggregation of Rpl3 and/or Rpl4 is sufficient to recruit Ifh1 into aggregates and to reduce RPG transcription from category I and II promoters. We have planned to carefully address these predictions in future studies.

Based on the results of our study, especially the finding that lethality of *∆caf130/∆tom1* cells is efficiently suppressed by simultaneously lowering Rpl3 (transcription of *RPL3* from weaker *RPL4B* promoter) and Rpl4 (deletion of major *RPL4A* copy) levels (see Figure 7C), we infer that the physiological role of the Caf130-associated Ccr4-Not complex consists in the negative regulation of the *RPL3* and *RPL4* mRNAs. As outlined in the Discussion (see also answer to point 7), coupling the availability of the dedicated chaperones Rrb1 or Acl4 to co-translational regulation of *RPL3* or *RPL4* mRNA levels represents an elegant and simple mechanism to survey the levels of unassembled Rpl3 and Rpl4 and, thus, the status of early pre-60S assembly. Notably, this mechanism has the advantage to enable a rapid cellular response both to insufficient or exceeding amounts of free Rpl3 or Rpl4. In our opinion, it is more difficult to envisage how the Caf130-associated Ccr4-Not complex and/or the presumed NAC-dependent recognition of nascent Rpl3 or Rpl4 could be specifically and swiftly (in)activated to rapidly respond to changes in the abundance of unassembled Rpl3 or Rpl4. At present, we presume that NAC and the Ccr4-Not complex represent constitutive elements of the regulatory system that, in case of insufficient Rrb1 or Acl4 availability, promote in a default manner degradation of the *RPL3* and *RPL4* mRNAs; however, it is possible that future insights may challenge this view.

Considering that cycloheximide (CHX) treatment is known to activate TORC1 (see for example Binda et al. Mol. Cell 2009, PMID: 19748353; Santos et al. Nucleic Acids Res. 2019, PMID: 30916348), the evaluation of its specific effects on *RPL3* and *RPL4* mRNA regulation may not be straightforward. Additionally, it also known that mRNAs undergo widespread Xrn1-depenendent co-translational degradation, a phenomenon that is exploited to study ribosome dynamics by a method called 5PSeq (Pelechano et al. Cell 2015, PMID: 26046441; Pelechano et al. Nat. Protoc. 2016, PMID: 26820793; Tesina et al. Nat. Struct. Mol. Biol. 2019, PMID: 30911188). According to our pilot experiments with wild-type cells, the non-normalized levels of the *RPL3* and *RPL4* mRNA, but also of the *RPL5* mRNA, similarly increase up to around two-fold within 30 min after CHX addition, suggesting a possible influence of the above-mentioned effects on TORC1 activity or general co-translational degradation. To at least exclude potentially interfering effects of increased *RPL3* or *RPL4* transcription, it may therefore be necessary to drive their transcription from a promoter that is not affected by changes in TORC1 activity. Moreover, to avoid as much as possible the potential influence of differences in the steady-state abundance of the *RPL3* and *RPL4* mRNAs between wild-type and, for example, *∆caf130* cells before CHX addition, it might be advantageous to initiate the synthesis of their mRNAs from an inducible promoter.

Bearing these potential limitations in mind, we have decided to perform the experiment by assessing the effects of CHX on the abundance of a reporter mRNA encoding the minimal regulation-conferring Rpl4a segment (residues 78-139), flanked by an N-terminal TAP tag and a C-terminal GFP moiety, and either containing the BI (enabling maximal negative regulation) or the BIV (disabling negative regulation) mutations (see Figure 4F). To enable rapid induction of mRNA production that, at the same time, does not depend on de novo protein synthesis, we placed the reporter constructs under the transcriptional control of the inducible *GAL1* promoter, which is efficiently induced under “reinduction memory” conditions (reviewed in Stockwell et al. Mol. Biosyst. 2015, PMID: 25328105). Specifically, cells were pre-grown in synthetic medium containing galactose, transferred for 1.5 h to glucose-containing medium (time point 0), and then, after one wash in sugar-free medium, placed into medium containing galactose. CHX was added to one half of the culture at a final concentration of 200 µg/ml. Then, aliquots were taken after 10, 20, 40, 60, and 90 min of culturing in galactose-containing medium +/- CHX and extracted RNAs were analyzed by qRT-PCR. In line with active translation being required for efficient negative regulation, this experiment revealed that addition of CHX eliminated the difference in abundance between the reporter mRNAs containing the BI or BIV mutations within the minimal regulation-conferring coding sequence and, moreover, also increased their overall abundance (see new Figure 4—figure supplement 2). We have incorporated the new text describing these findings on page 16, lines 25-28, of the revised manuscript.

Additional comments and suggestions:(1) In the Abstract, pg. 2 line 8 "prevents" should probably be changed to "reduces", since they show that even in conditions of normal chaperone levels there is still degradation.

Thank you very much for this suggestion. We have replaced “prevents” by “reduces”.

(2) Pg. 6, lines 13-16: simplify to something like "To gain further insight into this observation we isolated a large number of acl4Δ and acl4Δ/rpl4Δ suppressors and identified the causative mutations by whole-genome sequencing".

As suggested, we have rephrased this sentence: “To unravel the reason for this observation, we isolated a large number of ∆acl4 and ∆acl4/∆rpl4a suppressors and identified causative candidate mutations by whole-genome sequencing.” (see page 6, lines 13-15 of the revised manuscript).

(3) Not clear how causative mutations were identified without pooling and sequencing of sup+ and sup-isolates from a single backcross (pooled linkage analysis). Were there really no other polymorphisms in the suppressor strains.

Thank you very much for this remark. We have noticed that it would be more accurate to state that (see page 6, line 16 of the revised manuscript) were revealed by bioinformatics analysis of the sequenced genomes.

Causative candidate mutations were inferred ‘47 different candidate mutations’ by the recurring presence of independent mutations within the same gene in several different suppressor genomes (applicable for *CAF130*, *CAL4*, and *NOT1*). The identified mutation in the *RPL4A* gene was considered as a probable candidate mutation owing to the previously established link between Acl4 and Rpl4. In many cases, the assigned candidate mutation was the only mutation that could be identified with high confidence. Moreover, the genomes of several suppressors, which were in each case derived from the same parental strains, contained the same additional mutation(s), indicating that these mutations were already present before suppressor isolation. However, some suppressor genomes also contained further additional mutations (polymorphisms) that were exclusively present in one given suppressor strain. Importantly, we have shown that absence of Caf130 (*∆caf130*), Cal4 (*∆cal4*), or Not1’s N-terminal domain (*not1.163C*) conferred *∆acl4* suppression (see Figures 1A, 1B, and 2D), which provides strong evidence that the presumed candidate mutations within *CAF130*, *CAL4*, and *NOT1* are indeed the causative mutations. In the case of the identified *rpl4a.W109C* mutation, we have confirmed that the integrated *rpl4a.W109C* allele indeed suppresses the growth defect of *∆acl4* cells (see Figure 4B).

(4) It would be useful to show a cartoon of the NAC and Ccr4-Not complexes in Figure 1 to help the reader to follow the subsequent analysis.

A cartoon showing the NAC and Ccr4-Not complexes is part of the model illustrating the proposed regulatory mechanism (see Figure 8) In addition, a schematic representation of Not1’s domain organization and the surfaces mediating the interactions with components of the Ccr4-Not complex are shown in Figure 2A.

(5) In section "RPL4 mRNA levels are increased…" (pgs. 7-8) it would be interesting to know, as pointed out above, which of these effects are at the level of mRNA production versus degradation or stabilization. The effect of cal4Δ is hardly small, with both up and down-regulated genes. Why is that?

Concerning the first raised point (mRNA production versus degradation), see our answer to Essential revision point 2.

Unfortunately, we do not have a good explanation for the broader distribution of both up- and downregulated mRNAs in *∆cal4* cells (Figure 1H). One possible reason could be that the RNA-Seq analysis was done with three biological replicates for each mutant condition; therefore, differences in variability between triplicates of the different conditions could result in this observation. Importantly, however, the *RPL3* and/or *RPL4A/B* mRNAs do clearly not cluster with all other RPG mRNAs when full suppression of the *∆acl4* growth defect is observed; therefore, the data are good enough to confidently conclude that none of the remaining RPG mRNAs are subjected to a similar regulatory mechanism as the *RPL3* and *RPL4A/B* mRNAs. Nevertheless, it is possible that our analysis may have missed additional, non-RPG targets of the uncovered regulatory machinery; these would, however, be less important as our genetic data strongly indicate that the *RPL3* and *RPL4A/B* mRNAs are the physiologically relevant targets (see Figure 7B, C).

(6) The sentences on pg. 25, lines 11-14 and 27-31, are awkward and need to be rewritten for clarity ("only present in insufficient abundance"…?; too many subordinate clauses in the latter sentence).

Thank you very much for these specific recommendations how to improve the clarity of the text. Accordingly, we have rephrased these two sentences to: “ … and, therefore, can no longer bind to nascent Rpl3 or Rpl4 in a timely manner.” (See page 25, lines 11-14 of the revised manuscript) and “The need for such a tight regulation becomes apparent when Rpl3 and/or Rpl4 expression is deregulated, such as in the absence of Caf130 or Cal4, and cells cannot clear these excessively produced r-proteins via their Tom1-mediated ubiquitination (ERISQ pathway) and subsequent proteasomal degradation.” (see page 25, lines 27-29 of the revised manuscript).

Reviewer #4 (Recommendations for the authors):1. Figure 1: In the tree plots – indicating the zero with a dashed line might be helpful. This type of data is often presented as a volcano plot – with significance (adjusted P values) rather than counts on the axis. I quite like this version, however, in the absence of P-values for named gene classes in the graph, the authors should consider stating these for the most important changes, especially when marginal (eg Rpl3 and Rpl4 in ∆btt1, ∆egd1).

As suggested, we have indicated in all tree plots the zero with a dashed line. Moreover, we have also indicated the adjusted p-values for the changes in RPL3, RPL4A, RPL4B, UBI4, and YAP1 mRNA levels in parentheses behind these gene names in all tree plots. The adjusted p-values for all genes can be found in Supplementary file 4.

2. P12, P13; Figure 3: "Btt1 appears to exclusively interact with either Caf130 or Egd2, butnot simultaneously with both, since neither Caf130 nor Egd2 could co-purify each other…." Could this be an efficiency issue? I did not see any indication of the pairwise coprecipitation efficiency but according to the legend ~200 fold more precipitate was loaded relative to input, so the stoichiometry is unclear. If – for example – 10% of each protein is in complex with Btt1, only 1% would coprecipitate.

Thank you very much for pointing this out. Indeed, we cannot rule out that the lack of an observable interaction between Caf130 and Egd2 in the co-immunoprecipitation analyses could be due an efficiency issue. We have therefore removed this sentence from the revised manuscript. Moreover, taking this possibility into account but also considering that our Y2H data clearly show that both Caf130 and Egd2 interact with the NAC domain of Btt1, we have kept but modified the sentence suggesting that Btt1 associates with either Caf130 or Egd2 in a mutually exclusive manner (page 13, lines 19-21 of the submitted manuscript): “The finding that both Caf130 and Egd2 bind to the NAC domain of Btt1 corroborates, as already indicated by the lack of a detectable co-precipitation between these two (Figure 3C,F), a model in which Btt1 associates in a mutually exclusive manner with either Caf130 or Egd2.” (see page 13, lines 20-22 of the revised manuscript).

3. Figure 7S4: The general approach of using MS to assess the proteins present in the pellet fraction is good. However, the version used, involving the cutting of visible gel bands is surprising. By definition, most proteins detected during depletion are absent from the 0 h sample, since the corresponding regions were not analysed. A labelling based method, eg SILAC or iCAT, allowing quantitation of relative recovery for all proteins might have been better. The data presented are, however, sufficient to support the major conclusions, that protein aggregation is increased and multiple proteins are recruited.

We agree that obtaining a quantitative overview of the aggregated proteins would be very informative. Indeed, we have planned to carry out such analyses in the framework of a follow-up study, which is also intended to reveal co-aggregation of Ifh1 in different mutants and conditions. The intention behind performing the analysis in this way (fully migrated, Coomassie-stained gels) was to reveal whether Rpl3 and Rpl4 were, as suggested by the genetic experiments and the fluorescence microscopy data, indeed among the most excessively aggregating proteins when their expression is deregulated in Tom1-depleted cells. As affirmed by this reviewer, we also believe that the chosen approach is sufficient to show that massive Rpl3 and, to a lesser extent, Rpl4 aggregation precedes and, presumably, promotes the aggregation of multiple additional proteins, prominently including r-proteins, ribosome biogenesis factors, general chaperones, and translation factors.

4. Figure 8: What does the exclamation mark indicate?

The exclamation mark indicates that deregulated Rpl3 and/or Rpl4 expression together with the simultaneous inability to degrade these excessively produced r-proteins via the ERISQ pathway results in a massive Rpl3 and/or Rpl4 aggregation and a proteostatic collapse. To refer to the indicated exclamation mark, we have changed the text in the legend to Figure 8 as follows: “Also included in the model and highlighted by an exclamation mark is the finding that ….”.

5. In the PDF, commas separating thousands in numbers are inverted.

Thanks for pointing this out (it was inadvertently done the Swiss way). We have changed this throughout.

6. A great deal of data are presented, but even allowing for this the text is very long, with frequent repetitions in the text and again in the figure legends and Discussion. At over 8 pages, the Discussion seems particularly long. A result is that only afficionados may make it thought to the end.

We have tried to reduce unnecessary repetitions across the different sections (Results, Figure legends, and Discussion) and we have also removed, as suggested by reviewer #3, the Discussion paragraph “Possible conservation of the regulatory process” (see answer to Essential revision point 1). We believe, however, that the wealth and complexity of the data required to provide strong evidence for the proposed model, but also to illuminate important additional aspects such as the intriguing finding that Not1 is naturally produced as two translational isoforms, make it necessary to carefully guide readers through the different aspects and conclusions of our study. We fear that a less detailed description of the rationale, experimental design, involved factors, and conclusions would make it even harder for prospective readers to follow and understand our study.